# Ethylene modulates cell wall mechanics for root responses to compaction

Jiao Zhang[1], Zengyu Liu[1], Edward J. Farrar[2], Minhao Li[1,3], Hui Lu[1,3], Zhuo Qu[1], Osvaldo Chara[4,5], Nobutaka Mitsuda[6], Shingo Sakamoto[6], Feiyang Xue[7], Qiji Shan[8], Ya Yu[8], Jingbin Li[1], Xiaobo Zhu[9], Mingyuan Zhu[10,11], Jin Shi[1], Lucas Peralta Ogorek[2], Augusto Borges[12], Malcolm J. Bennett[2], Wanqi Liang[1,13 ✉], Bipin K. Pandey[2 ✉], Dabing Zhang[1,15] & Staffan Persson[1,14 ✉]

Soil stresses affect crop yields and present global agricultural challenges[1]. Soil compaction triggers reduction in root length and radial expansion driven by the plant hormone ethylene[2]. Here we report how ethylene controls cell wall biosynthesis to promote root radial expansion. We demonstrate how soil compaction stress, via ethylene, upregulates *Auxin Response Factor1* in the root cortex, which represses cellulose synthase (CESA) genes. CESA repression drives radial expansion of root cortical cells by modifying the thickness of their cell walls, which results in a thicker epidermis and thinner cortex. Our research links ethylene signalling with root cell wall remodelling, and reveals how dynamic regulation of cellulose synthesis controls root growth in compacted soil.

The reliance of modern agriculture on mechanization is causing soil degradation and compaction, which affect root growth and crop yield[1]. Plant roots expand radially when encountering compacted soil conditions, leading to shorter and thicker roots[3]. Radial root swelling is mainly due to expansion of cortex cell layers[4] that can cause soil fissures that aid soil penetration[3,5]. This adaptive response is driven by entrapment of the plant volatile hormone ethylene around the root as compaction reduces gas diffusion through the soil[2,6]. Despite this mechanistic insight, how ethylene controls cell wall remodelling to enable root radial expansion remains unclear.

Primary plant cell walls, and in particular the organization and amount of the load-bearing polymer cellulose, support anisotropic cell expansion and are key to organ growth[7]. Cellulose is produced at the plasma membrane by large, multimeric protein complexes known as CESA complexes (CSCs)[8]. In rice, OsCESA1, OsCESA3, OsCESA5, OsCESA6 and OsCESA8 form heterotrimeric CSCs that are active in growing root cells during primary wall synthesis[9]. When roots encounter soil compaction, the arrangement of cellulose microfibrils in their cortical cell walls is altered[10]. Although the mechanical stiffness of cell walls is crucial for anisotropic cell expansion, little is known about how cell wall properties regulate root penetration in compacted soil.

## Reduced cellulose aids root penetration

To explore whether changes in cellulose synthesis affect the ability of roots to penetrate compacted media, we initially grew rice seedlings for 5 days on normal (0.3%) and dense (0.6%) agar media, with and without varying concentrations of indaziflam. Indaziflam is a cellulose biosynthesis inhibitor that suppresses cellulose production by triggering an atypical increase in CSC density at the plasma membrane in plant cells[11,12]. Surprisingly, whereas high concentrations (250 pM) of the inhibitor suppressed root growth (Extended Data Fig. 1a,b), a lower concentration (150 pM) promoted root penetration through the dense growth medium (Extended Data Fig. 1a,b). To validate our results in soil conditions, we adopted a genetic approach and mutated *OsCESA6* using CRISPR–Cas9 technology. *OsCESA6* encodes a redundant CESA subunit (Extended Data Fig. 2a) that can be replaced by other CESAs in the CSCs[13]; mutated seedlings therefore exhibit mild cellulose developmental defects[14]. We identified a *cesa6* mutant line that carried a deletion of an adenine in the third exon, causing early termination of gene transcription (Extended Data Fig. 2b). Non-invasive X-ray computed tomography (CT) imaging revealed that *cesa6* mutant roots grew longer than wild-type roots in compacted (1.6 g cm$^{-3}$) than non-compacted (1.2 g cm$^{-3}$) soil conditions, as well as on dense agar media (Fig. 1a,b and Extended Data Fig. 2c,d), corroborating our results with indaziflam. Notably, whereas *cesa6* mutant plants typically have longer and swollen roots compared with wild type when grown under compacted conditions, the mutant and wild-type roots were of equal lengths when grown in germination pouches (Fig. 1c,d). These results indicate that minor perturbations in cellulose biosynthesis affect the ability of plant roots to extend under varying soil and media density conditions.

[1]Joint International Research Laboratory of Metabolic and Developmental Sciences, School of Life Sciences and Biotechnology, Shanghai Jiao Tong University, Shanghai, China. [2]Plant and Crop Science Division, School of Biosciences, University of Nottingham, Sutton Bonington, UK. [3]Department of Bioinformatics and Biostatistics, School of Life Sciences and Biotechnology, Shanghai Jiao Tong University, Shanghai, China. [4]Animal Sciences Division, School of Biosciences, University of Nottingham, Nottingham, UK. [5]Instituto de Tecnología, Universidad Argentina de la Empresa, Buenos Aires, Argentina. [6]Biomanufacturing Process Research Center, National Institute of Advanced Industrial Science and Technology (AIST), Tsukuba, Japan. [7]The Core Facility and Service Center (CFSC) for the School of Life Sciences and Biotechnology, Shanghai Jiao Tong University, Shanghai, China. [8]The Instrumental Analysis Center of Shanghai Jiao Tong University, Shanghai, China. [9]Institute of Crop Science, College of Agriculture and Biotechnology, Zhejiang University, Hangzhou, China. [10]Department of Biology, Duke University, Durham, NC, USA. [11]Howard Hughes Medical Institute, Duke University, Durham, NC, USA. [12]Graduate School of Quantitative Biosciences, Ludwig Maximilian University, Munich, Germany. [13]Yazhou Bay Institute of Deepsea Sci-Tech, Shanghai Jiao Tong University, Sanya, China. [14]Copenhagen Plant Science Center (CPSC), Department of Plant & Environmental Sciences (PLEN), University of Copenhagen, Frederiksberg, Denmark. [15]Deceased: Dabing Zhang. ✉e-mail: wqliang@sjtu.edu.cn; bipin.pandey@nottingham.ac.uk; Staffan.persson@plen.ku.dk

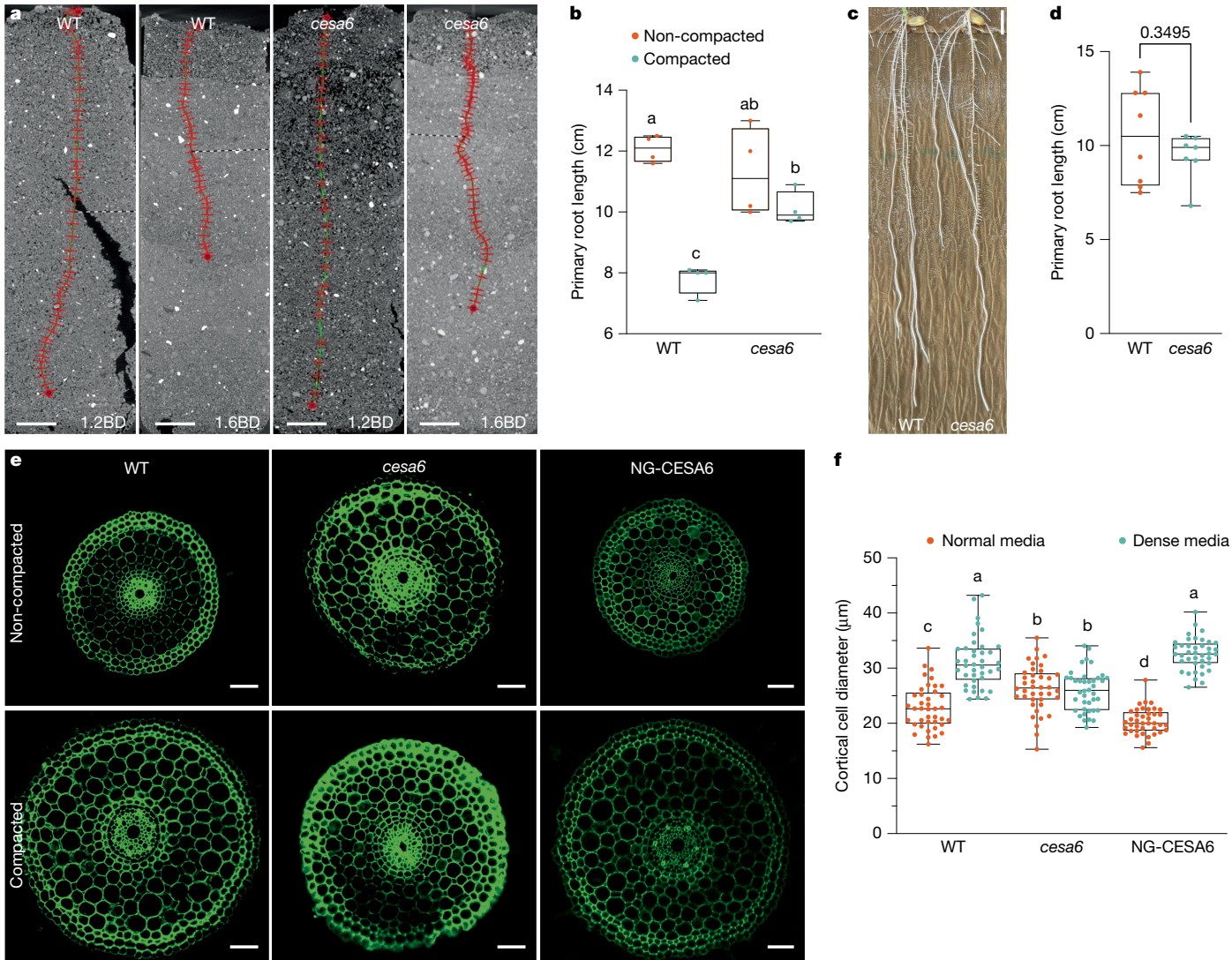

**Fig. 1 | *OsCESA6* negatively regulates rice root penetration in compacted soil. a**,**b**, Images (**a**) and primary root length (**b**) of 5-day-old wild-type (WT) and *cesa6* mutants grown in non-compacted (1.2 g cm⁻³ bulk density (1.2BD)) and compacted (1.6 g cm⁻³ bulk density (1.6BD)) soil conditions. Images are representative of four independent CT scans. Statistical analysis was conducted using two-way ANOVA followed by Tukey test, with different letters indicating significant differences ($P < 0.05$). $P$ (interaction) = 0.0032. Scale bars, 10 mm. **c**,**d**, Images (**c**) and primary root length (**d**) of 7-day-old wild-type and *cesa6* mutants grown in germination pouches. The experiment was repeated three times, consistently yielding similar trends ($n$ = 8 plants). Statistical analysis was performed using Welch's *t*-test (two-sided $P$ value is shown) with a significance threshold of 0.05, based on data from one representative trial. Scale bar, 1 cm. **e**, Root cross-sections of 5-day-old wild-type, *cesa6* and NG-CESA6 (*cesa6* complementation lines) seedlings from the elongation zones. Plants were grown in normal (0.3%) and dense (0.6%) media conditions. Sections were prepared using a vibratome at 50 μm thickness and observed under UV laser at 405 nm using an SP5 confocal microscope. Scale bars, 50 μm. **f**, Cortical cell diameters from wild-type, *cesa6* and NG-CESA6 roots, measured from sections as described in **e**. Measurements were taken from 40 cells across 5 sections for each genotype. The experiment was repeated three times, consistently yielding similar trends. Statistical analysis was performed using two-way ANOVA followed by Tukey test, with different letters indicating significant differences ($P < 0.05$). $P$ (interaction) < 0.0001. Based on data from one representative trial. Box plots in **b**,**d**,**f** show median (centre line), 25th–75th percentiles (box) and minimum to maximum range (whiskers).

We next examined root cross-sections of *cesa6* and indaziflam-grown seedlings and found that they exhibited larger cortical cell diameters than wild-type seedlings grown in normal conditions, but similar dimensions in dense media conditions (Fig. 1e,f and Extended Data Fig. 1c,d). These growth phenotypes were rescued in the *cesa6* mutant by complementation using a mNeonGreen-tagged *CESA6* construct driven by its native promoter[15] (NG-CESA6; Fig. 1e,f and Extended Data Fig. 2c,d).

## OsARF1 binds CESA promoters

To establish a connection between the changes in cellulose synthesis and upstream signals such as ethylene, we screened for potential regulators of cellulose synthesis. Genes associated with cellulose synthesis tend to be co-expressed (Extended Data Fig. 3a), indicating that a common set of regulators (that is, transcription factors) may control the process, similar to metabolic regulons[16,17]. Therefore, we conducted a comprehensive yeast one-hybrid (Y1H) assay using 9 promoters from co-expressed core primary wall cellulose synthesis-related rice genes as baits and 1,143 rice transcription factors as preys (Supplementary Data 1 and Supplementary Tables 1 and 2). By exclusively considering transcription factors that bind to multiple promoters, we aimed to minimize false positives and enrich for 'true' transcription factors that regulate primary wall cellulose synthesis. From these assays, we found several ethylene response factors (ERFs), including OsERF34 and

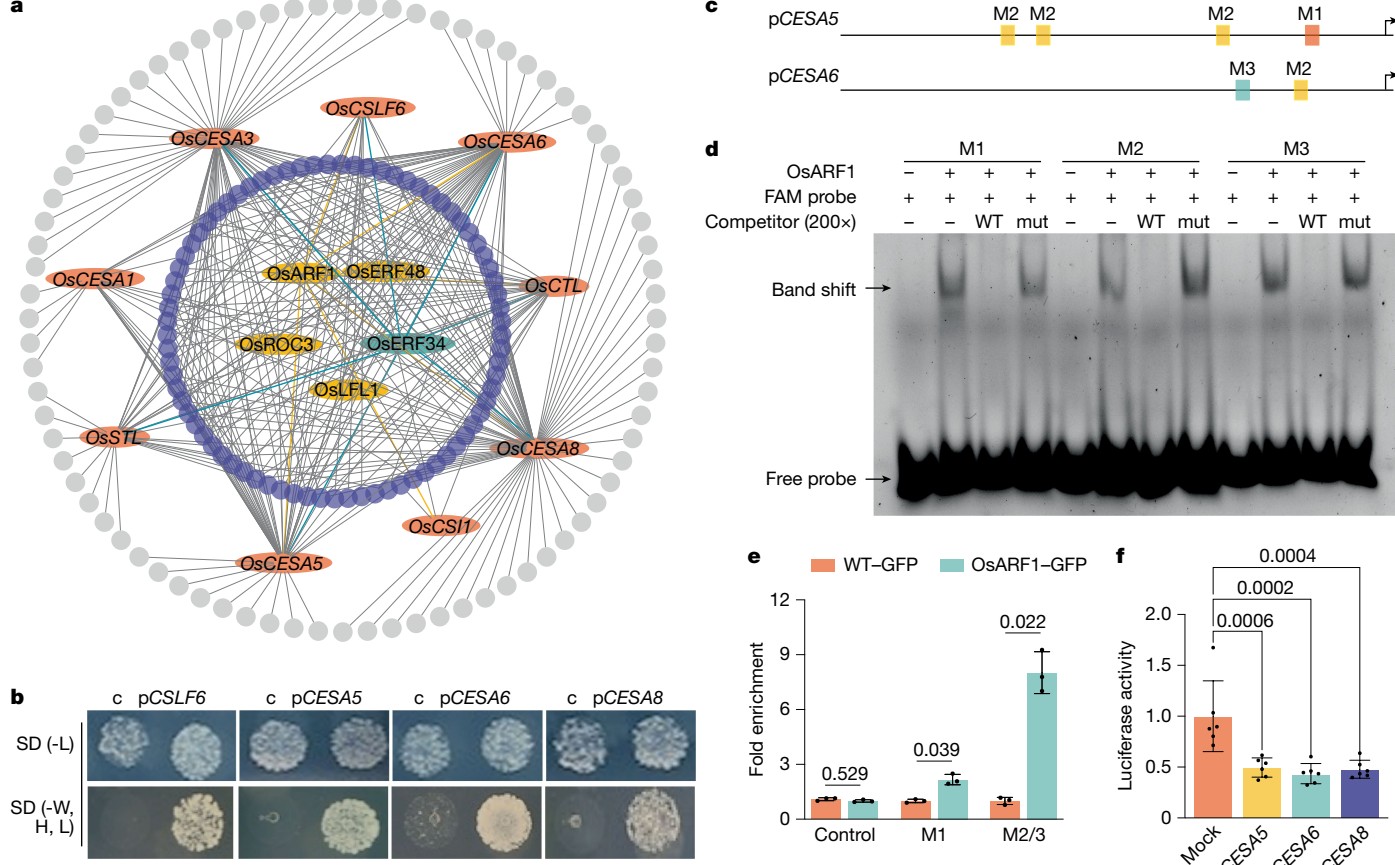

**Fig. 2 | OsARF1 directly binds to several primary wall cellulose-related promoters. a**, Promoters of *OsCESA1*, *OsCESA3*, *OsCESA5*, *OsCESA6*, *OsCESA8*, *OsCSLF6*, *OsCSI1*, *OsSTL* and *OsCTL* (orange ovals) were used as baits in Y1H screens against a library of 1,143 rice transcription factors. Grey lines represent interactions between transcription factors and baits. The screen identified transcription factors with diverse binding patterns, from multiple- (blue, 8 promoters; yellow, 6 promoters) to intermediate- (purple: 2–5 promoters) or single- (grey) promoter binding. **b**, Y1H assays demonstrate that OsARF1 can directly bind to the promoters of *OsCESA5*, *OsCESA6*, *OsCESA8* and *OsCSLF6*. c, empty vector control; SD, synthetic dextrose minimal medium; -L, leucine-deficient; -W, H, L, tryptophan, histidine and leucine-deficient. **c**, Analysis of OsARF1-binding *cis*-elements on the promoters of *OsCESA5* and *OsCESA6*. Boxes denote AuxRE-L (TGTCNN) *cis*-elements. M1 represents TGTCACCGACA, M2 represents CCGACA and M3 represents TGTCAC. **d**, EMSA evaluating OsARF1 binding to probes M1–M3 from **c**, with competition (200-fold excess of non-labelled probes) and mutated probes (TGTCNN to AAAAAA). Probes were 5'-labelled with fluorescein amidite (FAM) and visualized through the Cy2 channel (repeated three times). **e**, ChIP–qPCR experiments confirm in vivo binding of OsARF1 to M1, M2, M3 and a negative control (a sequence in promoters but without AuxRE-L binding motifs). WT-GFP, RUBQ2prom::free-GFP; OsARF1-GFP, OsARF1prom::gOsARF1-GFP; in wild-type background. Results are presented as mean ± s.d., based on one of three biological replicates (two-sided Welch's *t*-test with Bonferroni–Dunn method; adjusted *P* values are shown). **f**, Transient effector–reporter assays in rice protoplasts showed OsARF1-mediated transcriptional regulation (repeated twice, with similar trends). In one representative trial, six biologically independent samples were analysed. Individual data points are represented by dots on the graph (two-sided *P* value ranges of Dunnett test). Bars show mean ± s.d.

OsERF39, which bind to eight and five promoters, respectively (Fig. 2a). These ERFs are the closest homologues of *Arabidopsis* ERF35[18], the main primary cell wall-regulating transcription factor in *Arabidopsis* (Extended Data Fig. 3b), thereby confirming the validity of our approach to identify primary cell wall and cellulose synthesis regulators.

In our Y1H screen, we also observed that auxin response factor 1 (OsARF1) bound six of the promoter baits (Fig. 2a, Supplementary Data 1 and Supplementary Table 2). We verified the interactions between OsARF1 and the bait promoters using pairwise Y1H assays, electrophoretic mobility shift assay (EMSA) against potential ARF *cis*-elements and chromatin immunoprecipitation with quantitative PCR (ChIP–qPCR) experiments (Fig. 2b–e). These data corroborated that OsARF1 interacts with promoters of several primary wall CESA genes in rice. Next, we analysed whether OsARF1 could repress or activate the CESA promoters using a transient effector–reporter assay system in protoplasts extracted from rice leaves, and found that OsARF1 repressed CESA promoter activity (Fig. 2f). Notably, *OsARF1* is induced by compacted soil and media conditions (Fig. 3a,b), potentially linking cellulose synthesis regulation and soil compaction. A GUS reporter fusion revealed that *OsARF1* was expressed in the stele when grown in control conditions, but its expression was markedly induced across other root elongation zone tissues, particularly in the cortex of early elongation zone, in response to dense agar media (Fig. 3a), which we confirmed by RT–qPCR analysis (Fig. 3b). Therefore, we selected *OsARF1* for further analysis.

## OsARF1 controls root penetration via CESA

To investigate OsARF1 function in root growth in compacted soil and media, we generated the *osarf1* mutants *arf1-1* and *arf1-2* (Extended Data Fig. 4a) using CRISPR–Cas9 and *OsARF1* overexpression lines (OE-ARF1; Extended Data Fig. 4d). We first examined the expression patterns of CESA genes in the roots of both the CRISPR-generated mutants and overexpression lines. The expression levels of *CESA5*, *CESA6* and *CESA8* were significantly higher in the *arf1-1* mutant but, with the exception of *CESA5*, were suppressed in OE-ARF1 lines. These findings substantiate

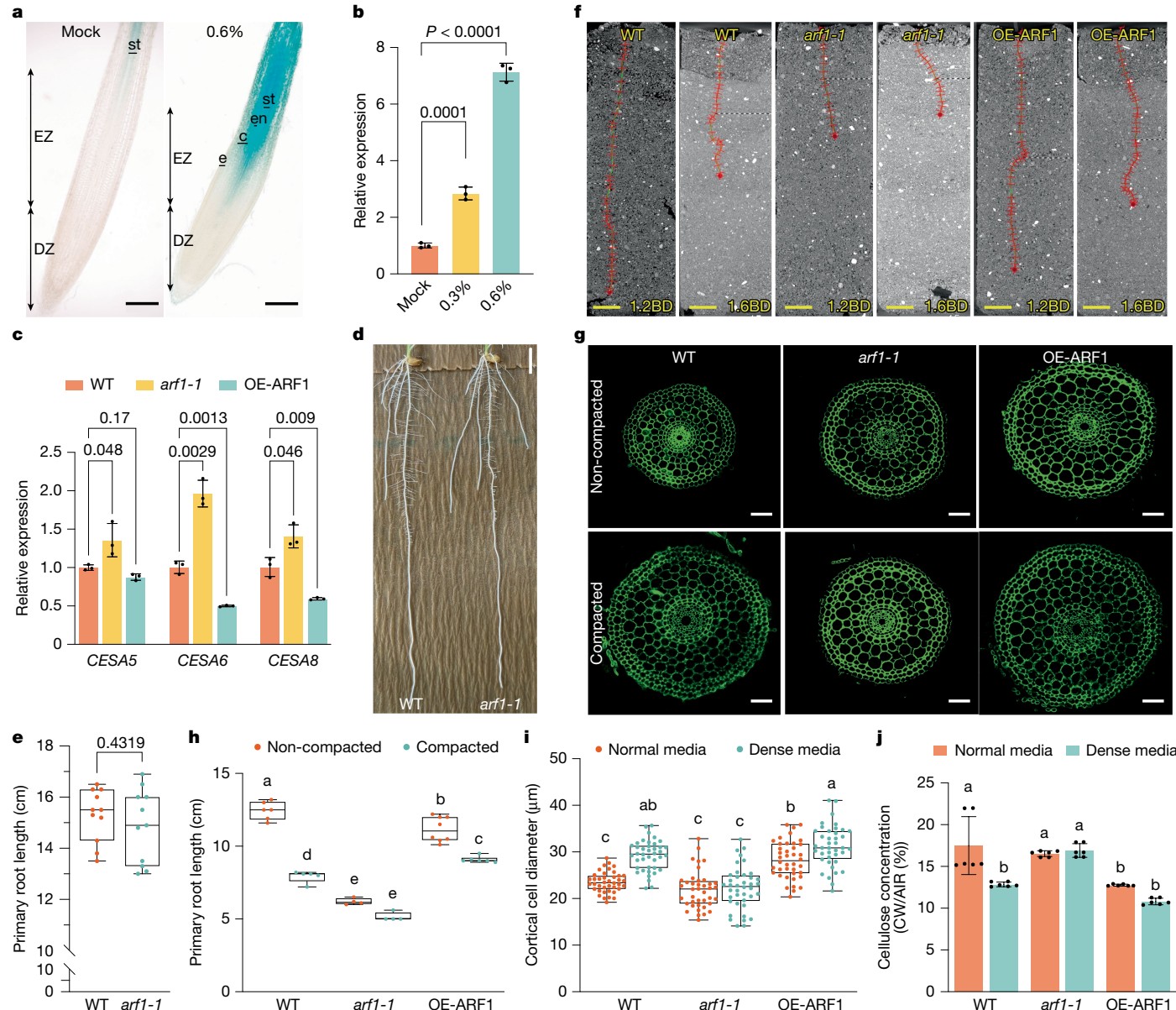

**Fig. 3 | OsARF1 enhances root penetration in compacted soils. a**, *ProOsARF1*: GUS expression in mock and dense media for 48 h (*n* = 18). Longitudinal sections (50 μm thick) were prepared by vibratome. Scale bars, 200 μm. c, cortex; DZ, division zone; e, epidermis; en, endodermis; EZ, elongation zone; st, stele. **b**, *OsARF1* expression in wild-type root elongation zone in mock, normal (0.3%) and dense (0.6%) media. Statistics from three biological replicates (two-sided one-way ANOVA, *P* value of Dunnett test). **c**, Expression of *CESA5*, *CESA6* and *CESA8* in wild-type, *arf1-1* and OE-ARF1 roots. Statistics from three biological replicates (two-sided Welch's *t*-test, *P* value of Dunnett test). **d**,**e**, Images (**d**) and primary root length (**e**) of nine-day-old wild-type and *arf1-1* plants. Statistics from one of 3 replicates (*n* = 11 plants, *P* values from two-sided Welch's *t*-test). Scale bar, 1 cm. **f**, CT scans of primary roots in wild-type, *arf1-1* and OE-ARF1 seedlings grown for five days in non-compacted and compacted soils.

Scale bars, 1 cm. **g**, Confocal images of root cortical diameter in elongation zones from 5-day-old wild-type, *arf1-1* and OE-ARF1 seedlings grown under normal and dense media. Scale bars, 50 μm. **h**, Primary root lengths (*n* = 4) from **f**. Statistical analysis conducted using two-way ANOVA followed by Tukey test. *P* (interaction) < 0.0001. **i**, Cortical cell diameter (40 cells from 5 sections per genotype). Statistics from one of three replicates (two-way ANOVA followed by Tukey test). *P* (interaction) = 0.0001. **j**, Cellulose content in five-day-old wild-type, *arf1-1* and OE-ARF1 roots grown under normal and dense conditions (*n* = 6 biological replicates). Dots indicate individual data points. Statistical analysis conducted using two-way ANOVA followed by Tukey test. *P* (interaction) = 0.001. Different letters indicate significant differences (*P* < 0.05) in **h**–**j**. Box plots in **e**,**h**,**i** show median (centre line), 25th–75th percentiles (box) and minimum to maximum range (whiskers). Bars indicate mean ± s.d. in **b**,**c**,**j**.

that OsARF1 functions as a transcriptional repressor of the CESA gene family (Fig. 3c).

To investigate root growth characteristics of *arf1-1* and OE-ARF1 lines, we grew wild-type, *arf1-1* and OE-ARF1 lines on normal or dense agar media, or in non-compacted or compacted soil for five days. Wild-type seedling root length was inhibited by 35% in compacted soil and by 26% in dense media (Fig. 3f,h and Extended Data Fig. 4b,c). By contrast, *arf1-1* displayed root growth defects in both non-compacted and compacted

conditions (Fig. 3f,h and Extended Data Fig. 4b,c), whereas OE-ARF1 roots were less affected than wild type in compacted conditions as root length was inhibited by 19% in soil but was not inhibited in dense media conditions (Fig. 3f,h and Extended Data Fig. 4d–f). Notably, the *arf1-1* mutant roots were of similar length as wild type when grown in germination pouches (Fig. 3d,e), indicating that OsARF1 may be associated with root growth regulation in response to different compaction conditions.

To reveal the effects of OsARF1 activity on the root radial response to compaction, we analysed root cross-sections from wild-type, *arf1* mutant and OE-ARF1 seedlings. Wild-type roots increased their diameter when subjected to compaction conditions, but *arf1-1* and *arf1-2* roots did not (Fig. 3g,i and Extended Data Fig. 4g,h). Notably, OE-ARF1 seedlings displayed constitutive root swelling in both non-compacted and compacted conditions (Fig. 3g,i), phenocopying *cesa6* mutants and indaziflam-treated roots. We reasoned that if OsARF1 regulates root radial expansion through *OsCESA6*, the expression patterns of *OsARF1* and *OsCESA6* should show opposite trends under compaction. Indeed, using *ProOsCESA6*:GUS reporter lines, we found that *OsCESA6* expression was significantly reduced in the cortex of early elongation zone (Extended Data Fig. 5), precisely where *OsARF1* expression was strongly induced by compaction (Fig. 3a).

To validate the regulatory relationship between OsARF1 and cellulose synthesis, we quantified cellulose content in roots grown under varying media conditions. Whereas dense media led to reduced cellulose content in wild-type roots compared with normal conditions, *arf1-1* maintained similar cellulose levels across both conditions, with significantly higher content than in the wild type under dense media conditions (Fig. 3j). Conversely, OE-ARF1 roots showed consistently lower cellulose content than wild type roots, particularly under normal media conditions (Fig. 3j). Further supporting these results, chemical reduction of cellulose synthesis using either indaziflam (150 pM) or 2,6-dichlorobenzonitrile (DCB)[19] (50 nM; DCB reduces CSC mobility without affecting plasma membrane CSC density) rescued the root growth defects of *arf1-1* mutants (Extended Data Fig. 6). Furthermore, the *osarf1* mutant root phenotypes in both normal and dense media conditions were rescued in *arf1cesa6* double-mutant lines (Extended Data Fig. 7). Collectively, these results demonstrate that OsARF1 orchestrates root responses to compaction through modulation of cellulose biosynthesis by suppressing CESA expression.

## Roots modify cell walls under compaction

Cell wall biosynthesis and structure are fundamental determinants of root elongation and expansion. Although the relationship between wall structures and cell growth has been studied extensively, certain regulatory mechanisms remain unknown. Previous studies have demonstrated that cell wall thickness and stiffness vary across different root regions and cell types[20]. For instance, cells in the elongation zone exhibit lower stiffness compared with those in the meristem zone, facilitating cell elongation[20]. Furthermore, cells with higher stiffness typically possess thicker cell walls[21,22]. On the basis of these observations, we hypothesized that radially expanded cortical cells should exhibit thinner and more flexible walls. To test this, we examined cortical cell wall thickness of wild-type, *arf1-1*, OE-ARF1, *cesa6* and *arf1cesa6* root elongation zone using transmission electron microscopy (TEM). After determining the elongation zone position, we compared wall thickness among genotypes using semi-thin transverse sections (Extended Data Fig. 8). To assure that we captured cell wall thickness variations, we measured the thickness at 15 different points per cell (Extended Data Fig. 9) using 20 cells per genotype, and calculated the average to represent cell wall thickness. We found that the wall thickness of wild-type cortical cells decreased significantly under dense media conditions compared with normal conditions (Fig. 4a–d). By contrast, *arf1-1* maintained thick root cortex walls in both non-compacted and compacted conditions, whereas OE-ARF1, *cesa6* and *arf1cesa6* root cortex consistently exhibited thinner walls regardless of growth conditions (Fig. 4a–d). Atomic force microscopy (AFM) analysis revealed similar patterns to the thickness measurements: the wild-type root cell walls exhibited lower stiffness values than *arf1-1*, whereas OE-ARF1, *cesa6* and *arf1cesa6* root cell walls had the lowest stiffness values in dense media conditions (Fig. 4a,f). As AFM measurements on sectioned material can only be used to compare cell wall stiffness across cells and samples,

they are not indicators of in vivo cell wall stiffness, as the loading conditions, wall geometry and hydration state differ substantially from the living state. Thus, our findings indicate that the cell wall thickness of cortex cells corresponded to the differential root responses during soil compaction: *arf1-1* cortex cells maintained their diameter, correlating with thicker and stiffer cortical cell walls, and the constitutive radial expansion observed in OE-ARF1 and *cesa6* corresponded to their thinner and softer cell walls.

The epidermal cell layer is also crucial for roots to grow and thrive in soil. For instance, thicker walls of outer small cortical cells in wheat and maize seedling roots enhanced root penetration ability[23]. Indeed, we found that all genotypes contained thicker epidermal cell walls when grown on dense media (Fig. 4a–c,e and Extended Data Fig. 8), with little difference across the genotypes. Epidermal wall AFM measurements differed between wild type and *arf1-1*, but not between wild type and OE-ARF1, *cesa6* or *arf1cesa6* (Fig. 4a,g). These observations suggest that thicker epidermis cell walls and thinner cortex cell walls facilitate root adaptation to soil compaction.

## Compaction induces *OsARF1* via ethylene

Our studies revealed that root compaction responses are controlled by OsARF1 repressing CESA expression and cell wall thickness in cortical cells, but it is unclear how OsARF1 is regulated by compaction stress. Soil compaction reduces diffusion of the gaseous hormone ethylene, causing it to accumulate around plant roots and trigger an ethylene response. This in turn upregulates auxin and abscisic acid (ABA) signals, which act to repress root elongation and promote root radial expansion, respectively[5,6]. We therefore investigated whether any of these hormones control *OsARF1* expression. To address this, we grew the *ProOsARF1*:GUS reporter line in control, 10 μM IAA (auxin), 25 μM ABA or 100 μM ACC (1-aminocyclopropane-1-carboxylic acid, an ethylene precursor) conditions. ACC treatment strongly induced *OsARF1* expression, similar to the induction observed in dense agar media, whereas IAA and ABA had no or little effect, respectively (Extended Data Fig. 10a). This finding suggests that ethylene may be the primary signal that links compaction and *OsARF1* expression. We also noted moderate induction of *OsARF1* by ABA (25 μM) in the cortical region, although less pronounced than that caused by the ACC treatment. Subsequent GUS quantification and RT–qPCR analyses confirmed that there was significant upregulation of *OsARF1* expression with ACC treatment, but showed reduced expression with ABA treatment (Extended Data Fig. 10b,c). This reduction in ABA-treated samples might reflect changes in overall root *OsARF1* expression, as RT–qPCR measurements encompassed whole roots, potentially masking localized expression increases observed in our GUS analyses. These results support ethylene as an inducer of *OsARF1* expression. We hypothesize that ABA may function downstream of ethylene, as suggested by previous studies[5], partially modulating *OsARF1*, although this relationship requires further investigation.

Given that *OsEIN2*, *OsEIL1* and *OsEIL2* encode key components of the ethylene response pathway, specifically a response effector and upstream transcription factors[24], respectively, we analysed whether *OsARF1* acts downstream of these genes by growing Nip (*Nipponbare*), *ein2* and *eil1eil2* seedlings under normal and dense media conditions. Under dense media conditions, *OsARF1* expression was significantly induced in both Nip and *eil1eil2* mutants, but not in *ein2* (Fig. 4h). This suggests that *OsARF1* functions downstream of *OsEIN2*. Furthermore, whereas dense media treatment led to downregulation of cellulose synthase genes (*OsCESA5*, *OsCESA6* and *OsCESA8*) in Nip, this response was abolished in *ein2* mutants (Fig. 4h). Of note, in the *eil1eil2* background, the three CESA genes showed distinct responses: *OsCESA5* maintained its downregulation, *OsCESA6* showed no significant change, and *OsCESA8* was upregulated (Fig. 4h). These differential responses highlight the complex regulatory network that connects ethylene signalling

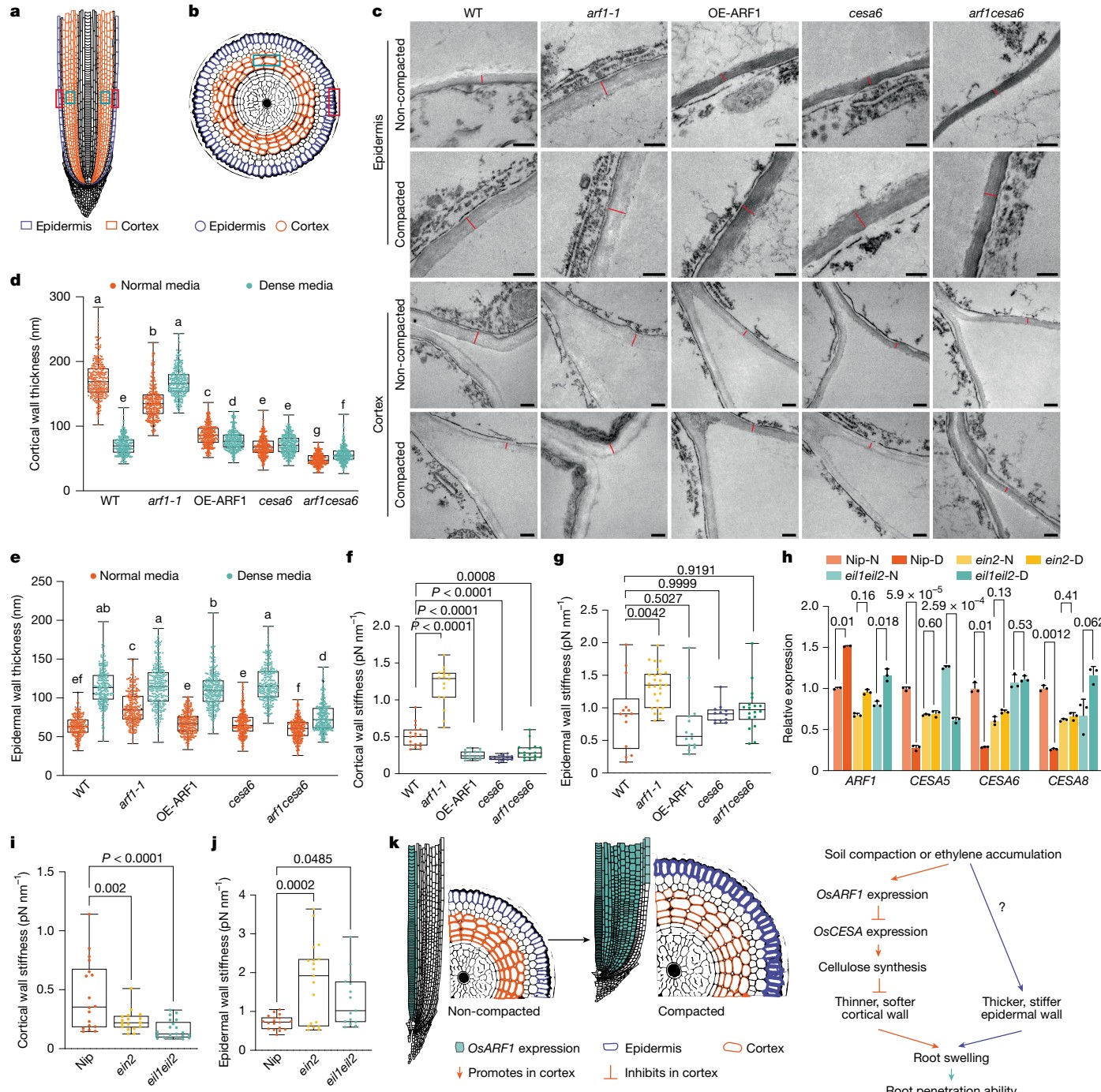

**Fig. 4 | A thicker epidermis and thinner cortex cell wall model facilitates rice root adaption to compaction. a**,**b**, Longitudinal (**a**) and cross-sectional (**b**) schematics showing TEM and AFM measurement locations in cortex (blue) and epidermis (red) of rice root cells. **c**, TEM images showing wall thickness (red lines) in root epidermis and cortex of five-day-old wild-type, *arf1-1*, OE-ARF1, *cesa6* and *arf1cesa6* seedlings grown in normal and dense media. Scale bars, 200 nm. **d**,**e**, Epidermal (**d**) and cortical (**e**) wall thickness statistical analysis (*n* = 300, 15 points from 20 cells, 1 of 5 replicates) was performed using two-way ANOVA followed by Tukey test. *P* (interaction) < 0.0001. **f**,**g**, Root cortical (**f**) and epidermal (**g**) wall stiffness in wild type, *arf1-1*, OE-ARF1, *cesa6* and *arf1cesa6* seedlings in dense media (*n* = 15 test points, 1 of 4 replicates; two-sided *P* values from Dunnett test). **h**, Expression of *ARF1*, *CESA5*, *CESA6* and *CESA8* in the elongation zone of Nip and *ein2* seedlings grown under normal (N) and

dense (D) media for five days (one of three replicates, two-sided *P* values of Welch's *t*-test with Bonferroni–Dunn method are shown). Bars show mean ± s.d. **i**,**j**, Root cortical (**i**) and epidermal (**j**) wall stiffness of Nip, *ein2* and *eil1eil2* plants grown in dense media (*n* = 17 plants, 4 replicates; two-sided *P* values from Dunnett test). Box plots in **d**,**g**,**i**,**j** show median (centre), 25th–75th percentiles (box), and minimum to maximum range (whiskers). Different letters indicate significant differences (*P* < 0.05) in **d**,**e**. **k**, Soil compaction and ethylene induce *OsARF1* expression mainly in cortex cells, suppressing primary wall CESA expression. Reduced cellulose biosynthesis causes radial expansion of cortex cells and root swelling due to decreased cortical wall thickness. By contrast, epidermal wall thickness and stiffness are maintained to support root penetration in compacted soils. Colour intensity denotes cell wall thickness.

and cellulose synthesis. Notably, roots of *ein2* and *eil1* seedlings, despite their slimmer morphology, successfully penetrate compacted soils[6], suggesting that ethylene-insensitive roots have enhanced soil penetration capacity. This observation appears to contrast our finding that OE-ARF1 and *cesa6* mutant roots, which are thicker than wild type, also show improved soil penetration. To further explore these apparent inconsistencies, we conducted growth experiments in which we germinated and grew wild type, *ein2* and *eil1eil2* seedlings in different media conditions with or without 150 pM indaziflam. The resulting root thickening (Extended Data Fig. 10d,e) suggests that cellulose synthesis modifications may occur downstream of OsEIN2, OsEIL1 and OsEIL2 to enhance radial swelling in rice roots, although the relationship between these pathways requires further investigation. Notably, *ein2* and *eil1eil2* mutants displayed epidermal and cortical wall stiffness patterns that were consistent with our model (Fig. 4i,j). We hypothesize that the absence of cell swelling response in *ein2* mutants under compacted conditions may be attributed to ABA accumulation defects[5]. Collectively, these data support that a thicker epidermis–thinner cortex model promotes root penetration capability in compacted soil conditions.

## Conclusion

Our study provides conceptual insights into how roots adapt to soil compaction stress through cortical cell expansion, and provides a plausible thicker epidermis–thinner cortex model of how the root effectively responds to soil compaction (Fig. 4k). Under compaction stress and the resulting ethylene response, *OsARF1* expression is induced primarily in the cortex of the elongation zone (Fig. 3a and Extended Data Fig. 10a–c), which suppresses *OsCESA6* (Extended Data Fig. 5) and other primary wall CESAs, reducing cellulose synthesis in cortex, leading to thinner cell walls and radial expansion of cortex cells. The thickening of epidermal cell walls is likely to involve unknown regulatory pathways that remain unidentified. This coordinated differential regulation between cortex and epidermis promotes root swelling and enhances root penetration ability in compacted soil conditions (Fig. 4k). The radial expansion of cortical cells exerts pressure on the outer epidermal cell layer, potentially providing axial stability to the root. The increased root diameter, combined with maintained epidermal stiffness, may enhance stability against ovalization and reduce buckling susceptibility[25]. The thicker epidermis–thinner cortex model therefore parallels engineering principles used in pipe design, where maintaining perimeter stiffness while increasing diameter improves structural stability. Future biomechanical testing and modelling, such as finite element modelling, will be well suited to explore the plausibility of this mechanical scenario in roots. Although OsARF1-mediated cellulose regulation affects root adaptation to compaction, we observed that epidermal cell wall thickening still occurred in all genotypes under dense media conditions (Fig. 4a–d). This suggests that multiple mechanisms contribute to the thick and stiff epidermal cell wall model. Of note, OsEIL1 promotes epidermal wall thickening through direct regulation of OsCSLC2-mediated xyloglucan biosynthesis[26].

In our study, root expansion resulted primarily from cortical cell expansion rather than changes in cell number, which has been identified as a distinct phene in maize roots[27]. By contrast, our analysis of cortical cell numbers in rice under various soil conditions revealed no consistent differences. The relationship between mechanical stress and cellular organization is closely linked to the arrangement of microtubules that coordinates directions of cellulose microfibrils in the cell walls. Indeed, the weakened cortex cell walls may cause an increased pressure on the epidermal cell layers, which in turn could affect, and perhaps hyper-align, the microtubules in the epidermis. This could in turn cause an increase in transverse microfibrils that would stiffen the epidermis to prevent further radial swelling of the roots and to aid directed growth of the roots. Although challenging in rice root cells,

imaging of microtubules in root adaptation to soil compaction remains an important area for future research[28].

Our research establishes a crucial link between ethylene signalling and root adaptation through the regulation of cell wall synthesis, highlighting the dynamic role of cellulose synthesis in root penetration of soil. ABA is also regulated by ethylene during root compaction, and may function as a downstream signal affecting *OsARF1* levels, as evidenced by its induction of *OsARF1* expression in cortex layers (Extended Data Fig. 10a). In addition, brassinosteroids regulate hypocotyl elongation through epidermal wall loosening rather than wall thickness[29], and brassinosteroid-responsive genes *HOMEOBOX FROM ARABIDOPSIS THALIANA 7* (*HAT7*) and *GT-2-LIKE 1* (*GTL1*) are expressed in cortex to mediate cell elongation by activating cell wall-related genes[30]. However, Zhu et al. did not observe any significant trends in brassinosteroid gene expression under compacted soil conditions[31]. In summary, our findings have significant implications for developing crop varieties better adapted to challenging soil conditions through targeted modification of cell wall properties and hormone responses.

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

# Methods

## Plant material construct development and growth conditions

The background/wild type of *osarf1*, OE-ARF1, *cesa6*, *arf1cesa6* and NG-CESA6 in this study was rice (*Oryza sativa ssp. japonica*) variety 9522. *Nipponbare* (Nip), *ein2* and *eil1eil2* were sourced from the group of R. Huang. The *osarf1*, *cesa6* and *arf1cesa6* knock out alleles were generated using CRISPR–Cas9 technology, with mutant genotyping conducted through PCR amplification of the target regions followed by sequencing of the products. We amplified *OsARF1* coding (2,100 bp) and promoter (2,915 bp in front of ATG) sequences and then cloned them into pTCK303 and pCAMBIA1301 to produce OE-ARF1 and *ProOsARF1*:GUS, respectively. We also amplified *OsCESA6* promoter (2,877 bp upstream of ATG) sequence and then cloned the segment into pCAMBIA1301 to produce *ProOsCESA6*:GUS. mNeon-Green was fused to the *OsCESA6* genomic sequence under the control of the *OsCESA6* promoter and was cloned into pCAMBIA1301 to create NG-CESA6, as described in reference[15]. All plasmids were constructed using the In-Fusion HD Cloning Kit (Takara) and were verified by sequencing at BGI, Beijing, China. These plasmids were introduced into the calli of 9522 or *cesa6* via Agrobacterium (EHA105)-mediated transformation, with hygromycin B used for selection as outlined in reference[32]. All primers used are listed in Supplementary Table 1.

All rice samples were grown and collected at the paddy field of Shanghai Jiao Tong University, under natural conditions from June to October. Seedlings were cultivated in a plant incubator at 28 °C with a 16 h:8 h light:dark cycle and 50%–70% humidity.

## Soil compaction details

Loamy sand from the Newport series (comprising 83.2% sand, 12.1% clay and 4.7% silt; with 2.93% organic matter and a pH of 6.35; classified as FAO Brown Soil) and clay soil were collected from the University of Nottingham farm at Bunny, Nottinghamshire, UK (52.52° N, 1.07° W). Both soils were passed through a 2-mm sieve. The moisture content was determined by drying the soil at 45 °C until a constant weight was achieved.

Mesocosms were prepared by packing them with the sieved soil to achieve a bulk density of 1.2 g cm$^{-3}$, to simulate a non-compacted condition, and 1.6 g cm$^{-3}$ for a compacted condition. Rice seeds were sterilized with a 25% bleach solution for 10 min and subsequently rinsed six times with sterilized water. The seeds were then placed on filter paper for 48 h in a dark incubator chamber to allow for germination. One germinated seedling per mesocosm was positioned on the soil surface and covered with a 1 cm top layer of soil at 1.2 g cm$^{-3}$, in both non-compacted and compacted soil conditions. The seedlings were then placed in a rice growth chamber, which was maintained at 28 °C, with a 12 h photoperiod and 70% relative humidity.

## X-ray CT imaging

Five-day-old seedlings of wild type, *osarf1* and *cesa6* were grown in 3D-printed columns (33 mm in diameter and 100 nm in height) filled with sandy loam soil under non-compacted (1.2 g cm$^{-3}$) and compacted (1.6 g cm$^{-3}$) conditions, respectively. They were then imaged using a GE Phoenix v|tome|x M 240 kV X-ray tomography system at The Hounsfield Facility, University of Nottingham. Three-dimensional image reconstruction was carried out using Datos|REC software (GE Inspection Technologies). The roots were segmented from the soil using a polyline tool in VGStudioMAX V2.2 (Volume Graphics) to demonstrate the root length phenotype. The scanning protocol involved collecting 3,240 projection images in FAST mode (continuous rotation), with the X-ray tube set to an energy of 140 kV and a current of 200 µA. The detector's exposure time was 131 ms, and the voxel resolution was 57 µm. Each scan had a duration of 7 min.

## Agar experimental details

Agar at concentrations of normal (0.3%) and dense (0.6%) (Sigma Aldrich 7002) was boiled and poured into tanks measuring 7.5 cm in diameter and 10 cm in length to simulate non-compacted and compacted soil conditions, respectively. After the agar solidified, germinated seeds were placed on the media surface, followed by watering (1 cm height). The seeds were then grown in a plant incubator (28 °C, 16 h light:8 h dark, 70% relative humidity) for 5 days. Finally, root phenotypes were imaged using a Nikon camera, and the images were analysed with ImageJ software.

## Imaging of root tip thickness

Root tips, including the root cap, meristematic, elongation and differentiation zones (approximately 1 cm of the rice root tip), were washed 3 times with sterilized water. They were then embedded in 5% melted agarose (LabTop Biotechnology). Transverse sections, 50 µm thick, were cut using a Leica Vibratome (VT 1000 S) and imaged with a Leica SP5 confocal microscope utilizing the UV laser.

## RNA isolation and RT–qPCR

Root tips, including the root cap, meristematic zone, elongation zone, and differentiation zones (approximately 1.5 cm of the rice root tip), grown in soil or agar media conditions, were sampled in three biological replicates. Total RNA was extracted using Trizol reagent (Invitrogen) followed by purification with chloroform/isopentyl alcohol. The cDNA was synthesized using the FastQuant RT Kit with gDNase (Tiangen Biotech). RT–qPCR was conducted using the SYBR Green Premix Pro Taq HS qPCR Kit (Accurate Biotechnology). The reaction mixture consisted of 7.5 µl SYBR, 2 µl cDNA, 0.3 µl each of forward and reverse primers, and 4.9 µl double-distilled water. The Ubiquitin gene served as the reference for assessing gene expression levels. All RT–qPCR primers are listed in Supplementary Table 1, and gene information is listed in Supplementary Table 2.

## Y1H screening

Nine promoter fragments—753 bp for *OsCESA1*, 1,973 bp for *OsCESA3*, 1,928 bp for *OsCESA5*, 1,964 bp for *OsCESA6*, 1,751 bp for *OsCESA8*, 1,943 bp for *OsCSI1*, 1,872 bp for *OsCSLF6*, 2,628 bp for *OsCTL*, and 923 bp for *OsSTL*—as well as a 1,738 bp fragment for the *OsHARPIN1* promoter were cloned into the R4L1pDEST_HIS2 plasmid[33]. The cDNA library encompasses 1,143 rice transcription factors, representing all known transcription factor families. The appropriate 3AT concentration was determined through autoactivation testing. Subsequently, plasmids were transformed into YM4271 yeast strain (Clontech/TAKARA) using TE/LiAc and PEG solution. Yeast plates were incubated at 30 °C for 3–7 days, after which images were captured using a camera.

## Dual-luciferase assay

Rice seedlings were grown for 7 days post-germination in a plant incubator set to 28 °C with a 16 h:8 h light:dark cycle and 50%–70% humidity. Seedling shoots were collected and digested in an enzyme solution containing 0.2 g Cellulase 'Onozuka' RS (Yakult), 0.05 g Macerozyme R10 (Yakult) and β-mercaptoethanol to a final concentration of 10 mM. After degassing twice, the solution was incubated at 28 °C and 80 rpm for 2 h. Protoplasts were isolated using W5 solution (150 mM NaCl, 125 mM CaCl$_2$, 5 mM KCl, 2 mM MES, pH 5.7) and MMg solution (400 mM mannitol, 15 mM MgCl$_2$·6H$_2$O, 4 mM MES, pH 5.7), examined under a microscope, and adjusted to the appropriate concentration (2.0–2.5 × 10$^5$ cells per ml). The *OsARF1* and *eGFP* coding sequences, serving as effectors, were cloned into the pZmUBQ1p_SX_HSPG plasmid. Similarly, promoter sequences of cellulose synthesis genes, acting as reporters, were cloned into the pGL4.1HSP plasmid. Both effectors and reporters were introduced into the protoplasts using PEG/Ca$^{2+}$ and incubated for 16–18 h at 28 °C with 80 rpm agitation. The relative reporter activity was determined by normalizing the reporter luciferase value

to the Renilla luciferase reference value. The incubated protoplasts were lysed using the Dual-LUC reporter assay kit (TOYOINK) and the luminescence was measured using a multi-scan spectrometer (TECAN).

## EMSA

First, we amplified the full-length coding sequence of *OsARF1* and cloned it into the pSP6-T7 vector (Promega). The OsARF1 protein was then expressed using an in vitro translation kit (TNT T7/SP6 Coupled Wheat Germ Extract System; Promega) and analysed by western blot (anti-His: M20003, Abmart, 1:2,000 dilution; Goat Anti-Mouse IgG HRP: M21001, Abmart, 1:5,000 dilution). Double-stranded M1, M2 and M3 hot probes were labelled with FAM dye at the 5′ end. For competition assays, we used 200-fold excess of non-labelled probes (wt) and labelled mutated probes (mut). The protein-probe binding reaction mixture, which included 250 mM Tris-Acetate, 10 mM DTT, 1 mg ml$^{-1}$ BSA, and 20 mM magnesium acetate, was incubated at 25 °C for 20 min. The reaction products were then resolved on a 6% native polyacrylamide gel and visualized using the Cy2 channel of a ChemiDoc MP imaging system (Bio-Rad).

## ChIP–qPCR

The ChIP assay was conducted as described[34], with slight modifications. Approximately 2 g of root tips (about 1.5 cm in length) from wild-type and *ProOsARF1*:gOsARF1–GFP plants (verified by anti-GFP: G1544, Sigma, 1:5,000 dilution; and Goat Anti-Rabbit IgG Antibody: AP132, Sigma, 1:5,000 dilution) were crosslinked with 1% (v/v) formaldehyde in extraction buffer (0.4 M sucrose, 10 mM Tris-HCl, 5 mM β-mercaptoethanol, 0.1 mM PMSF, and a protease inhibitor cocktail, pH 8.0), then pulverized in liquid nitrogen. Chromatin was subsequently isolated and sonicated to generate DNA fragments ranging in size from 200 to 500 bp. GFP-Trap Magnetic Agarose (ChromoTek) was utilized to precipitate OsARF1–DNA complexes. Fragments of the *OsCESA5* and *OsCESA6* promoters were quantified by RT–qPCR using primers listed in Supplemental Table 1. Enrichment in the *ProOsARF1*:gOsARF1–GFP samples was compared with the levels in wild-type plants.

## Hormone and indaziflam treatment conditions

Rice seeds of *ProOsARF1*:GUS were sterilized with 70% (v/v) ethanol for 1 min and 3% (v/v) sodium hypochlorite solution for 15 min, followed by at least 6 rinses with sterile double-distilled water, for 3 min each. The seeds were then placed on filter paper in germination boxes for 3 days under a 16 h:8 h light:dark cycle. After germination, the seeds were exposed to various conditions: mock (water), 1 μM IAA, 25 μM ABA and 100 μM ACC for 2 days.

Similarly, germinated seeds of wild-type, *osarf1* and OE-ARF1 plants were transferred to black boxes containing growth medium supplemented with either 150 pM (final concentration) DMSO or 150 pM indaziflam, and 250 pM DMSO or 250 pM indaziflam, and 50 nM DMSO or 50 nM DCB. These were incubated in a plant incubator at 28 °C, with a 16 h:8 h light:dark cycle and 50%–70% humidity for 5 days. The primary root lengths of the mock-treated and experimental groups were measured and compared using ImageJ. This experiment was performed in triplicate.

## GUS histochemical and quantification

Seedlings of the rice auxin response factor transcriptional reporter (*ProOsARF1*:GUS and *ProOsCESA6*:GUS) were grown in conditions including mock (water with added DMSO), 0.6% agar media, 1 μM IAA, 25 μM ABA and 100 μM ACC. Root tips were then collected and incubated overnight in GUS buffer (50 mM NaPO$_4$ buffer, pH 7.0, 10 mg ml$^{-1}$ X-Gluc, and 0.02% (v/v) Triton X-100 at 37 °C. Subsequently, the samples were washed with 70% (v/v) ethanol until they became transparent. Cross-sections were prepared using a Leica vibratome (VT 1000 S) at a thickness of 50 μm. Images were taken with a Leica light microscope (M205A) equipped with a CCD camera, and additional photographs were obtained using a Nikon H600L light microscope (Tokyo).

For GUS activity quantification, approximately 0.1 g of *ProOsARF1*: GUS roots (1.5 cm) were collected after 2-day treatment with mock (DMSO), 1 μM IAA, 25 μM ABA, or 100 μM ACC. Root tissues were ground in liquid nitrogen and homogenized in 1 ml GUS extraction buffer (1 M Na$_2$HPO$_4$, 1 M NaH$_2$PO$_4$, 10% SDS, 0.5 M EDTA, Triton X-100, β-mercaptoethanol). The homogenate was centrifuged at 12,000 rpm for 5 min at 4 °C, and protein concentration in the supernatant was determined using the Bradford method. For the GUS activity assay, 100 μl of protein extract was mixed with 400 μl pre-warmed (37 °C) GUS extraction buffer and 500 μl MUG substrate (2 mM). The reaction mixture was incubated at 37 °C, and 200 μl aliquots were collected at 0, 15, 30, 45 and 60 min intervals. Each aliquot was immediately mixed with 800 μl stop solution (0.2 M Na$_2$CO$_3$) and stored at room temperature in the dark. Fluorescence intensity was measured using a fluorescence microplate reader (excitation: 365 nm, emission: 455 nm, slit width: 10 nm). The rate of fluorescence intensity change was calculated from the slope of the fluorescence intensity versus time plot and normalized to total protein content to determine specific GUS activity.

## Cellulose measurements

Cellulose content was quantified as described[35], with modifications. Five-day-old seedlings of wild-type, *osarf1* and OE-ARF1 were grown in 0.3% or 0.6% agar media (Sigma-Aldrich-7002) at 28 °C under a 16 h:8 h light:dark cycle. Root tips (approximately 1.5 cm) were collected and processed to obtain alcohol-insoluble residue (AIR). In brief, samples were incubated in 1.5 ml 70% ethanol at 70 °C for 1 h, followed by a second 45 min incubation with fresh ethanol. After ethanol removal, samples were treated with 1 ml acetone at room temperature for 2 min. The resulting AIR was dried, weighed, and transferred to 15 ml Falcon tubes. For cellulose extraction, 3 ml of acetic/nitric reagent was added to each sample (including a blank control), and tubes were heated in a boiling water bath for 30 min. After cooling, samples were centrifuged at 2,000*g* for 10 min. The supernatant was carefully discarded, and pellets were washed with 8 ml water, gently resuspended and incubated for 15 min before centrifugation (2,000*g*, 10 min). A final wash was performed with 4 ml acetone, followed by a 5 min incubation and centrifugation (2,000*g*, 10 min). Samples were dried in a vacuum oven at 30–40 °C. The dried samples were hydrolysed in 1 ml of 67% sulfuric acid with shaking (180 rpm) at room temperature until complete dissolution. For colorimetric determination, 40 μl of each hydrolysate was mixed with 1 ml ice-cold anthrone reagent and heated in boiling water for 5 min, followed by immediate ice quenching. Cellulose content was calculated as percentage of cell wall (CW) per AIR using glucose standard curves according to the formula: CW (%) = (glucose (μg))/ (AIR (μg)) × 100% × (H$_2$SO$_4$ volume)/(sample volume).

## Cell wall thickness experimental details

Roots from 5-day-old seedlings of wild-type, *osarf1*, OE-ARF1 and *cesa6* plants were cultivated in 0.3% and 0.6% agar media (Sigma-Aldrich-7002) at 28 °C under a 16 h:8 h light:dark cycle. Root tips, approximately 1 cm in length, were collected and fixed in a solution of 3% (w/v) formaldehyde (formaldehyde freshly prepared from paraformaldehyde) and 0.25% glutaraldehyde in 0.2 N sodium phosphate buffer (pH 7.0). These root tips were then post-fixed in 2% OsO$_4$ in PBS (pH 7.2). The samples were dehydrated through a graded ethanol series (70% for 30 min, 90% for 30 min, and 100% 3 times, each for 30 min) and then transitioned through ethanol/epoxypropane mixtures (2:1, 1:1 and 1:2), followed by pure epoxypropane, each for 10 min. Samples were embedded in acrylic resin (London Resin Company) and placed in a drying oven at 37 °C for 5–12 h to eliminate bubbles, then the temperature was increased to 45 °C for 2 h, and finally to 65 °C for 48 h. We carefully selected comparable elongation zone regions among all genotypes by 2-μm semi-thin sections (Extended Data Fig. 8a), and then made ultra-thin sections (50 to 70 nm; Extended Data Fig. 8b), which were double-stained with 2% (w/v) uranyl acetate and 2.6% (w/v) lead citrate aqueous solution and

examined with a JEM-1230 transmission electron microscope (JEOL) at 80 kV. Wall thickness was estimated via ImageJ software. To account for cell wall thickness heterogeneity, we measured a single wild-type cortical cell divided into 8 segments and found that cell wall thickness variations can effectively be averaged by measuring wall thickness at 15 different points (Extended Data Fig. 9). Therefore, we selected 15 points around each cell of a total of 20 cells per genotype. These experiments were repeated five times.

## Cell wall stiffness experimental details

Roots from 5-day-old seedlings of wild-type, *osarf1*, OE-ARF1, *cesa6* and *arf1cesa6* plants were grown in 0.6% agar media (Sigma-Aldrich-7002) at 28 °C with a 16 h:8 h light:dark cycle. Primary root tips, measured to 1.5 cm, were embedded in 5% melted agarose (LabTop Biotechnology) for sectioning. Longitudinal sections of 50 μm thickness were produced using a Leica vibratome (frequency 50 Hz, amplitude 1 mm, VT 1000 S), and light microscopy was employed to ensure the stele and cortical tissues were correctly positioned with the elongation zone visible. The sections were then stored in deionized water at 4 °C overnight.

AFM assays were conducted following the methods described[36]. A Dimension ICON (Bruker Nano) equipped with NanoScope Analysis 9.4 software was utilized to probe all root samples. The MLCT-C probe (Bruker Nano), with an average spring constant of 0.01 N m$^{-1}$, indentation depth (varying from samples, 200–300 nm), speed (3,000 nm s$^{-1}$), and force limit (300 pN) was used in the experiments. Root sections set in agarose were affixed to glass slides with tape and hydrated with deionized water for 30 min prior to AFM analysis. While operating in force-spectroscopy mode under water-hydrated conditions, 10–15 independent areas within the visible cortex and epidermis zones (as shown in Fig. 4a) were examined. Four biological replicates were conducted for each plant line. Apparent stiffness (in pN nm$^{-1}$) values were extracted from individual force-distance curves using a contact point based fit and a linear stiffness model, using NanoScope Analysis 1.8 software.

## Reporting summary

Further information on research design is available in the Nature Portfolio Reporting Summary linked to this article.

## Data availability

All information supporting the conclusions are provided with the paper. Primers used in this study are provided in Supplementary Table 1. Gene accession number information is available in Supplementary Table 2. Y1H screening results are provided in Supplementary Data 1, and raw data for EMSA (gel) is provided in Supplementary Data 2. Source data are provided with this paper.

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

**Acknowledgements** We thank R. Huang for providing the rice *ein2* and *eil1eil2* lines; A. Geitmann for useful comments on finite element modelling (FE) simulations, which finally were not included in the manuscript; S. Kijima, K. M. Chung and F. Tobe for their technical contributions to the Y1H screens; X. Chen for generating rice transgenic lines; and G. Wang, X. Zhang, H. Li, M. Hua, L. Zhu, Z. Wang and J. Zheng for their assistance with TEM experiments. This study was supported by the National Natural Science Foundation of China (32130006), the Project of Hainan Research Institute, Shanghai Jiao Tong University (grant no: HRSJ-ZSZX-006) to W.L.; S.P. acknowledges Villum Investigator (project ID: 25915), DNRF Chair (DNRF155), Novo Nordisk Laureate (NNF19OC0056076), Novo Nordisk Emerging Investigator (NNF20OC0060564), Novo Nordisk Data Science (NNF0068884) and Lundbeck foundation (Experiment grant, R346-2020-1546) grants; B.K.P. acknowledges BBSRC Discovery Fellowship no. BB/ V00557X/1; Royal Society Research Grant, RGS\R1\231374 and UKRI Frontiers Research (ERC StG, EP/Y036697/1); M.J.B. acknowledges BBSRC grant, BB/W008874/1 and ERC SYNERGY (101118769—HYDROSENSING) grant. O.C. was funded by a Biotechnology and Biological Sciences Research Council grant (grant number BB/X014908/1); J.Z. acknowledges China Postdoctoral Science Foundation Funded Project (23Z020705425) and Young Scientists Fund of the National Natural Science Foundation of China (24Z033004245).

**Author contributions** J.Z., M.J.B., W.L., B.K.P., D.Z. and S.P. conceptualized the study. J.Z., E.J.F., B.K.P. and S.P. were responsible for the methodology. J.Z. performed root penetration phenotype, gene expression test, cellulose content measurements, EMSA, ChIP–qPCR and cortical cell diameter measurements. Z.L. offered *cesa6* mutants, cytoskeletons and CESA marker obeservations. J.Z. and Z.L. performed wall thickness quantification. E.J.F. and B.K.P. performed CT scans. M.L., H.L. and O.C. performed FE model simulations. O.C., L.P.O. and A.B. performed ForSys analysis. J.Z., F.X. and Y.Y. performed TEM. J.Z., J.L., Z.Q. and Q.S. performed AFM. J.Z. and S.S. performed LUC. J.Z. and N.M. performed Y1H and statistical methods. X.Z. performed plasmids for Y1H. M.Z. performed single-cell analysis associated with CESA and brassinosteroid-related genes. J.S. performed Y1H results figure. J.Z., W.L., B.K.P., M.J.B. and S.P. were responsible for funding acquisition. M.J.B., W.L., B.K.P., D.Z. and S.P. administered the project and supervised the study. J.Z., D.Z. and S.P. wrote the original draft. J.Z., N.M., O.C., M.J.B., W.L., B.K.P., D.Z. and S.P. reviewed and edited the manuscript.

**Competing interests** The authors declare no competing interests.

**Additional information**
**Correspondence and requests for materials** should be addressed to Wanqi Liang, Bipin K. Pandey or Staffan Persson.

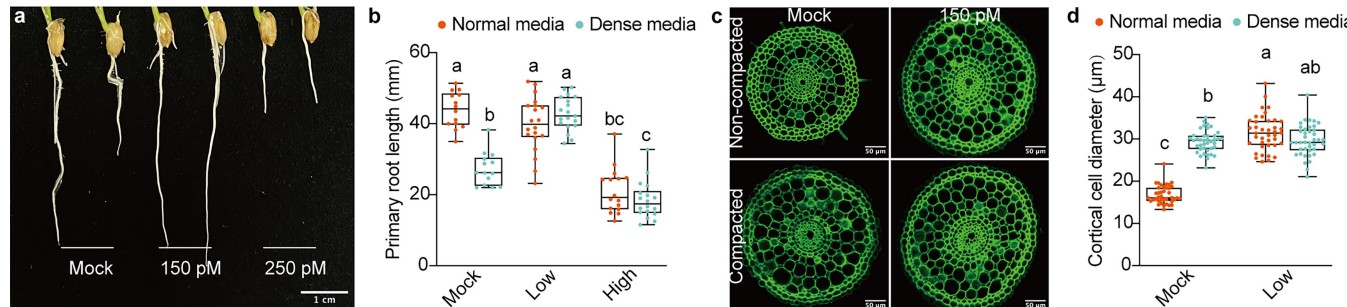

**Extended Data Fig. 1 | Low concentrations of Indaziflam enhance rice root penetration. a-b**, Primary root length of 7-day-old WT seedlings grown on media supplemented with 150 pM or 250 pM Indaziflam under normal (left panel) and dense (right panel) conditions, with DMSO as the mock treatment. The experiment was conducted in triplicate, yielding consistent trends (n = 15 plants per trial). Statistical analysis was performed using two-way ANOVA followed by Tukey test. Different letters denote significant differences at 0.05 level. The $p$ (interaction) <0.0001. Data shown are from one of three representative trials. Scale bar in (**a**): 1 cm. **c**, Cross-sections of 5-day-old WT roots grown on media supplemented with DMSO (mock) or 150 pM Indaziflam, under normal or dense conditions. Sections (50 μm thickness) were prepared using a vibratome and visualized under UV light (405 nm) with an SP5 confocal microscope. Scale bar: 50 μm. **d**, Cortical cell diameter measurements from sections similar to those in (**c**). Data represent 40 cells from 5 sections per condition. The experiment was repeated three times with consistent results. Data shown from one replicate (two-way ANOVA followed by Tukey test. Box plots show 50th percentile median (centre), 25th-75th percentiles (box), and minimum to maximum range (whiskers) in (**b**) and (**d**). Different letters denote significant differences at 0.05 level. The $p$ (interaction) <0.0001).

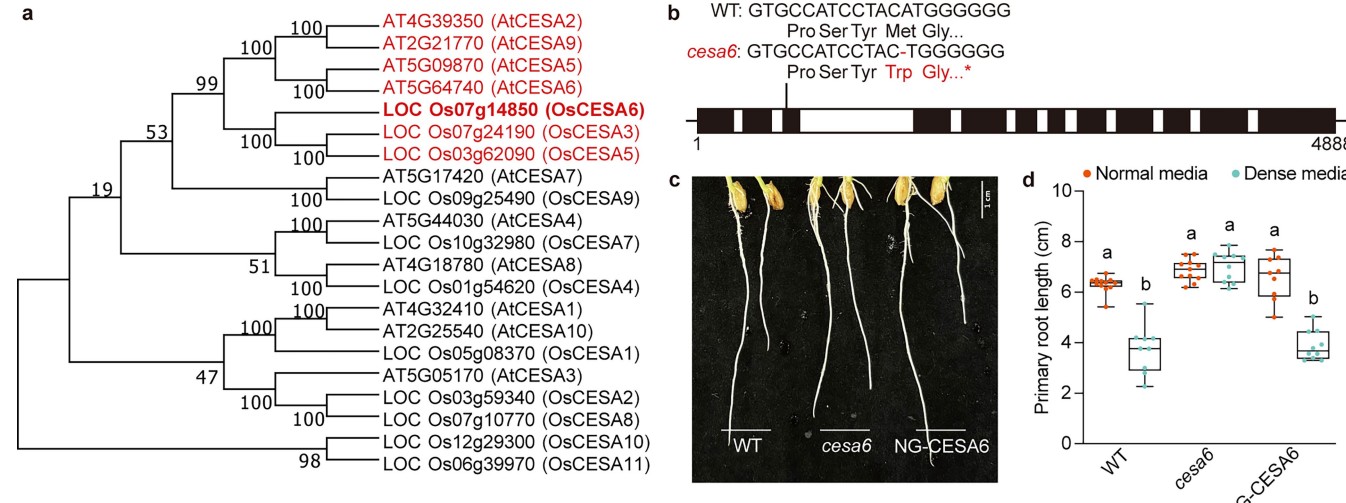

**Extended Data Fig. 2 | Root penetration phenotype of *cesa6* mutants in agar media. a**, A phylogenetic tree of the 10 CESA proteins from *Arabidopsis* and the 11 CESA proteins from rice. AtCESA6 is functionally redundant to CESA2, 5 and 9[37] implying a similar relationship of OsCESA6 and OsCESA3 and 5. Analysis was performed using MEGA 7.0. Bootstrap values indicate the homology of amino acid sequences. **b**, Genotypic characterization of CRISPR-Cas9 *cesa6* mutant lines revealed a deletion of an 'A' in the third exon, resulting in a frameshift mutation and premature termination. Black and white boxes indicate exons and introns, respectively. **c-d**, Primary root penetration phenotypes of WT,

*cesa6* mutants, and NG-CESA6 (complementation line of *cesa6*) seedlings grown on normal (left panel) and dense (right panel) agar media for 5 days. The experiment was conducted in triplicate, yielding consistent trends (n = 10 plants per trial). Statistical analysis was performed using two-way ANOVA followed by Tukey test. Different letters denote significant differences at 0.05 level. The *p* (interaction) <0.0001. Based on one of three replicates. Box plots show 50th percentile median (centre), 25th-75th percentiles (box), and minimum to maximum range (whiskers) in (**d**).

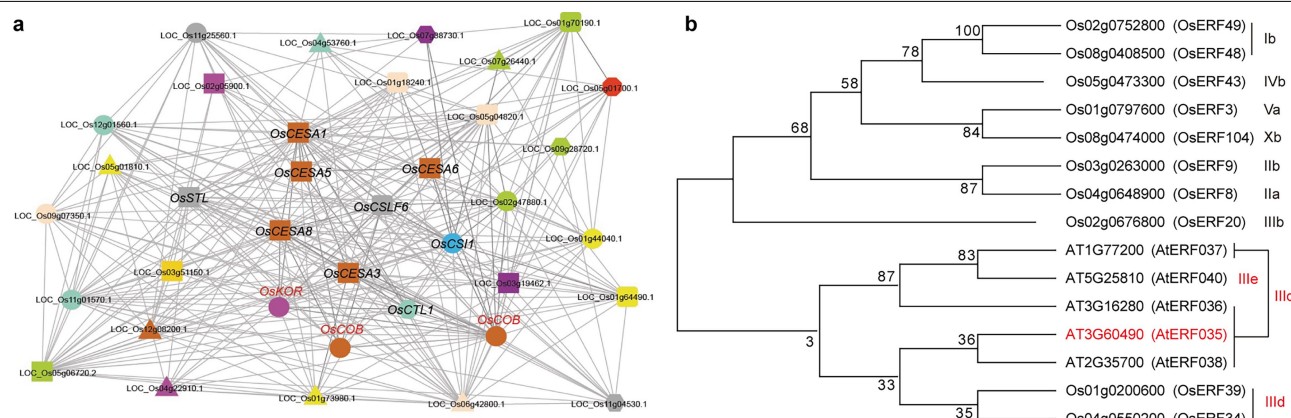

**Extended Data Fig. 3 | Co-expression network of rice *CESA* genes and ERF transcription factor phylogeny. a**, Co-expression network centered on primary wall *CESA* genes (represented by orange squares) in rice. Nodes of various colours and shapes correspond to genes from different gene families. Edges indicate significant co-expression relationships between *CESAs* and the genes represented by the nodes. Network was generated, truncated reformatted from co-expression tool PlaNeT (Mutwil et al.[38]) using OsCESA8 as bait gene. **b**, A phylogenetic tree of selected subclades of rice and *Arabidopsis ERFs*. *OsERF34*, which binds to the promoters of seven cellulose biosynthesis genes, and *OsERF39*, are both close homologs of *AtERF35*, a major primary cell wall regulator in Arabidopsis.

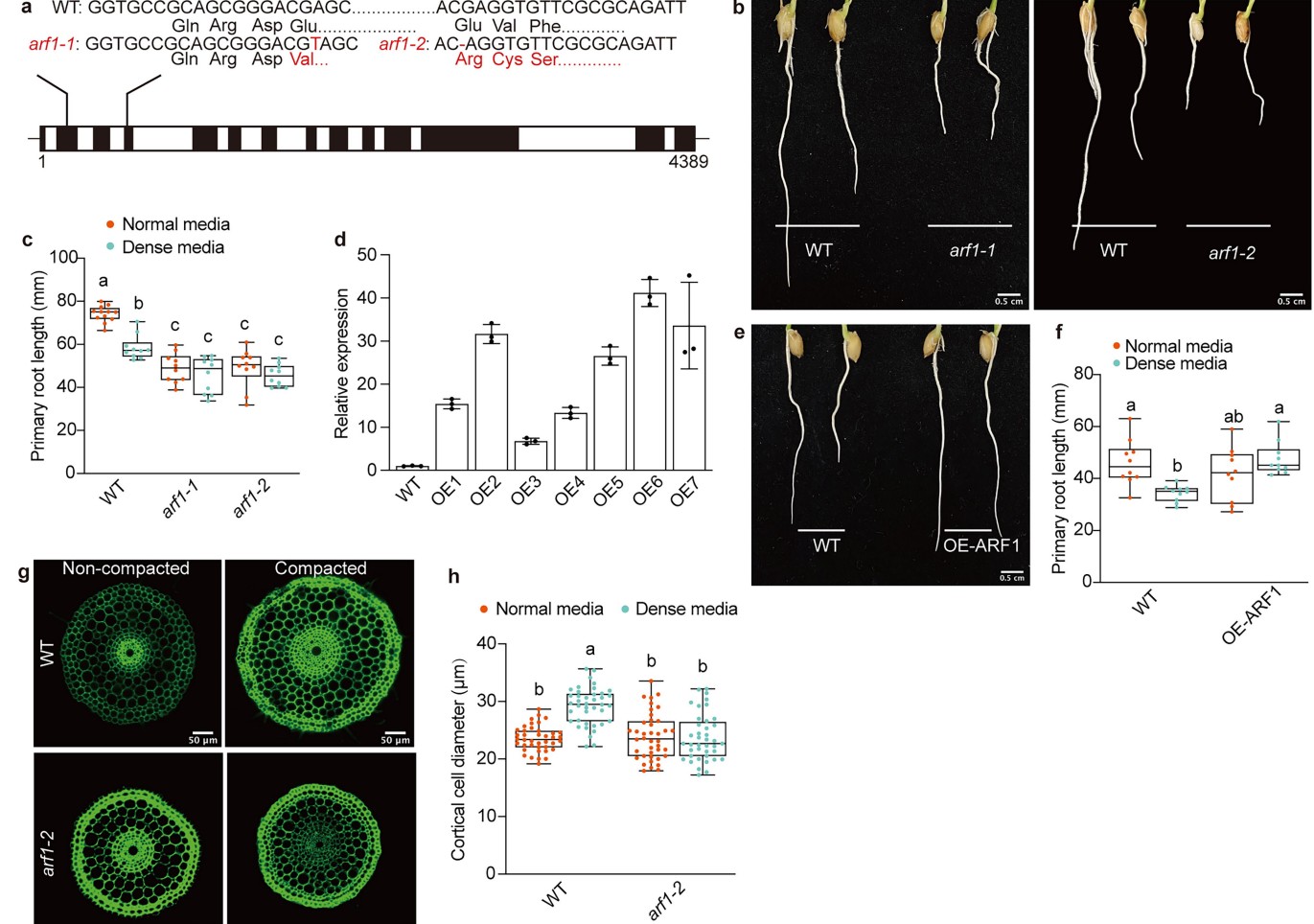

**Extended Data Fig. 4 | OsARF1 promotes rice root elongation and swelling in dense media conditions. a**, Schematic representation of two *arf1* mutant lines: *arf1-1* and *arf1-2*. The *arf1-1* line features a 'T' insertion in the second exon, while *arf1-2* has a 'G' deletion in the fourth exon. Both mutations induce frameshifts leading to premature termination. Black boxes: exons. White boxes: introns. **b-c**, Root penetration phenotypes of WT, *arf1-1*, and *arf1-2* seedlings grown on normal and dense media for 5 days. Statistical analysis was performed on one of three replicates (n = 10 per trial) using two-way ANOVA followed by Tukey test. Different letters denote significant differences at 0.05 level. The *p* (interaction) = 0.004. **d**, Expression level of *OsARF1* in seven overexpression lines was quantified using RT-qPCR (n = 3 technical duplication per OE plant). Seedling roots were sampled for analysis. The maize ubiquitin gene used to drive the expression of *OsARF1* for assessing gene expression levels. Bars indicate mean ± s.d. **e-f**, Root penetration phenotype and length of WT and OE-ARF1 seedlings grown on normal and dense media conditions for 5 days. Statistical analysis was performed on one of three replicates (n = 10 plants) using two-way ANOVA followed by Tukey test. Different letters denote significant differences at 0.05 level. The *p* (interaction) = 0.0008. **g**, Cross-sections of 5-day-old WT and *arf1-2* (elongation zone) grown under normal and dense media conditions. Sections (50 μm thickness) were visualized under UV light (405 nm laser) with an SP5 confocal microscope. Scale bar: 50 μm. **h**, Cortical cell diameter measurements from sections similar to those in (**g**). Data represent 40 cells from 5 sections per condition. Statistical analysis was performed on one of three replicates using two-way ANOVA followed by Tukey test. Different letters denote significant differences at 0.05 level. The *p* (interaction) <0.0001. Box plots show 50th percentile median (center), 25th-75th percentiles (box), and minimum to maximum range (whiskers) in (**c**), (**f**) and (**h**).

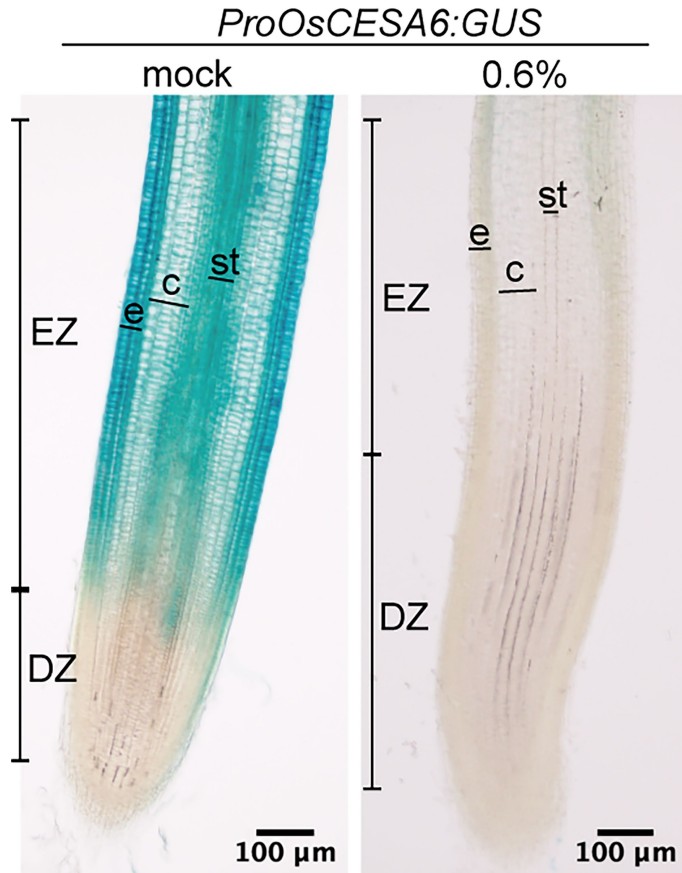

**Extended Data Fig. 5 | Dense media reduced *OsCESA6* level in the elongation zone of rice roots.** *ProOsCESA6*:GUS lines were cultured in mock (water; left panel) and dense (0.6%) media (right panel) for 48 h, followed by GUS staining and stereoscopic imaging. Longitudinal sections (50 μm thickness) were prepared using a vibratome. This experiment was repeated using 18 roots from 6 independent GUS lines. Scale bars: 100 μm. EZ: elongation zone, DZ: division zone, e: epidermis, c: cortex, st: stele.

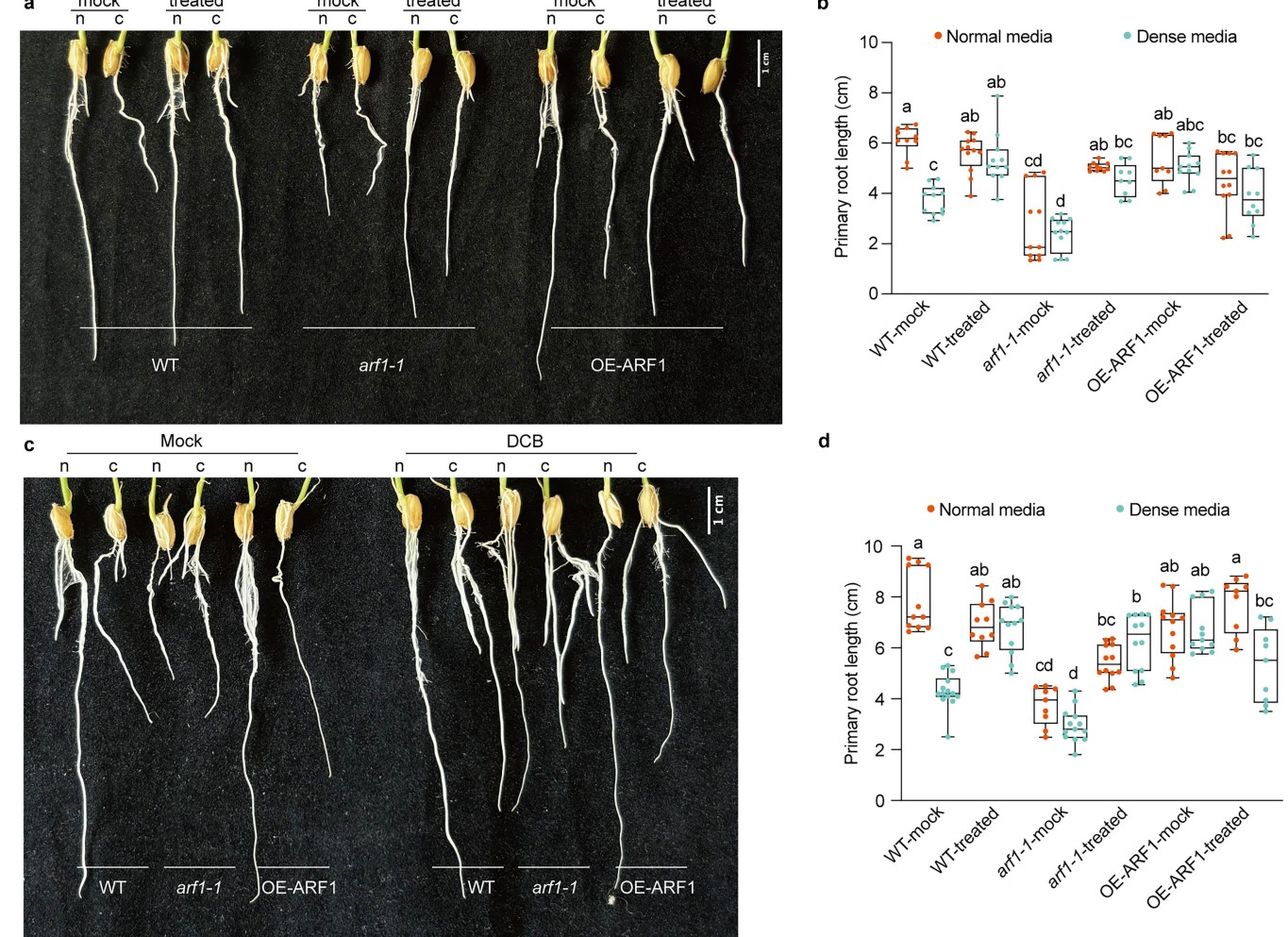

**Extended Data Fig. 6 | Indaziflam and DCB treatment mitigates the root phenotype of *arf1-1* mutants. a**, Comparison of primary root lengths in WT, *arf1-1*, and OE-ARF1 seedlings grown for 5 days on normal (n) and dense (c) media. Each group shows roots grown with mock (DMSO, left two roots) or (150 pM Indaziflam, right two roots). Scale bar: 1 cm. **b**, Quantification of root lengths from (**a**). The experiment was repeated thrice, consistently yielding similar trends (n = 10 plants per trial). Statistical analysis was performed using two-way ANOVA followed by Tukey test. Different letters denote significant differences at 0.05 level. The *p* (interaction) = 0.001. **c**, Comparison of primary root lengths in WT, *arf1-1*, and OE-ARF1 seedlings grown for 5 days on normal (n) and dense (c) media. Each group shows roots grown with mock (DMSO) and 50 nM DCB treatments. Scale bar: 1 cm. **d**, Quantification of root lengths from (**c**). The experiment was repeated thrice, consistently yielding similar trends (n = 10 plants per trial). Statistical analysis was performed using two-way ANOVA followed by Tukey test. Different letters denote significant differences at 0.05 level. The *p* (interaction) < 0.0001. Box plots show 50th percentile median (centre), 25th-75th percentiles (box), and minimum to maximum range (whiskers) in (**b**) and (**d**).

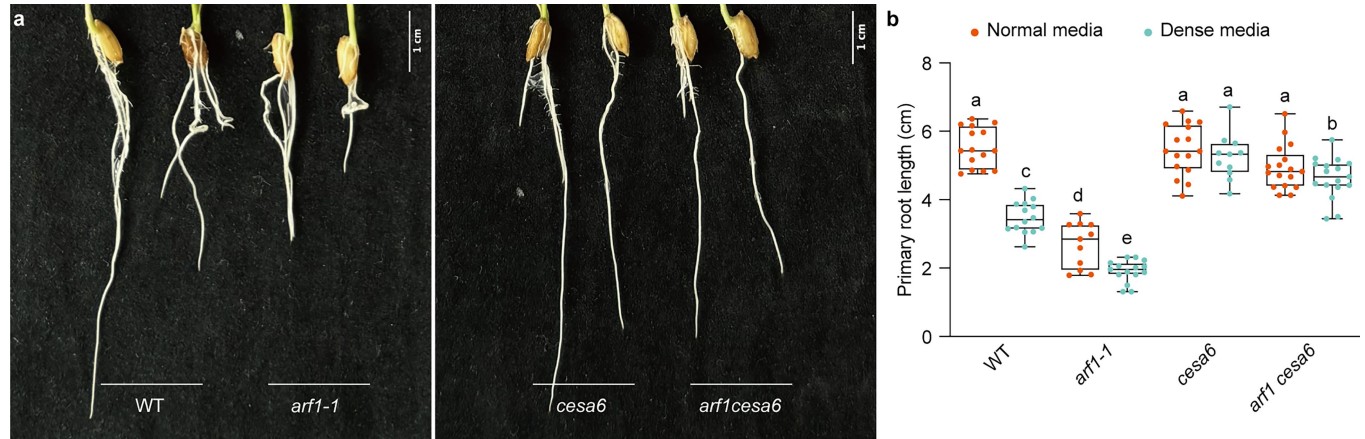

**Extended Data Fig. 7 | The *arf1* root phenotype is rescued in *arf1cesa6* double mutant line. a**, Root penetration phenotype and length of WT, *arf1-1*, *cesa6* and *arf1cesa6* seedlings grown on normal and dense media conditions for 5 days. The experiment was conducted in triplicate, yielding consistent trends. Scale bar=1 cm. **b**, Statistical analysis (n = 15 plants per trial) of (**a**) was performed on one replicate using two-way ANOVA followed by Tukey test. Different letters denote significant differences at 0.05 level. The *p* (interaction)<0.0001. Box plots show 50th percentile median (center), 25th-75th percentiles (box), and minimum to maximum range (whiskers).

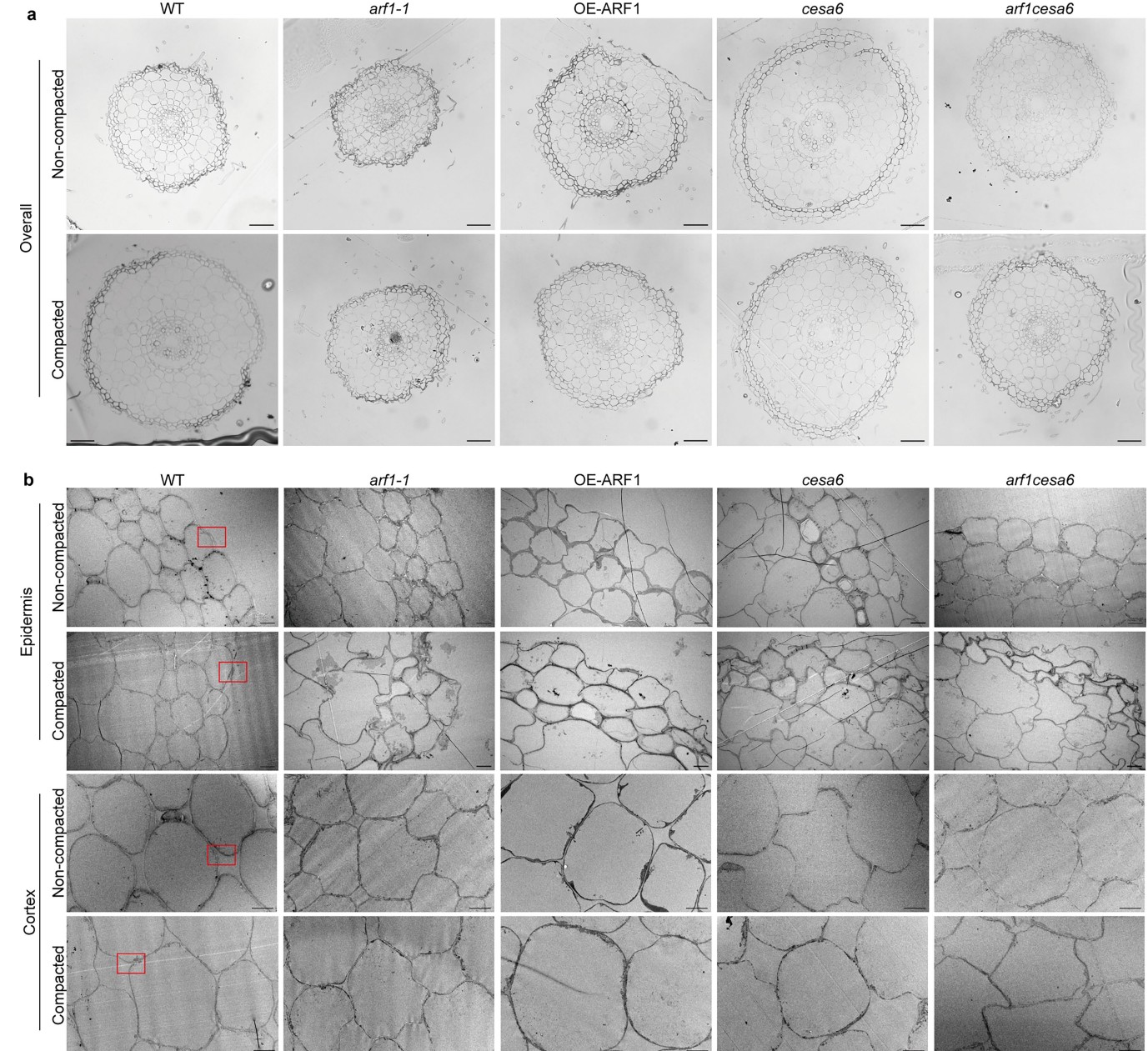

**Extended Data Fig. 8 | Locations for measuring the thickness of the epidermal and cortical cell walls. a**, Semi-thin sections (2 μm thickness, 1 cm from root tip) of WT, *arf1-1*, OE-ARF1, *cesa6* and *arf1cesa6* lines. Scale bar=50 μm. **b**, Magnified views of root epidermis and cortex sections from WT, *arf1-1*, OE-ARF1, *cesa6*, and *arf1cesa6* lines. Measurement of cell wall thickness were done on walls facing intercellular spaces to ensure capturing a single walls (indicated by red boxes). The experiment was repeated five times. Scale bars: 5 μm for epidermis and 1 μm for cortex regions.

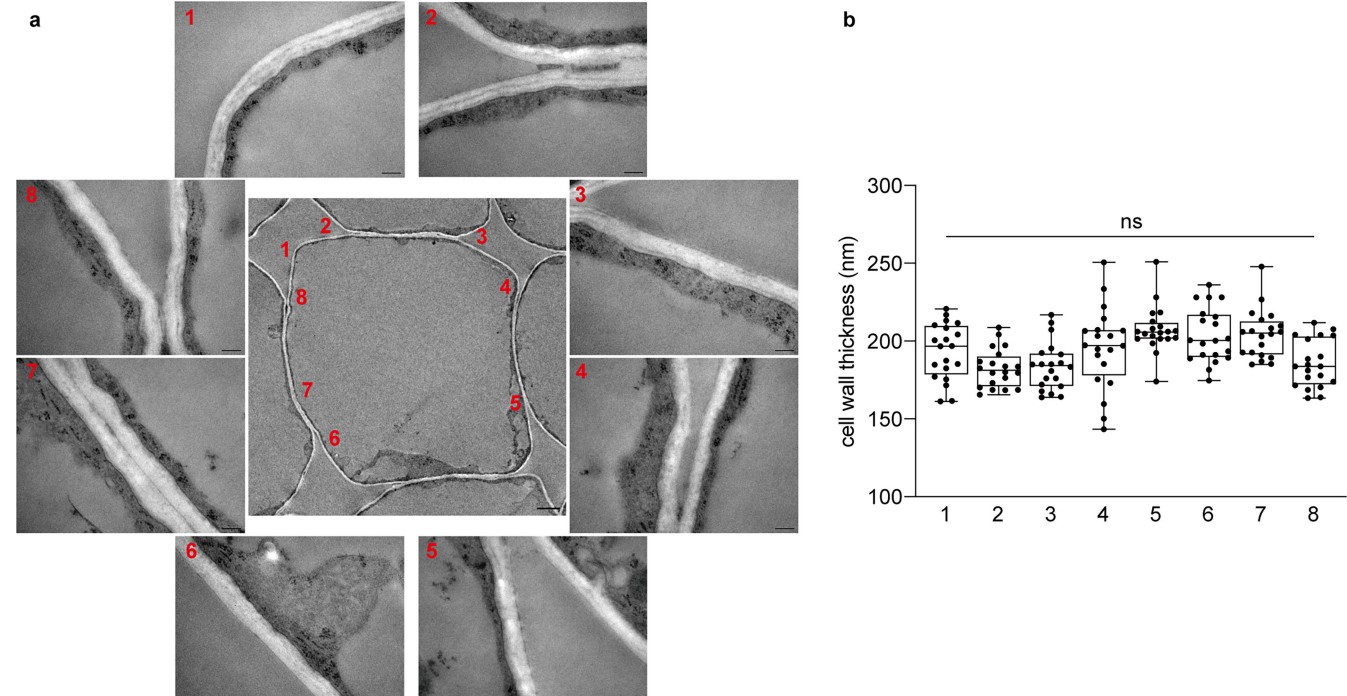

**Extended Data Fig. 9 | Cell wall thickness variation analyses of single root cortex cell.** Cell wall thickness and statistical analysis of individual WT cells, divided into 8 segments to evaluate cell wall thickness variations. Fifteen points were measured per segment to determine cell wall thickness. Scale bar, 200 nm in segmental micrographs, 2 μm in the central micrograph. Statistical analysis was performed using one-way ANOVA and no significant difference of variance was detected. Box plots show 50th percentile median (centre), 25th-75th percentiles (box), and minimum to maximum range (whiskers).

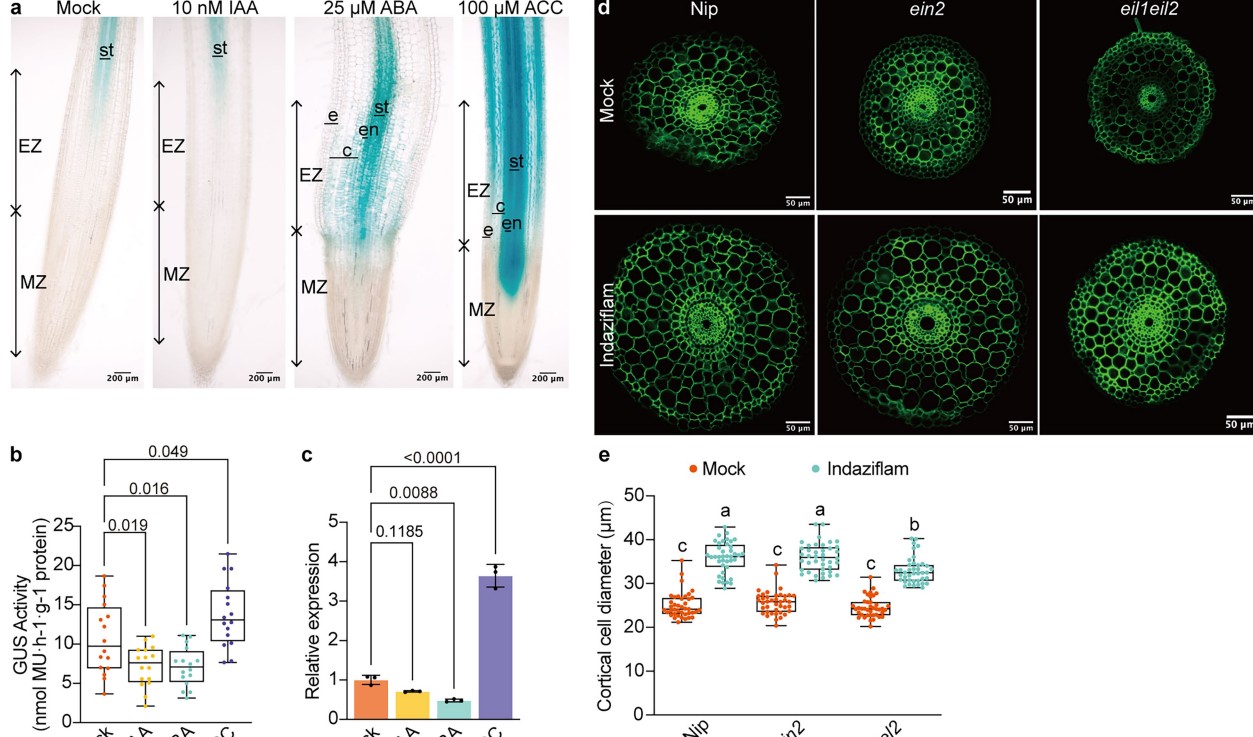

**Extended Data Fig. 10 | OsARF1 functions downstream of ethylene in soil compaction responses of roots. a**, *ProOsARF1*:GUS seedling roots were grown for 48 h on media supplemented with mock (DMSO), IAA (10 μM), ABA (25 μM), or ACC (100 μM), then stained for GUS activity and imaged using a stereoscope (n=18). Longitudinal sections (50 μm thickness) were prepared using a vibratome. Scale bar: 200 μm. EZ: elongation zone, MZ: division meristem zone, e: epidermis, c: cortex, en: endodermis, st: stele. **b**, GUS quantification (change in MU protein amount per unit time) for samples in (**a**). Each treatment had 16 replicates. Statistical analysis was performed using Dunnett test (two-sided *P* values are shown.). **c**, *OsARF1* expression levels in the elongation zone of WT seedling roots after 48-hour treatment with mock (DMSO), IAA (10 μM), ABA (25 μM), or ACC (100 μM). Samples were collected from six seedling root tips per group,

with the experiment repeated three times. Statistical analysis was performed using Dunnett test (two-sided *P* values [range] is shown.). Bars indicate mean ± s.d. **d-e**, Root sections from the elongation zones of 5-day-old WT, *ein2*, and *eil1eil2* seedlings grown on media supplemented with DMSO (mock) or Indaziflam. Measurements were taken from 40 cells across 5 sections per genotype. The experiment was conducted in triplicate with consistent trends. Statistical analysis was performed from one replicate using two-way ANOVA followed by Tukey test. Different letters denote significant differences at 0.05 level. The *p* (interaction) = 0.0211. Box plots show 50th percentile median (centre), 25th-75th percentiles (box), and minimum to maximum range (whiskers) in (**b**) and (**d**).

# Reporting Summary

## Statistics

For all statistical analyses, confirm that the following items are present in the figure legend, table legend, main text, or Methods section.

| n/a | Confirmed | |
|---|---|---|
| ☐ | ☒ | The exact sample size (*n*) for each experimental group/condition, given as a discrete number and unit of measurement |
| ☐ | ☒ | A statement on whether measurements were taken from distinct samples or whether the same sample was measured repeatedly |
| ☐ | ☒ | The statistical test(s) used AND whether they are one- or two-sided *Only common tests should be described solely by name; describe more complex techniques in the Methods section.* |
| ☒ | ☐ | A description of all covariates tested |
| ☐ | ☒ | A description of any assumptions or corrections, such as tests of normality and adjustment for multiple comparisons |
| ☐ | ☒ | A full description of the statistical parameters including central tendency (e.g. means) or other basic estimates (e.g. regression coefficient) AND variation (e.g. standard deviation) or associated estimates of uncertainty (e.g. confidence intervals) |
| ☐ | ☒ | For null hypothesis testing, the test statistic (e.g. *F*, *t*, *r*) with confidence intervals, effect sizes, degrees of freedom and *P* value noted *Give P values as exact values whenever suitable.* |
| ☒ | ☐ | For Bayesian analysis, information on the choice of priors and Markov chain Monte Carlo settings |
| ☒ | ☐ | For hierarchical and complex designs, identification of the appropriate level for tests and full reporting of outcomes |
| ☒ | ☐ | Estimates of effect sizes (e.g. Cohen's *d*, Pearson's *r*), indicating how they were calculated |

*Our web collection on statistics for biologists contains articles on many of the points above.*

## Software and code

Policy information about availability of computer code

| Data collection | CT scan was imaged using a GE Phoenix v\|tome\|x M 240 kV X-ray tomography system; Root sections were taken by a Leica Vibratome (VT 1000 S), and imaged with a Leica TCS SP5 confocal microscope; GUS images were taken by a a Nikon H600L light microscope (Tokyo, Japan); TEM images were taken by a JEM-1230 transmission electron microscope (JEOL) at 80 kV; AFM results were taken by a Dimension ICON (Bruker Nano). |
|---|---|
| Data analysis | The co-expression network of CESAs analyzed via https://conekt.sbs.ntu.edu.sg/; The CT three-dimensional image reconstruction was carried out using Datos\|REC software (GE Inspection Technologies, Wunstorf, Germany); Statistical analyses of all graphs were performed using GraphPad Prism (Version 9.2.0) (https://www.graphpad.com/); Phylogenetic analysis data of CESA homologs was analyzed using the MEGA software (Version 5.0); Microscopy images were analyzed by ImageJ2 (Version 2.14.0/1.54f); AFM results were analyzed by the NanoScope Analysis 9.4 software. |

For manuscripts utilizing custom algorithms or software that are central to the research but not yet described in published literature, software must be made available to editors and reviewers. We strongly encourage code deposition in a community repository (e.g. GitHub). See the Nature Portfolio guidelines for submitting code & software for further information.

## Data

Policy information about availability of data

All manuscripts must include a data availability statement. This statement should provide the following information, where applicable:
- Accession codes, unique identifiers, or web links for publicly available datasets
- A description of any restrictions on data availability
- For clinical datasets or third party data, please ensure that the statement adheres to our policy

All information supporting the conclusions are provided with the paper. Primers used in this study is provided in Supplementary Table 1. Gene accession number information is available in Supplementary Table 2. Yeast one-hybrid screening results is provided in Supplementary Data 1, and raw data for EMSA (gel) is provided in Supplementary Data 2. Source Data for Main Figures and Extended Data Figures are also provided with this paper.

## Research involving human participants, their data, or biological material

Policy information about studies with human participants or human data. See also policy information about sex, gender (identity/presentation), and sexual orientation and race, ethnicity and racism.

| | |
|---|---|
| Reporting on sex and gender | N/A |
| Reporting on race, ethnicity, or other socially relevant groupings | N/A |
| Population characteristics | N/A |
| Recruitment | N/A |
| Ethics oversight | N/A |

Note that full information on the approval of the study protocol must also be provided in the manuscript.

# Field-specific reporting

Please select the one below that is the best fit for your research. If you are not sure, read the appropriate sections before making your selection.

☒ Life sciences ☐ Behavioural & social sciences ☐ Ecological, evolutionary & environmental sciences

For a reference copy of the document with all sections, see nature.com/documents/nr-reporting-summary-flat.pdf

# Life sciences study design

All studies must disclose on these points even when the disclosure is negative.

| | |
|---|---|
| Sample size | Sample sizes were determined based on standard practices in plant biology research and experimental objectives. For example, for root cortical diameter measurements, approx. 40 cells from 5 sections per genotype were imaged per biological replicate, with three biological replicates performed. For AFM analysis, 15 measurement points were tested per genotype across four biological replicates. For TEM imaging, two sections were examined per biological replicate, with five biological replicates performed per genotype. These are typical sample sizes and repetitions compared to similar experimental analyses. |
| Data exclusions | We did not exclude any data from the results. |
| Replication | Each experiment was repeated independently from twice to five replicates, with all replicates showing consistent trends. |
| Randomization | All the experiments were performed without prior knowledge of the final outcome, and therefore randomization was not applied. |
| Blinding | All the experiments were performed without prior knowledge of the final outcome, and therefore blinding was not applied on most of our analyses. However, we did perform blind experiments on the cell wall thickness analyses as these are sampling error prone. |

# Reporting for specific materials, systems and methods

We require information from authors about some types of materials, experimental systems and methods used in many studies. Here, indicate whether each material, system or method listed is relevant to your study. If you are not sure if a list item applies to your research, read the appropriate section before selecting a response.

## Materials & experimental systems

| n/a | Involved in the study |
|---|---|
| ☐ | ☒ Antibodies |
| ☒ | ☐ Eukaryotic cell lines |
| ☒ | ☐ Palaeontology and archaeology |
| ☒ | ☐ Animals and other organisms |
| ☒ | ☐ Clinical data |
| ☒ | ☐ Dual use research of concern |
| ☐ | ☒ Plants |

## Methods

| n/a | Involved in the study |
|---|---|
| ☒ | ☐ ChIP-seq |
| ☒ | ☐ Flow cytometry |
| ☒ | ☐ MRI-based neuroimaging |

## Antibodies

| | |
|---|---|
| Antibodies used | Commercial antibodies used: α-GFP (G1544, Sigma, 1:5000 dilution), α-His (M20003, Abmart, 1:2000 dilution), Goat Anti-Mouse IgG HRP (M21001, Abmart, 1:5000 dilution), Goat Anti-Rabbit IgG Antibody (AP132, sigma, 1:5000 dilution). |
| Validation | Validation statements relevant citations of commercial antibodies available from manufactures<br>α-GFP (G1544, Sigma): https://www.sigmaaldrich.com/US/en/product/sigma/g1544<br>anti-His (M20001, Abmart): http://www.ab-mart.com.cn/page.aspx?node=%2059%20&id=%20959<br>Goat Anti-Mouse IgG HRP (M21001, Abmart): https://www.ab-mart.com.cn/page.aspx?node=%2062%20&id=%20960<br>Goat Anti-Rabbit IgG Antibody (AP132, Sigma): https://www.sigmaaldrich.com/US/en/product/mm/ap132 |

## Dual use research of concern

Policy information about dual use research of concern

### Hazards

Could the accidental, deliberate or reckless misuse of agents or technologies generated in the work, or the application of information presented in the manuscript, pose a threat to:

| No | Yes | |
|---|---|---|
| ☒ | ☐ | Public health |
| ☒ | ☐ | National security |
| ☒ | ☐ | Crops and/or livestock |
| ☒ | ☐ | Ecosystems |
| ☒ | ☐ | Any other significant area |

### Experiments of concern

Does the work involve any of these experiments of concern:

| No | Yes | |
|---|---|---|
| ☒ | ☐ | Demonstrate how to render a vaccine ineffective |
| ☒ | ☐ | Confer resistance to therapeutically useful antibiotics or antiviral agents |
| ☒ | ☐ | Enhance the virulence of a pathogen or render a nonpathogen virulent |
| ☒ | ☐ | Increase transmissibility of a pathogen |
| ☒ | ☐ | Alter the host range of a pathogen |
| ☒ | ☐ | Enable evasion of diagnostic/detection modalities |
| ☒ | ☐ | Enable the weaponization of a biological agent or toxin |
| ☒ | ☐ | Any other potentially harmful combination of experiments and agents |

## Plants

| | |
|---|---|
| Seed stocks | arf1, cesa6, OE-ARF1, NG-CESA6, arf1cesa6 were obtained in lab. Nip, ein2, eil1eil2 were obtained from Rongfeng Huang's lab. All seeds were grown and collected at the paddy field of Shanghai Jiao Tong University, under natural conditions from June to October. |
| Novel plant genotypes | arf1, cesa6 and arf1cesa6 were obtained by CRISPR/Cas9, OE-ARF1, NG-CESA6 were transformed into wild type (9522) or cesa6 mutant background by the staff in lab. |
| Authentication | The genotype of each mutant was obtained from DNA extracted from a single plant, amplified by PCR, and verified by sequencing. The mutant are stable plant lines in which Cas9 was out-crossed. The RNA extracted from each over-expressed plant was verified by qPCR. These operations are outlined in the method. |

