## [Peer Review File · Nature]

Ethylene modulates cell wall mechanics for root responses to compaction

Corresponding Author: Dr Staffan Persson

Version 0:

Reviewer comments:

Referee #1

(Remarks to the Author)

In the manuscript "Ethylene dynamically controls cell wall mechanics to support root responses to compaction" by Jiao Zhang and colleagues, the authors investigate how ethylene leads to radial expansion of root cells and their cell wall properties by a cascade that involves ethylene, Auxin Response Factor 1, and Cellulose Synthase genes. Based on their findings they postulate a "stronger-epidermis-weaker cortex" model that links many of their observations with functional root growth outcomes in compacted soils. Overall, the authors report very interesting findings that link regulatory responses to root physiology and might have important conceptual and practical relevance.

The manuscript is written well and in a concise manner. The work is technically very impressive, elegant, and conceptually interesting while grounded in state-of-the-art experimental data. However, there are several potentially profound issues highlighted below that need to be considered.

Major

1. The *arf1-1* mutant does display much slower growth in non-compacted and compacted soils (Fig 2E). There is a major concern that the general growth phenotype of the *arf1-1* mutant might confound results and conclusions.
 - o The *arf1-1* mutant has a general growth phenotype; how can the authors disentangle this from the response to compacted soil/agar? Could the observed phenotypes be a stress response due to the general growth defects?
 - o The authors claim that the application of low concentrations of Indaziflam ameliorated the growth defect of *arf1-1* mutant roots on media (Extended Data Fig. 7b,c), but there are no mock treatment controls. How can they exclude that they exposed WT to a stress by this treatment, and it thus displays the same growth defects as observed in *arf1-1*? If it is like this, it might not support their conclusions.
 - o According to the authors, TEM imaging revealed that the wall thickness of WT cortex cells was reduced in compacted conditions compared to non-compacted conditions (Fig. 4a,b; Extended Data Fig. 1c). How did they make sure they imaged comparable sections of the root in WT and *arf1-1*?
 - o In case of multi-genotype, multi-condition comparisons, 2-way ANOVA will be required to draw firm conclusions. For instance, the comparisons of WT, *arf1-1*, and OE-ARF1 lines on normal and dense agar media, or in non-compacted and compacted soils will require this. Otherwise, the results can be misleading.
2. The authors pose the model that ARF1 that regulates cellulose synthases is regulated by ethylene to achieve root penetration via leading to a stronger-epidermis and a weaker cortex layer. However, the data is not as clear-cut as it might seem and some of the results contradict the "stronger-epidermis-weaker cortex" model or at least might diminish its relevance.
 - o Is ARF1 mainly regulated by ethylene? Line 193ff: "In contrast to IAA and ABA, ACC treatment resulted in strong induction of OsARF1 expression as observed in dense agar media, (Fig. 5a). This result suggests that OsARF1 is mainly regulated by ethylene, rather than via auxin or ABA." However, from the figure it seems obvious that all treatments elicited changes. In particular, the ABA treatment seems to lead to ARF1 upregulation in a region (elongation-differentiation zone) that might be of relevance for ARF1 function. It is also not clear whether the cross-sections and the longitudinal GUS pictures agree. A quantitative approach for GUS quantification might be helpful. This is of particular importance, as ABA has been described (also by the accompanying manuscript) to have a role in root penetration.

o Ethylene signaling is disrupted in *ein2* and *eil1* mutants. However, responses of the *ein2* and *eil1* mutants seem sometimes to be similar and sometimes not. E.g. Line 211: "Additionally, we discovered that cortex wall stiffness remained unchanged in *ein2* compared to WT, whereas *eil1* cortex cell walls were stiffer (Extended Data Fig. 10c), suggesting that they are not more penetrative than WT in our view. " (also note that the authors should elude more on what their view means) and then Line 214: "In addition, both *ein2* and *eil1* displayed reduced epidermal wall stiffness compared to WT (Extended Data Fig. 10d)" and then Line 207ff: "To further explore these apparent inconsistencies, we conducted growth experiments where we germinated and grew WT, *ein2* and *eil1* on different media conditions. Our results indicate that while *ein2* exhibited similar penetration characteristics to WT, *eil1* performed worse (Extended Data Fig. 10a,b)." So how does this fit all together? Where are the differences between mutants that supposedly are insensitive to ethylene come from? How does that fit with previous studies?

o *ein2* had better penetrability but reduced epidermal wall stiffness. How does that fit with the "stronger-epidermis-weaker cortex" model?

o Line 215: "Interestingly, when we treated *ein2* and *eil1* with 150 pM Indaziflam, the roots became thicker (Fig. 5d,e), corroborating that cellulose synthesis changes occur downstream of OsEIN2 and OsEIL1 to increase radial swelling of rice roots." How can it be excluded that it is not independent? What does it mean with regards to differences of *ein2* and *eil1*? Is corroborated the right word?

o Line 199: "The results showed that OsARF1 induction by ACC was suppressed in *ein2*, and to a lesser extent *eil1* (Fig. 5b)." According to the presented data, the effect of *eil1* was not significant. This should be clarified and put into context.

3. Complementation Fig S2: Why the variance of the AtCESA6 complementation that high. Are the phenotypes of the complemented line significantly different from those of the mutant?

4. Statistics: There are several figures in which it was not indicated whether differences were significant (and at which significance level or not), e.g. Fig S1a.

5. The ChIP analysis doesn't seem to be very rigorous. The precipitate ProARF1:ARF1-GFP was compared to the WT precipitate (negative control). However, it would be good practice to normalize to one or several other genomic regions that are expected not to be enriched – or (better) to conduct a ChIPseq experiment.

6. It is unclear how the scRNA data (Extended Data Fig. 4) that is part of the companion manuscript is necessary; if it is used it should be explored a bit more (e.g. is OsARF1 expressed to the same extent in all tissues upon soil compaction? Is OsARF1 co-expressed with other genes of interest in this study, for example its direct targets?).

7. Line 201 ff: "Consistently, OsCESA3, 5, 6, 8 were down-regulated after ACC treatment in WT; however, not in *arf1-1*, supporting that OsCESA3, 5, 6, 8 expression needs OsARF1 to be regulated by ACC (Fig. 5c)." This is not correct. CESAA3 was not significantly downregulated. Also, it seems that CESA5 and 6 are upregulated in the mutant upon ACC, but it is labelled as NS while the error bars and the effect size seem to be comparable to some other significant labelled data in this figure. The authors should carefully check.

Minor

1. Line 180: "Taken together, these observations suggest that a "thicker epidermis-thinner cortex" cell wall model may drive the ability of plant roots to adapt to soil compaction." How can the authors know that this drives this and is not just linked to processes driving it?

2. L95 "From these assays, we found several ethylene response factors (ERFs), including OsERF34 and 39 (binding to seven and five promoters, respectively; Fig. 2a). " Are ERFs statistically enriched?

3. In line 132 the authors correctly conclude without overstating "Hence, OsARF1 activity appears to control root penetration ability, particularly during soil compaction stress conditions." However, the title of the section "OsARF1 is upregulated by compaction stress and represses CESA gene expression" seems slightly overstated

4. The titles of Extended Data Fig. 9 and Fig 4 seem to be slightly misleading

5. Line 199: "three h" to "three hours"

6. Line 199: "The results showed that OsARF1 induction by ACC was suppressed in *ein2*" is suppressed the right word here?

7. Indaziflam is used in a number of experiments. It would be good to comment on the specificity of the inhibitor.

8. Introduction: "While the mechanical strength of cell walls is crucial for anisotropic cell expansion, little is known about how cell wall properties regulate root penetration of soil, in particular, compaction stress conditions." This sounds like an overstatement. See for instance Schneider et al. 2021 and referenced work therein.

Referee #2

(Remarks to the Author)

This manuscript identifies some new pieces in the puzzle of how roots respond to compacted soils by cortical cell swelling, building on previous work indicating this root growth response is mediated in part by ethylene. Zhang et al. report that ethylene upregulates the transcriptional repressor ARF1 which reduces cellulose synthase (CESA) gene expression in roots of rice seedlings grown in soils of high bulk density. The manuscript shows that cell walls of cortical cells are thinner in roots grown in compacted soil and display greater resistance to lateral indentation by a 20-nm AFM probe. In contrast, epidermal/outer cortical cells do not change in these ways. These changes are attributed to repression of cellulose synthesis, which may be selective for cortical cells, but I do not think this aspect is actually demonstrated. Could this specificity be demonstrated by in-situ hybridization assessments of CESA transcript levels?

The evidence for ARF1 regulation of CESAs includes (a) promoter analysis and (b) phenotypes of *arf1* and ARF1 overexpression lines, the *cesa6* mutant, and comparison with plants treated with the CESA inhibitor Indaziflam. The evidence for this conclusion seems sound and convincing.

The headline for Fig 1 is that CESA6 negatively regulates root penetration of compacted soil. However, I found the reasoning and data confusing. The data in figure 1c indicate that root length is relatively less impacted by 'compacted' soil in the *cesa6* mutant than in WT (root length is decreased by 13% versus 34%). This *cesa6* phenotype is curious and seems to be the foundation of the current study as told here. I note that statistical significance tests are not shown for this figure (likewise for Fig 3f and some others). Moreover it seems that the relatively diminished effect of soil compaction in the *cesa6* mutant arises mostly from the inhibition of root elongation by the *cesa6* mutation: its growth is already stunted in non-compacted soil (Fig. 1c, but where is the statistical test?), so there is less effect of compacted soil. These results seem at odds with the photograph of the *cesa6* mutant in Fig 1a where the root appears a bit longer in non-compacted soil than WT. Not shorter, as Fig 1c indicates. Am I misinterpreting the results? Also, the Y-axis does not start at zero in 1c and 1d, which contributes to the difficulty for interpreting the graphs.

Cortical cell diameters are substantially increased in roots grown in high-density soil (Fig. 1d). This follows the trend of increased root width (Fig. 1b), but the greater root width is apparently not solely due to wider cortical cells but additional changes in root anatomy may contribute: judging from the photographs in Fig 1b, there are changes in the number of cell layers in the cortex, the sub-epidermal layers, and in the stele. These aspects are not discussed in this study.

I infer from the data in the manuscript that the anisotropy of root cell growth is greatly changed upon growth in dense soil, although growth anisotropy is not measured or discussed. A change in growth anisotropy is a classic effect of ethylene and is traditionally considered to be a consequence of alteration in the orientation of cortical microtubules, leading to changes in the orientation of cellulose microfibril deposition. This potential mechanism is not discussed in the manuscript. This seems to be a notable gap, as I suspect it contributes to the root response to compacted soil. Have the authors characterized changes in cortical microtubule and cellulose microfibril orientations in the compacted roots?

CESA mutations and cellulose biosynthesis inhibitors are known to induce wall integrity signals that activate multiple hormonal pathways (i.e. brassinosteroids), which then change expression of many genes. It seems plausible that this occurs in the *cesa6* mutant and in plants treated with Indaziflam for five days, as well as in roots facing large mechanical resistance to soil penetration. This may be particularly relevant for cortical cell growth in light of the recent results of Nolan et al. (Science 379: 1314 (2023) (not cited in this manuscript)) that highlighted the root cortex as a site "where brassinosteroids activate cell wall-related genes and promote elongation". Have you looked for evidence of activation of cell wall integrity signaling? What impact would such an effect have on the interpretation of the results and the mechanism of ethylene-induced swelling?

This study concludes that "CESA repression drives radial expansion of root cortical cells by modifying their cell wall mechanics, whilst epidermal cell wall strength is maintained. This differential pattern of cell wall regulation creates "stronger epidermis-weaker cortex" changes." I have some questions about this conclusion.

(a) I am uncertain what the authors mean by "stronger epidermis-weaker cortex". Do you mean the root epidermis restrains root growth, as is the case in hypocotyls, implying tissue tensions? If so, could you state the idea more clearly and perhaps discuss it as a consequence of stress anisotropy (Baskin, T.I., and Jensen, O.E. (2013). On the role of stress anisotropy in the growth of stems. *J Exp Bot* 64, 4697-4707. 10.1093/jxb/ert176.)? It would be helpful to show evidence for the role of tissue tensions, perhaps using the approach of Kelly-Bellow et al. (2023). Brassinosteroid coordinates cell layer interactions in plants via cell wall and tissue mechanics. *Science* 380, 1275-1281. 10.1126/science.adf0752.

(b) It seems to me that the ideas and data in this study run counter to those of Petrova, A., Gorshkova, T., and Kozlova, L. (2021). Gradients of cell wall nano-mechanical properties along and across elongating primary roots of maize. *J Exp Bot* 72, 1764-1781. 10.1093/jxb/eraa561). They propose that the inner cortex has the stiffest walls and serves as the growth limiting tissue in maize roots. Perhaps this paper could be cited and the contrary conclusions evaluated.

(c) The conclusions depend on AFM indentation methods that refer to Fusi et al. (2022), but this article lacks sufficient details for the method to be reproduced. What is the depth of indentation and force limit of the indentation? What is the speed of indentation? The graph in Ext. Fig 8b lacks numbers on its axes, which is not acceptable. The text states that the root cells are plasmolyzed, yet the methods state that the sliced sections were stored in water. Then why are the cells plasmolyzed? That does not make sense to me. Fusi et al report stiffness in units of pN/nm, which are correct units of stiffness, whereas in this manuscript they are reported in KPa, which are units for a modulus, not stiffness, and indicates the data were fitted to a model, which was not identified. This should be clarified and the fitting method described. The values given are as low as 200 Pa and 300 Pa; this is too soft to represent cell wall material properties, so I am unsure what the values mean and what kind of wall mechanical properties were measured.

(d) Can one validly compare stiffness values for epidermis and cortex, as implied by the phrase "stronger epidermis-weaker cortex"? Cell shape, cell connections, and tissue geometry likely influence the indentation results. These geometrical details are very different for epidermis and cortex, but are lacking in the text. Were they factored into the calculation of stiffness?

(e) The cellulose concentrations are based on results from a kit where the principle of the method is not described and no citations for validation are given. In reading the online methods booklet, I can guess that the reducing sugars that are reported may include cellulose and other glycans such as mixed-linked glucan and xylan (I don't think xylan is removed in the procedure). Thus it is difficult for me to assess the soundness of the method. What have the authors done to validate the cellulose measurements? The reported values of cellulose concentration may be subject to error. For instance, if the absolute cellulose amount is unchanged but other substances are accumulated (pectins, solutes, proteins), then the 'concentration' will appear to be reduced. Could this be the basis for the reduced value reported for compacted roots?

Some additional points:

1) Line 86: The text reads "To establish a connection between the changes in cellulose synthesis..." I did not see measurements of cellulose synthesis or changes in its synthesis.

- 2) Extended Fig. 7a shows a cellulose concentration of 0.062 for the *arf1* mutant, which is the same value as the value for WT quoted elsewhere. Shouldn't it be higher than in WT, according to the proposed action of ARF1 in suppressing CESAs? And according to the results in Figure 4? This seems to be an internal inconsistency.
- 3) Line 179 "Extended Data Fig. 8a;" - this figure does not show stiffness data. Maybe you mean Fig 8b? But Fig 8 b - needs numbers on the two axes;
- 4) Line 390 - "depicted in (b)"... but b is a graph, not an image.

In summary, this study provides new information implicating inhibition of cellulose synthesis in the root response to compacted soils, but lacks crucial information about cellulose anisotropy and lacks a specific model of how thinner walls and swelling of the cortex leads to root elongation, not merely root swelling (as occurs in the *rsw1* CESA mutant).

Referee #3

(Remarks to the Author)

Soil compaction induced ethylene accumulation, leading to root length reduction and radial expansion. This study reports a candidate pathway from compaction to ethylene and ARF1, which causes inhibition of CESA6, leading to root expansion. A *cesa6* mutant is first identified to show longer and swelled roots under compaction. Its upstream regulator ARF1 inhibiting CESA6 expression is then found to have longer and swelled roots under compaction. The root expansion/swelling is likely related to larger cortical cells with thin cell walls and reduced stiffness. Ethylene/ACC induced ARF1 but reduced CESA6 expression. This study provides some novel insights into the regulatory pathway of root expansion growth under compaction stress. However, the relation of the pathway with root elongation under compaction seems not that consistent. The following points should be addressed further.

- 1 , Line 70. The statement that 'the *oscesa6* mutant roots grew better than WT in compacted soil' is not accurate. 'grew longer' may be appropriate. In Fig 1c, the data may be subjected to further statistical analysis for multiple comparison. Then we could know clearly whether the mutant root length is longer than the WT in compact soils.
 - 2 , Fig 1b, the 'NG-CESA6' is a mutant complementation line. However, its root diameter in cross-section is much smaller than the WT one in compacted soil. Is the complementation affected by the transgene?
 - 3 , Fig 3d, since the CESA3 and CESA5 are not significantly induced in mutants or repressed in OE-ARF1 plants, the description in lines 123-124 is not proper. Please clarify.
 - 4 , Although the ARF1 binds to the promoter region of CESA6 and homologous genes, and inhibits their promoter activity and gene expression for possible root elongation in compacted soil, the genetic analysis of double mutant *arf1 cesa6*, in comparison with the single mutants and WT, should be performed to further confirm the interaction of the two genes in compacted soils.
 - 5 , In *arf1* mutant, the expression of CESA6 etc. is increased compared to WT. However, the cellulose content in the *arf1* is very similar to the WT level, in contrast to the expectation that the *arf1* should have more cellulose than WT. I would suggest that the cellulose contents in OE-ARF1 plants and *cesa6* mutant were all measured under non-compact and compact soils, and then compared to those in the WT and *arf1* under the same conditions to see if the cellulose contents are consistent with the authors' conclusion.
 - 6 , In Fig3c, the ARF1 gene is induced by high-density gel. The CESAs genes could be examined similarly to see what happens.
 - 7 , The OE-ARF1 plants showed constitutive root swelling. The authors may want to test the cellulose content to see if the ARF1 would inhibit this cellulose level.
 - 8 , Extended Fig1f, g. The cellulose contents are usually 30-50% in rice. However, in this analysis, the cellulose contents are only 5-6%. It is possible that the method used is not accurate. It would be much more accurate if the contents were measured by GC.
 - 9 , Line 153-168, the authors should clearly mention whether cell wall thickness has any direct correlation with cell wall stiffness, which may lead to radial expansion of roots.
 - 10 , Fig 4b, the cell wall thickness in WT is very similar to that in OE-ARF1 and *cesa6*, while the latter two genotypes grow longer roots in compacted soils. So this cell wall parameter may not be consistent with the longer root phenotype.
 - 11 , Fig 4d, the meaning or indication of the ratio should be explained.
 - 12 , Extended Fig 4, the expression of CESA6 and other homologs could be examined to see if they have any difference in different cell types of roots. Then readers may have an idea where the genes may work since WT cortical cells have thin cell wall while the epidermis cells have thick cell walls.
 - 13 , Fig 5b, ethylene-induced ARF1 expression depends on OsEIN2 but not on OsEIL1. Is it possible that this induction requires both OsEIL1 and OsEIL2?
 - 14 , Fig 5f, the authors found that the module involving compaction-ethylene-ARF1-CESA6 controls root swelling probably through cell wall remodeling. However, the relation of the root elongation with the pathway could not be well explained under compaction stress. The effects of compaction and ethylene are not consistent with those of ARF1 and *cesa6* regarding to root length. Actually, compaction stress and ethylene all induced short but swelled roots as derived from the Science (2021) paper of the same team. However, OE-ARF1 seems to induce longer but swelled roots under compaction, and *cesa6* mutant also had longer but swelled roots under compaction stress. The authors should recheck their data carefully to smooth their pathway for a specific root phenotype.
- The authors provided extensive data. However, the connection of these should be further examined in the proposed pathway, especially in terms of root elongation under compaction stress.
- 15 , Throughout the MS, all the figures and supplemental data Figures should be further analyzed statistically for multiple comparison, specifically for Fig1cd, Fig3fh, Fig4b-d, and Fig5e.

Version 1:

Reviewer comments:

Referee #1

(Remarks to the Author)

The authors have addressed all issues that I raised and I commend them on their work. A few minor issues:

Line 180 : “compliant walls” unclear what this means

Line 197: “Complementary Atomic Force Microscopy (AFM) analysis” not clear

Line 215: “stress accumulates in the outer rims” specify outer rims

Line 312: “we did not observe any significant trends in BR gene expression under different soil conditions (Zhu et al., co-submitted)³²” as the author list is different, “we” is not accurate

Rongfeng Huang lab: Specify affiliation

Referee #3

(Remarks to the Author)

This MS has been improved based on my previous comments. However, the following points may be further addressed for clarity.

1 , For extended Data Fig 9, the single *cesa6* mutant should also be included for comparison of phenotypes and root length measurement. After all the data are included, the genetic interaction of the ARF1 and CESA6 genes can be estimated. If the root (compacted) of the double mutant looks like the *cesa6*, then one may say that the CESA6 acts downstream of ARF1, further confirming other results. However, the root length of the double mutant under compacted condition showed no significant difference compared to *arf1* root length under the same condition. Did this mean that the CESA6 does not have strong effects on root length, acting downstream of ARF1? These should be further clarified.

2 , For Fig4 b, c and e, data from double mutant of *arf1 cesa6* is better included for comparison.

3 , The data about the epidermal cell wall thickness and stiffness from Extended Data Fig 10 may be placed in the normal Fig 4. Then readers would easily know the difference between cortex and epidermal cells.

4 , Fig 4j, the pathway at the most right needs some clarification. I would suggest that the ‘T’ symbols indicating negative regulation above cellulose biosynthesis and cell wall thickness may be replaced by arrows indicating positive regulation? Could the authors add cortex cells or epidermal cells here in the pathway? Otherwise, the pathway is quite difficult to understand since the authors mentioned two different cell types namely cortex and epidermal cells.

Referee #4

(Remarks to the Author)

I am commenting on the revised paper entitled “Ethylene modulates cell wall mechanics for root responses to compaction”. I read the initial reviews, author responses, and the revised paper. The initial reviews were comprehensive and constructive, and the authors were responsive. I will focus on follow-up with prior comments rather than launching into anything new.

Some of the revisions were un-nerving. New results that contradict the prior work are now introduced, the authors hypothesize that it was due to genetic background effects. Other new results greatly improved the paper, as the original submission contained minimal and barely sufficient results to support the interconnections proposed to mediate the growth response (e.g. including DCB was a good idea). Several new experiments with additional mutants and promoter analyses bolster the case for the ethylene-ARF-CESA transcriptional response components of the paper.

The major weaknesses that remain relate to trying to *cesa* and *arf* mutant cell wall phenotypes with the biomechanics of the root growth response. The geometries, tissue layer interactions and biomechanical parameters that drive root swelling are not known, and the authors make a mistake by trying to close this entire loop. It is not possible for any lab to cover all of this ground with limited data in one figure. Despite this major weakness, the work is highly novel and interesting without this poorly executed work in Figure 4.

Major weaknesses:

1) The cell wall thickness measurements are not reliable and could not be reproduced based on the methods. There is no description of the protocols followed to identify similar regions across genotypes. There is no analysis of the spatial heterogeneity in wall thickness within individual cells, only the aggregated datapoints in Figure 4c. The example images in 4b are at an insufficient magnification to reveal the wall boundaries that are used to measure thickness. Many example images should be shown in the supplemental and the sampling needs to be better justified. The way it is done now is very subjective, creating opportunities for investigator bias in choosing the points for measurement. This type of experiment would be best conducted blind.

2) The stiffness values and methods are better described. However, collecting data on selected regions of sectioned

material does not capture the stiffness of intact cells and the parameters measured do not necessarily correspond to those that generate the radial swelling response. These limitations need to be fully acknowledged rather than over-sold.

3) The links between wall material properties (thickness and stiffness) are unreliable, but even if those data were collected and analyzed with more care, using these data and the geometries of the transitions of tricellular junctions to make conclusions about the biomechanical controls of organogenesis is not acceptable. These questions are beyond what can be addressed with the current data and methods in any paper, because the tools to accurately measure stresses and the underlying material properties that are important are not known. The work and conclusions which would only muddy the water for the field and create confusion. I agree with the reviewer in the first round that was directing the authors in the direction of FE modeling, which has its limitations, but at least the underlying shell theory and ability to accommodate complex geometries enables FE to make plausible and testable predictions. The unpublished mechanical model used here was developed based on the biomechanics of epithelium morphogenesis and is centered on tensional forces at vertices driven by coupling of cortical actomyosin-dependent contraction with the ECM. These mechanics do not apply to the system under study here. The authors would be better served to use their time-lapsed data to analyze strain rather than to simulate the biomechanics of the system. They would at least have direct and accurate data on growth rate differences between tissues and among genotypes.

Version 2:

Reviewer comments:

Referee #3

(Remarks to the Author)

This MS has been improved further based on my comments. I have one comment about the working model. In Fig 4k, the genetic relation in the working model in the most right panel is not consistent with the results, and from this model, the soil compaction/ethylene would inhibit root swelling and penetration. Actually, the relation between 'cellulose synthesis' and 'thinner/softer cortical wall' should be a negative regulatory manner since *cesa6* mutant showed thin cortical wall (Fig4e). A blunt 'T' end should be added above the term 'thinner/softer cortical wall' to replace the arrow. In this way, the 'soil compaction/ethylene accumulation' would finally promote 'root swelling' and 'root penetration ability', largely consistent with the results. Please clarify the model.

Referee #4

(Remarks to the Author)

The authors took the prior comments seriously and the resubmitted work clearly demonstrates the cell wall thickness phenotype. However, the AFM experiment and the FE models developed are not conducted or interpreted in a reasonable way. As a result, the overall biomechanical explanation of ethylene-mediated radial growth and the conclusions presented throughout the paper are not substantiated.

1) AFM on sectioned root material is a useful approach to compare the relative stiffness of cell wall domains from different genotypes. However, it cannot be used to estimate the properties of cell walls *in vivo*. The loading conditions, wall geometry, and properties in the AFM experiment are completely different from the living state. The stiffness values, estimated by the authors to be in the ~1 kPa range, are about 104 to 105 times less than the stiffness measured from native or hydrated and intact cell walls (e.g. Cosgrove, *Science*, 2021; Turner, *Plant Phys.* 2021). The AFM should be treated as a phenotyping tool and not a method to extract realistic values for cell wall properties. The authors continue to over-interpret the AFM data:

"We acknowledge, however, that the AFM measurements were done on cross 212 sections of fixed cells, which might not directly relate to cell wall stiffness in living 213 root cells."

These errors are compounded further by using AFM-generated stiffness values to inappropriately parameterize an FE model (see #2 below).

2) In this paper, at best, the FE method can be used to determine if a model in which wall thickness in the cortex could alter root expansion patterns is plausible. There are too many unknowns and assumptions in the model for it to "validate": many wall material parameters are guessed, the geometry and mechanics of the tissue connectivity are not known, the boundary conditions have unknown effects. This point is important because the power of the model is vastly overestimated.

The authors state: "These observations suggest that a "thicker 221 epidermis-thinner cortex" cell wall model facilitates root adaptation to soil compaction. 222 As an attempt to validate this model, we conducted simplified finite element (FE) 223 simulations examining how variations in cell wall thickness and stiffness influence root 224 tissue mechanical behavior during changes in external mechanical pressure."

More problematic is the inappropriate parametrization and design of the model. A key parameter, Modulus/E is assigned to be 1 kPa based on the AFM, this is not at all realistic (see above). All of the resulting strains are therefore not physiologically relevant. In addition to these major weaknesses in model implementation, the author's failure to conduct simulations that directly test the central concept of the paper, which is that cortical wall thinning is a plausible mechanism to promote radial expansion. The FE implementation therefore is severely flawed and fails to advance the hypothesis.

Minor points.

1) The FE element approach needs further explanation in order for the reader to understand why it is an appropriate method and its demonstrated utility in the field of morphogenesis. The authors could cite its prior use to discover of an importance of cell wall thickness gradients in morphogenesis control (Yanagasawa et al. Nat. Plants , 2015). The key features of the model and how cells are connected needs to be introduced. Each model in any publication should be uniquely named and made publicly available.

2) To much space is allocated to the cell wall thickness methods. This section can be condensed with methodological details moved to the methods section.

"To test this 189 hypothesis, we examined cortical cell wall thickness of WT, arf1-1, OE-ARF1, cesa6, 190 and arf1 cesa6 in the root elongation zone using transmission electron microscopy 191 (TEM). Tips (1 cm) of roots grown in normal and dense media for five days were 192 chemically fixed and processed. After determining the elongation zone position using 193 semi-thin transverse sections (Extended Data Fig. 10), we prepared ultrathin sections 194 for measuring cortical cell wall thickness (Extended Data Fig. 11). To account for cell 195 wall thickness heterogeneity, we measured a single WT cortical cell divided into eight 196 segments and found that cell wall thickness variations can effectively be averaged by 197 measuring wall thickness at fifteen different points (Extended Data Fig. 12). Using this 198 approach, we found that the wall thickness of WT cortical cells decreased significantly 199 under dense media conditions compared to normal conditions (Fig. 4a-c,e). By contrast, 200 arf1-1 maintained thick root cortex walls in both non-compacted and compacted 201 conditions, whereas OE-ARF1, cesa6, and arf1 cesa6 consistently exhibited thinner 202 walls regardless of growth conditions (Fig. 4a-c,e). To further characterize cell wall 203 properties, we harvested root tips from all genotypes grown in dense agar media, 204 prepared 50- μ m thick longitudinal sections and assessed cortex cell wall stiffness using 205 atomic force microscopy (AFM)."

Version 4:

Reviewer comments:

Referee #3

(Remarks to the Author)

The working model has been corrected based on my previous comment and I have no further comments.

Referee #4

(Remarks to the Author)

The authors have responded to comments regarding over interpretation of the AFM results. The FE has been removed. This was an unfortunate waste of time. The FE could have been useful, but it is a lot to take on. I understand the change. The cost of this is that there is no test of the biomechanical plausibility of this system in terms of explaining root adaptation to compaction. This gap should be integrated into the paper and two suggestions in this vein are provided below. Otherwise my concerns have been addressed and the paper is now greatly improved.

1) The first sentence of the conclusion is strange. The paper does not demonstrate how the model responds to compaction. The FE would have been the first step in that direction. The paper aggregates the data to generate a plausible model of how root response to compaction occurs.

275 Our study provides new conceptual insights into how roots adapt to soil compaction 276 stress through cortical cell expansion, demonstrating how a "thicker epidermis-thinner 277 cortex" model effectively responds to soil compaction (Fig. 4k).

Is something like this more accurate?

275 Our study provides new conceptual insights into how roots adapt to soil compaction 276 stress through cortical cell expansion, and provides a plausible "thicker epidermis-thinner 277 cortex" model of how the root effectively responds to soil compaction (Fig. 4k).

2) Again in the conclusions section. The authors add the nice analogy:

"The " thicker 289 epidermis-thinner cortex " model therefore parallels engineering principles used in pipe 290 design, where maintaining perimeter stiffness while increasing diameter improves 291 structural stability.

This is where a sentence about the need for future testing/analyzing of the biomechanical plausibility of this model in roots should be added. It is not proven in this paper.

Referees' comments:

Referee #1 (Remarks to the Author):

In the manuscript “Ethylene dynamically controls cell wall mechanics to support root responses to compaction” by Jiao Zhang and colleagues, the authors investigate how ethylene leads to radial expansion of root cells and their cell wall properties by a cascade that involves ethylene, Auxin Response Factor 1, and Cellulose Synthase genes. Based on their findings they postulate a “stronger-epidermis-weaker cortex” model that links many of their observations with functional root growth outcomes in compacted soils. Overall, the authors report very interesting findings that link regulatory responses to root physiology and might have important conceptual and practical relevance.

The manuscript is written well and in a concise manner. The work is technically very impressive, elegant, and conceptually interesting while grounded in state-of-the-art experimental data. However, there are several potentially profound issues highlighted below that need to be considered.

Major

1. The *arf1-1* mutant does display much slower growth in non-compacted and compacted soils (Fig 2E). There is a major concern that the general growth phenotype of the *arf1-1* mutant might confound results and conclusions.

o The *arf1-1* mutant has a general growth phenotype; how can the authors disentangle this from the response to compacted soil/agar? Could the observed phenotypes a stress response due to the general growth defects?

Response to comment: We agree with these comments from the reviewer. Following the recommendation, we conducted additional experiments by growing wild type and *arf1-1* in germination bags, virtually in liquid media, without any confounding solid media affecting growth. Under these conditions, *arf1-1* roots grew similar to wild type (see new Figures 3d and 3e). These findings support our conclusion that OsARF1 plays an important role in rice root responses to dense media and compacted soil.

o The authors claim that the application of low concentrations of Indaziflam ameliorated the growth defect of *arf1-1* mutant roots on media (Extended Data Fig. 7b,c), but there are no mock treatment controls. How can they exclude that they exposed WT to a stress by this treatment, and it thus displays the same growth defects as observed in *arf1-1*? If it is like this, it might not support their conclusions.

Response to comment: We certainly agree with the reviewer. We therefore conducted additional experiments incorporating both mock (DMSO) and treated (150 pM Indaziflam) conditions (see new Extended Data Figure 7), In addition, we confirmed the results using an additional cellulose synthesis inhibitor, DCB, with appropriate controls and found similar effects as with Indaziflam (see new Extended Data Figure 8). These new results demonstrate that cellulose inhibitors effectively rescue the phenotypes of *arf1-1* under dense media conditions.

o According to the authors, TEM imaging revealed that the wall thickness of WT cortex cells was reduced in compacted conditions compared to non-compacted conditions (Fig. 4a,b; Extended Data Fig. 1c). How did they make sure they imaged comparable sections of the root in WT and *arf1-1*?

Response to comment: This is a crucial point made by the reviewer. After embedding the roots in resin, we sampled 1 cm root tips and minutely checked the position (elongation zone) of the sections under a Nikon H600L light microscope (Tokyo, Japan) to make sure that we targeted the elongation zone. Here, we started to cut the sections and stain them to make our observations. We have also clarified this in the methods section of the revised paper. While different cell types inherently maintain variations in cell shapes and sizes, we made sure that we selected cell walls of the outmost cell layer and of cortex layers in the elongation zone (as also indicated by the boxes in Figures 4a and 4d).

o In case of multi-genotype, multi-condition comparisons, 2-way ANOVA will be required to draw firm conclusions. For instance, the comparisons of WT, *arf1-1*, and OE-ARF1 lines on normal and dense agar media, or in non-compacted and compacted soils will require this. Otherwise, the results can be misleading.

Response to comment: We thank the reviewer for this suggestion. We have therefore revised our approach and applied a two-way ANOVA to re-analyze the significance among multiple groups in Figures 1b, 1f, 3h, 3i, 3j, 4c, and Extended Data Figures 1b, 1d, 2d, 4c, 4e, 5c, 7b, 8b, 9b, 10b and 13b. The results of this re-analysis confirm our original conclusions, providing further support to our findings.

2. The authors pose the model that ARF1 that regulates cellulose synthases is regulated by ethylene to achieve root penetration via leading to a stronger-epidermis and a weaker cortex layer. However, the data is not as clear-cut as it might seem and some of the results contradict the “stronger-epidermis-weaker cortex” model or at least might diminish its relevance.

o Is ARF1 mainly regulated by ethylene? Line 193ff:” In contrast to IAA and ABA, ACC treatment resulted in strong induction of OsARF1 expression as observed in dense agar media, (Fig. 5a). This result suggests that OsARF1 is mainly regulated by ethylene, rather than via auxin or ABA.” However, from the figure it seems obvious that all treatments elicited changes. In particular, the ABA treatment seems to lead to ARF1 upregulation in a region (elongation-differentiation zone) that might be of relevance for ARF1 function. It is also not clear whether the cross-sections and the longitudinal GUS pictures agree. A quantitative approach for GUS quantification might be helpful. This is of particular importance, as ABA has been described (also by the accompanying manuscript) to have a role in root penetration.

Response to comment: We appreciate your suggestions on this matter. Following the suggestions, we have conducted longitudinal sections of GUS-stained roots, which provide clearer distinctions between different cell types. Our findings reveal that 100 μ M ACC treatments (see new Extended Data Figure 12a) produced effects similar to those observed under dense media conditions (cf. Figure 3a), suggesting that ethylene may be the primary mediator linking compaction and *OsARF1* expression levels. In Extended Data Figure 12a, we also observed that 25 μ M ABA induced *OsARF1* expression in the cortex area, albeit to a lesser extent than ACC. Interestingly, the GUS line showed no apparent response to 10 nM IAA treatment. To further quantify these observations, we performed GUS quantification and RT-qPCR assays, as shown in new Extended Data Figures 12b and 12c. These analyses revealed significant increase in *OsARF1* expression levels in ACC-treated conditions, but reduced expression in ABA-treated conditions. This reduction in ABA-treated samples may be due to changes in overall *OsARF1* expression in the roots (as RT-qPCR is conducted on whole roots) that could dilute the increased local expression observed in our GUS analyses. In conclusion, our results support that ethylene induces *OsARF1* expression. We speculate that ABA may act downstream of ethylene as per results from (Huang et al., 2022), thus partially modulating *OsARF1*, but these relationships would need further research. We have revised the manuscript in accordance with the above updated results.

o Ethylene signaling is disrupted in *ein2* and *eil1* mutants. However, responses of the *ein2* and *eil1* mutants seem sometimes to be similar and sometimes not. E.g. Line 211: “Additionally, we discovered that cortex wall stiffness remained unchanged in *ein2* compared to WT, whereas *eil1* cortex cell walls were stiffer (Extended Data Fig. 10c), suggesting that they are not more penetrative than WT in our view.” (also note that the authors should elude more on what their view means) and then Line 214: “In addition, both *ein2* and *eil1* displayed reduced epidermal wall stiffness compared to WT (Extended Data Fig. 10d)” and then Line 207ff: “To further explore these apparent inconsistencies, we conducted growth experiments where we germinated and grew WT, *ein2* and *eil1* on different media conditions. Our results indicate that while *ein2* exhibited similar penetration characteristics to WT, *eil1* performed worse (Extended Data Fig. 10a,b).” So how does this fit all together? Where are the differences between mutants that supposedly are insensitive to ethylene come from? How does that fit with previous studies?

Response to comment: We are very grateful to the reviewer for the comments on the ethylene-related results. Indeed, we also thought that some of the results were a bit odd during our first submission. After substantial checks, we discovered that the previously used *ein2* and *eil1* seeds, obtained from another lab, contained a DR5 marker line with a different wild-type background, which may have caused genotype-related alterations in our measurements. We have therefore re-run all the results related to the ethylene experiments in our revised manuscript.

Using correct *ein2* and *eil1eil2* lines, we found that both lines exhibited enhanced root growth on dense agar media conditions as compared to WT, consistent with data in soil conditions. Our new AFM measurements also revealed that both *ein2* and *eil1eil2* mutants maintain softer cortex tissue and stiffer epidermal layers, aligning with our proposed model (see new Figures 4g and 4h). Notably, neither *ein2* nor *eil1eil2* mutants showed root swelling to compaction in either soil or media

conditions. We hypothesize that this lack of response may be attributed to impaired ABA biosynthesis in both mutants (see relationship of ethylene and ABA in root compaction in Huang et al., 2022), which consequently affects root expansion. Nevertheless, roots of both mutant lines respond to cellulose inhibitors similar to WT, indicating that ethylene signaling may impact other aspects than only cell wall strength in root growth on compacted media. We have revised the text in accordance with these new results.

o *ein2* had better penetrability but reduced epidermal wall stiffness. How does that fit with the “stronger-epidermis-weaker cortex” model?

Response to comment: We re-ran these analyses (as per above). Our results demonstrate that *ein2* and *eil1eil2* mutants exhibit softer cortex tissue and stiffer epidermal layers, consistent with our mechanical characterization by modeling.

o Line 215: “Interestingly, when we treated *ein2* and *eil1* with 150 pM Indaziflam, the roots became thicker (Fig. 5d,e), corroborating that cellulose synthesis changes occur downstream of OsEIN2 and OsEIL1 to increase radial swelling of rice roots.” How can it be excluded that it is not independent? What does it mean with regards to differences of *ein2* and *eil1*? Is corroborated the right word?

Response to comment: We thank you for these comments. Indeed, we cannot rule out that the responses of the cellulose synthesis occur independently of the ethylene related mutants. Hence, we revised the text to “The resulting root thickening (Extended Data Fig. 13) suggests that cellulose synthesis modifications may occur downstream of OsEIN2, OsEIL1 and OsEIL2 to enhance radial swelling in rice roots, though the relationship between these pathways requires further investigation.” in line 262-265.

o Line 199: “The results showed that OsARF1 induction by ACC was suppressed in *ein2*, and to a lesser extent *eil1* (Fig. 5b).” According to the presented data, the effect of *eil1* was not significant. This should be clarified and put into context.

Response to comment: We appreciate this comment. As per response above, we reran the *OsARF1* expression analyses for *ein2* and *eil1eil2* backgrounds, and found that the compaction-induced upregulation of *OsARF1* was clearly suppressed in *ein2*, but only to a lesser extent in *eil1eil2*, as compared to WT (in Fig. 4i). These results indicate that *OsARF1* may function downstream of *OsEIN2*, but that a more complex regulation of the gene may occur via *OsEIL1* and *OsEIL2*. Notably, the suppression of several of the primary wall *CESAs* in response to compacted media is released in *ein2*, but these relationships are changed in *eil1eil2*. We have revised the text to better convey these changes and possible regulatory connections.

3. Complementation Fig S2: Why the variance of the AtCESA6 complementation that high. Are the phenotypes of the complemented line significantly different from those of the mutant?

Response to comment: In response to the reviewer’s concern, we have repeated the experiment using new seed batches of NG-CESA6 lines, with results presented in Extended Data Figures 2c

and 2d. These data confirm that the construct rescues the *cesa6* mutant phenotype under both normal and dense media conditions.

4. Statistics: There are several figures in which it was not indicated whether differences were significant (and at which significance level or not), e.g. Fig S1a.

Response to comment: We have revised our statistical testing and applied a two-way ANOVA with Tukey test to re-analyze the significance among multiple groups in Figures 1b, 1f, 3h, 3i, 3j, 4c, and Extended Data Figures 1b, 1d, 2d, 4c, 4e, 5c, 7b, 8b, 9b, 10b and 13b; a one-way ANOVA with Dunnett test in Figures 2f, 3b, 3c, 4e, 4g, 4h, and Extended Data Figures 10c, 12b, and 12c; a Welch's t-test with Bonferroni-Dunn in Figures 2e and 4i; a Welch's t-test in Figures 1d and 3e. The results of this re-analysis confirmed our original conclusions.

5. The ChIP analysis doesn't seem to be very rigorous. The precipitate ProARF1:gARF1-GFP was compared to the WT precipitate (negative control). However, it would be good practice to normalize to one or several other genomic regions that are expected not to be enriched – or (better) to conduct a ChIPseq experiment.

Response to comment: We greatly appreciate this comment. In response, we have included a negative control consisting of sequences lacking AuxRE-L binding motifs in the *OsCESA6* promoter region, as shown in new Figure 2e. Our results demonstrate that OsARF1 shows no binding affinity to this negative control sequence, thus validating the specificity of the interaction.

6. It is unclear how the scRNA data (Extended Data Fig. 4) that is part of the companion manuscript is necessary; if it is used it should be explored a bit more (e.g. is OsARF1 expressed to the same extent in all tissues upon soil compaction? Is OsARF1 co-expressed with other genes of interest in this study, for example its direct targets?).

Response to comment: We concur with your assessment regarding the relevance of the scRNA seq dataset for our study. Since the scRNA-seq experiments were conducted primarily on two- to three-day-old roots, while our observations focused on the elongation zones of five- to seven-day-old root tips, direct comparisons would be methodologically inappropriate. Given these differences, we have opted to remove this dataset from our manuscript.

7. Line 201 ff: "Consistently, OsCESA3, 5, 6, 8 were down-regulated after ACC treatment in WT; however, not in *arf1-1*, supporting that OsCESA3, 5, 6, 8 expression needs OsARF1 to be regulated by ACC (Fig. 5c)." This is not correct. CESA3 was not significantly downregulated. Also, it seems that CESA5 and 6 are upregulated in the mutant upon ACC, but it is labelled as NS while the error bars and the effect size seem to be comparable to some other significant labelled data in this figure. The authors should carefully check.

Response to comment: We greatly appreciate these comments. In response, we have re-examined the expression levels of the main primary wall *CESAs*, *CESA5*, *CESA6*, and *CESA8* (we removed *CESA3* here, because OsARF1 did not directly bind to its promoter in our yeast one hybrid assays),

in both normal and dense media conditions, comparing wild type (Nipponbare), *ein2* and *eil1eil2* mutant (see new Fig. 4i) backgrounds. Our results demonstrate that media compaction significantly reduces the expression levels of *CESA5*, *CESA6*, and *CESA8* in wild type roots, while this response is suppressed in *ein2* (Figure 4i). Interestingly, in the *eil1eil2* background, the three *OsCESA* genes showed distinct responses: *OsCESA5* maintained its downregulation, *OsCESA6* showed no significant change, and *OsCESA8* was upregulated (Fig. 4i). These differential responses highlight the complex regulatory network connecting ethylene signaling and cellulose synthesis (see also comment/answer above regarding ARF1's regulation in *ein2* and *eil1eil2*).

Minor

1. Line 180: "Taken together, these observations suggest that a "thicker epidermis-thinner cortex" cell wall model may drive the ability of plant roots to adapt to soil compaction." How can the authors know that this drives this and is not just linked to processes driving it?

Response to comment: We agree with the reviewer and have revised this statement to "These observations suggest that a "stiff epidermis-soft cortex" cell wall model facilitates root adaptation to soil compaction." in line 203-204; and "Our results show that under soil compaction, stress accumulates in the outer rims and endodermis and decreases in the cortex." in line 214-216.

2. L95 "From these assays, we found several ethylene response factors (ERFs), including OsERF34 and 39 (binding to seven and five promoters, respectively; Fig. 2a). " Are ERFs statistically enriched?

Response to comment: Thanks for this comment. Indeed, ERFs show significantly enriched in our Y1H screening results (See below: TF families shown in the left were statistically significantly enriched in our Y1H screening results. The size and color of circle represents the number of interactions identified in our screenings and $-\log_{10}$ value of P value of binomial test to evaluate statistical enrichment, respectively. Horizontal axis represents \log_2 value of the fold enrichment to the expectation. Arrows indicate circles which are tiny due to low number of the identified interactions). While we have not included these results in the revised manuscript, we are happy to do so if the reviewer thinks that this is needed.

3. In line 132 the authors correctly conclude without overstating :”Hence, OsARF1 activity appears to control root penetration ability, particularly during soil compaction stress conditions.” However, the title of the section “OsARF1 is upregulated by compaction stress and represses CESA gene expression“ seems slightly overstated

Response to comment: We thank you for this comment. We have exchanged the subtitle to “OsARF1 regulates *CESA* expression to control root compaction responses” in line 124.

4. The titles of Extended Data Fig. 9 and Fig 4 seem to be slightly misleading

Response to comment: We have revised the titles of the extended data files as per this suggestion. They now read: Fig. 4 (A “stiff-epidermis and soft-cortex” cell wall model facilitates rice root adaptation to soil compaction); Extended Data Fig. 10 (Thickened and stiff epidermal cell walls facilitate root adaptation to soil compaction).

5. Line 199: “three h” to “three hours”

Response to comment: Thanks, we have corrected this.

6. Line 199: “The results showed that OsARF1 induction by ACC was suppressed in *ein2*” is suppressed the right word here?

Response to comment: We agree with this comment and we have revised the text to: “Under dense media conditions, *OsARF1* expression was significantly induced in both *Nip* and *eil1eil2* mutants, but not in *ein2* (Fig. 4i). This suggests that *OsARF1* functions downstream of *OsEIN2*” in line 246-249.

7. Indaziflam is used in a number of experiments. It would be good to comment on the specificity of the inhibitor.

Response to comment: We appreciate this comment. In response, we have incorporated a description of Indaziflam's function in the revised manuscript (line 66-68). Please note that we also confirmed these results using another cellulose inhibitor DCB which shows similar results. We have therefore also included a brief description of DCB in the revised manuscript (line 163).

8. Introduction:” While the mechanical strength of cell walls is crucial for anisotropic cell expansion, little is known about how cell wall properties regulate root penetration of soil, in particular, compaction stress conditions.” This sounds like an overstatement. See for instance Schneider et al. 2021 and referenced work therein.

Response to comment: We agree, and we have therefore revised this statement to: While the mechanical strength of cell walls is crucial for anisotropic cell expansion, little is known about how cell wall properties regulate root penetration of soil (line 58-60).

Referee #2 (Remarks to the Author):

This manuscript identifies some new pieces in the puzzle of how roots respond to compacted soils by cortical cell swelling, building on previous work indicating this root growth response is mediated in part by ethylene. Zhang et al. report that ethylene upregulates the transcriptional repressor ARF1 which reduces cellulose synthase (CESA) gene expression in roots of rice seedlings grown in soils of high bulk density. The manuscript shows that cell walls of cortical cells are thinner in roots grown in compacted soil and display greater resistance to lateral indentation by a 20-nm AFM probe. In contrast, epidermal/outer cortical cells do not change in these ways. These changes are attributed to repression of cellulose synthesis, which may be selective for cortical cells, but I do not think this aspect is actually demonstrated. Could this specificity be demonstrated by in-situ hybridization assessments of CESA transcript levels?

Response to comment: We certainly agree with this comment. We tried the *CESA in situ* hybridization experiments many times, but these experiments did not work and we have therefore addressed this question through alternative GUS promoter based approaches. First, we performed longitudinal sections of *ProOsARF1*:GUS roots and found that dense media primarily enhanced *OsARF1* expression in the cortex of the elongation zone rather than in the epidermal layer (see new Figure 3a). This observation supports a reduced expression of primary wall *CESAs* in the cortex under these conditions, as ARF1 suppresses the *CESA* expression. We further examined *CESA* genes expression under different agar conditions and discovered that dense media significantly reduced *CESA* gene expression levels in wild type by RT-qPCR analysis (Figure 4i). Furthermore, we generated *ProOsCESA6*:GUS lines and examined these on different media. Here, GUS activity was substantially reduced in roots grown on dense media in cortex cells of elongation zone (see Extended Data Fig. 6). We hope that these additional data address your concerns.

The evidence for ARF1 regulation of *CESAs* includes (a) promoter analysis and (b) phenotypes of *arf1* and ARF1 overexpression lines, the *cesa6* mutant, and comparison with plants treated with the *CESA* inhibitor Indaziflam. The evidence for this conclusion seems sound and convincing.

Response to comment: We appreciate this comment and the support of the work.

The headline for Fig 1 is that *CESA6* negatively regulates root penetration of compacted soil. However, I found the reasoning and data confusing. The data in figure 1c indicate that root length is relatively less impacted by 'compacted' soil in the *cesa6* mutant than in WT (root length is decreased by 13% versus 34%). This *cesa6* phenotype is curious and seems to be the foundation of the current study as told here. I note that statistical significance tests are not shown for this figure (likewise for Fig 3f and some others). Moreover it seems that the relatively diminished effect of soil compaction in the *cesa6* mutant arises mostly from the inhibition of root elongation by the *cesa6* mutation: its growth is already stunted in non-compacted soil (Fig. 1c, but where is the statistical test?), so there is less effect of compacted soil. These results seem at odds with the photograph of the *cesa6* mutant in Fig 1a where the root appears a bit longer in non-compacted soil than WT. Not

shorter, as Fig 1c indicates. Am I misinterpreting the results? Also, the Y-axis does not start at zero in 1c and 1d, which contributes to the difficulty for interpreting the graphs.

Response to comment: We appreciate the comments here, and hope the below can provide further clarification to the questions raised by the reviewer. While WT and *cesa6* exhibit similar phenotypes in non-compacted soil, *cesa6* demonstrates enhanced performance compared to WT specifically under compacted soil conditions. We have performed two-way ANOVA analysis to validate this differential response, and the results support our conclusions (see revised Figure 1b). Furthermore, to exclude the possibility that this phenomenon might be attributed to inherent phenotypic defects of *cesa6*, we compared the growth of WT and *cesa6* in germination bags, i.e. basically in liquid media without any confounding effects from differences in media density. Our results showed no significant differences in root length between the two genotypes under these conditions (Figures 1c and 1d). These findings support that some suppression of cellulose synthesis may enhance root elongation and radial expansion under soil compaction, or at least dense media, conditions.

Cortical cell diameters are substantially increased in roots grown in high-density soil (Fig. 1d). This follows the trend of increased root width (Fig. 1b), but the greater root width is apparently not solely due to wider cortical cells but additional changes in root anatomy may contribute: judging from the photographs in Fig 1b, there are changes in the number of cell layers in the cortex, the sub-epidermal layers, and in the stele. These aspects are not discussed in this study.

Response to comment: Thank you for your suggestion. We agree that cell expansions in exodermis, cortex and stele all contributes to root swelling (as demonstrated in our co-submitted paper). However, the cell expansion is clearly most pronounced in the cortex cells. We also agree that we sometimes noted differences in cell numbers/layers in the root sections, but these differences were not specific to a certain treatment. Thus, we conclude that cortical cell swelling serves as the principal mechanism driving the root thickening.

I infer from the data in the manuscript that the anisotropy of root cell growth is greatly changed upon growth in dense soil, although growth anisotropy is not measured or discussed. A change in growth anisotropy is a classic effect of ethylene and is traditionally considered to be a consequence of alteration in the orientation of cortical microtubules, leading to changes in the orientation of cellulose microfibril deposition. This potential mechanism is not discussed in the manuscript. This seems to be a notable gap, as I suspect it contributes to the root response to compacted soil. Have the authors characterized changes in cortical microtubule and cellulose microfibril orientations in the compacted roots?

Response to comment: We agree with these comments. We have now included a brief discussion related to the reviewer's comments, i.e. how cell wall anisotropy and microtubule organization may contribute to the observed phenotypic outcomes, in the revised manuscript (see lines 294 to 302). To try to directly address the reviewer's concerns, we also set out to assess microtubule organization in rice root cells grown under varying agar concentrations. We note that this is a challenging task as rice is inherently more difficult to image than Arabidopsis owing to its multiple cell layers and thickness. Nevertheless, we attempted to image the microtubule organization at two regions of the

roots; in elongating cells just above the proliferation zone and just below the maturation zone (see images below). While we did not observe any substantial differences in microtubule organization in the epidermal cells (n=5 cells from 20 roots) in the different conditions, we noted that dense agar concentrations appeared to yield an increased frequency of microtubule ends (red arrows) and shorter microtubules (yellow arrows). While these observations might be interesting, we acknowledge that this might be potential artifacts from mechanical stress during root extraction from the agar medium (we could only image roots from the two different media by first growing them on vertical plates and then move them to the cover slips). Please note that the imaging could only be performed on the outer cell cortex of epidermal cells due to dim signal in cells of cortex cells. We are happy to include these results in the manuscript, with added quantification, if the reviewer deems this essential.

In an attempt to visualize microfibril arrangements, we developed a marker line by fusing mNeonGreen with the OsCESA6 genomic sequence driven by its native promoter (*ProOsCESA6:mNeonGreen-gCESA6*). Unfortunately, the fluorescent signal intensity in rice seedling roots proved insufficient for reliable visualization of CESA tracks at the plasma membrane. The main fluorescence signal emanated from intracellular objects, most likely Golgi (see representative images below). Nevertheless, as most CESA complexes track along cortical microtubules, we would expect a result aligned with the above microtubule imaging.

CESA mutations and cellulose biosynthesis inhibitors are known to induce wall integrity signals that activate multiple hormonal pathways (i.e. brassinosteroids), which then change expression of many genes. It seems plausible that this occurs in the *cesa6* mutant and in plants treated with Indaziflam for five days, as well as in roots facing large mechanical resistance to soil penetration. This may be particularly relevant for cortical cell growth in light of the recent results of Nolan et al. (Science 379: 1314 (2023) (not cited in this manuscript)) (BR signal promotes a shift from proliferation to elongation associated with increased expression of cell wall-related genes in cortex) that highlighted the root cortex as a site “where brassinosteroids activate cell wall-related genes and promote elongation”. Have you looked for evidence of activation of cell wall integrity signaling? What impact would such an effect have on the interpretation of the results and the mechanism of ethylene-induced swelling?

Response to comment: We are thankful for the valuable comments. We certainly agree that changes in cell wall integrity will influence gene expression. These changes may be difficult to interpret as RNA seq analyses will include many different cell types that then may be exposed to differences in cell wall integrity changes with corresponding differences in gene expression. We therefore checked genes that have been associated with cell wall integrity signaling (mainly related to brassinosteroid-related processes as per the reviewer’s comments) from the scRNA-seq data in the accompanying study. However, we did not see much changes of the most obvious candidates (see list below) in their expression when comparing non-compacted and compacted soil conditions. It is important to here note, that most of these candidates come from the Arabidopsis field with little corresponding data in rice. In response to the reviewer’s comments, we have also included a segment in the discussion acknowledging that we cannot rule out influence of additional factors that may be regulated by cell wall integrity signaling transduction (see lines 308 to 313), and we have added the reference noted by the reviewer.

This study concludes that “CESA repression drives radial expansion of root cortical cells by modifying their cell wall mechanics, whilst epidermal cell wall strength is maintained. This differential pattern of cell wall regulation creates “stronger epidermis-weaker cortex” changes.” I

have some questions about this conclusion.

(a) I am uncertain what the authors mean by "stronger epidermis-weaker cortex". Do you mean the root epidermis restrains root growth, as is the case in hypocotyls, implying tissue tensions? If so, could you state the idea more clearly and perhaps discuss it as a consequence of stress anisotropy (Baskin, T.I., and Jensen, O.E. (2013). On the role of stress anisotropy in the growth of stems. *J Exp Bot* 64, 4697-4707. 10.1093/jxb/ert176.)? It would be helpful to show evidence for the role of tissue tensions, perhaps using the approach of Kelly-Bellow et al. (2023). Brassinosteroid coordinates cell layer interactions in plants via cell wall and tissue mechanics. *Science* 380, 1275-1281. 10.1126/science.adf0752.

Response to comment: Thank you for this comment. We would like to clarify that our model is based on the differential responses of epidermal and cortical layers to non-compacted versus compacted soil conditions, rather than a direct comparison between epidermal and cortical cells. Our data show that soil compaction triggers increased cell wall thickness and stiffness in the epidermis, while simultaneously inducing cell wall thinning and reduced stiffness in the cortex. This dynamic response pattern is further supported by our mathematical inferences in Figure 4f based on cell wall stress patterns across different cell types of a root in compacted vs non-compacted soil. These results are based on ForSys (a noninvasive stress inference simulation method to compute mechanical stress on cell walls) estimates as described in Method section and Figure S11 in the revised manuscript. We have also included a brief discussion about the regulatory role of BR in cell growth and cell wall mechanics. Finally, we have modified our statement of “stronger epidermis-weaker cortex” to “stiff epidermis-soft cortex”.

(b) It seems to me that the ideas and data in this study run counter to those of Petrova, A., Gorshkova, T., and Kozlova, L. (2021). Gradients of cell wall nano-mechanical properties along and across elongating primary roots of maize. *J Exp Bot* 72, 1764-1781. 10.1093/jxb/eraa561). They propose that the inner cortex has the stiffest walls and serves as the growth limiting tissue in maize roots. Perhaps this paper could be cited and the contrary conclusions evaluated.

Response to comment: We appreciate this comment. Indeed, as the reviewer notes, the study of Petrova et al. (2021) notes stiffer inner cortex (adjacent to the endodermis) in cross sections. However, there are two important distinctions to make. First, our "stiff epidermis-soft cortex" model addresses the relative responses of epidermis and cortex cells to the two soil conditions, rather than the mechanical interaction across the tissues. Second, addressing your concern, our ForSys based mathematical analysis across cells of the elongation zone revealed distinct patterns of cell wall stiffness distribution. Specifically, we found that stiffness varies among different cell types, with cell walls of the epidermis becoming stiffer, but softer in cortex cells, supporting our other experimental results (Figure 4f). Variations seen across maize and rice roots may of course also be attributed to species-specific differences, perhaps in context of cell wall contrasts. We have tried to incorporate these points in the revised version of our paper.

(c) The conclusions depend on AFM indentation methods that refer to Fusi et al. (2022), but this article lacks sufficient details for the method to be reproduced. What is the depth of indentation and force limit of the indentation? What is the speed of indentation? The graph in Ext. Fig 8b lacks

numbers on its axes, which is not acceptable. The text states that the root cells are plasmolyzed, yet the methods state that the sliced sections were stored in water. Then why are the cells plasmolyzed? That does not make sense to me. Fusi et al report stiffness in units of pN/nm, which are correct units of stiffness, whereas in this manuscript they are reported in KPa, which are units for a modulus, not stiffness, and indicates the data were fitted to a model, which was not identified. This should be clarified and the fitting method described. The values given are as low as 200 Pa and 300 Pa; this is too soft to represent cell wall material properties, so I am unsure what the values mean and what kind of wall mechanical properties were measured.

Response to comment: We apologize for the lack of information and agree with the reviewer's comments. In response to these concerns, we have provided detailed AFM methodology specifications, including indentation depth, speed, and force limit (line 755-756). We have removed the previous Extended Data Figure 8b as we agree that it did not contribute additional meaningful information to our findings. Regarding the units, we have recalculated the units based on acceptable standards, and now present all results in 'pN/nm' (Figures 4e, 4g, 4h, and Extended Data Figure 9c). We would like to clarify that all AFM measurements were performed in aqueous solution to try to maintain an appropriate environment for the cells and cell walls. We have revised the text to hopefully prevent potential misunderstandings regarding this point.

(d) Can one validly compare stiffness values for epidermis and cortex, as implied by the phrase "stronger epidermis-weaker cortex"? Cell shape, cell connections, and tissue geometry likely influence the indentation results. These geometrical details are very different for epidermis and cortex, but are lacking in the text. Were they factored into the calculation of stiffness?

Response to comment: We would like to clarify that our model is based on the differential responses of epidermal and cortical layers to non-compacted versus compacted soil conditions, rather than a direct comparison between epidermal and cortical cells. Our data show that soil compaction triggers increased cell wall thickness and stiffness in the epidermis, while simultaneously inducing cell wall thinning and reduced stiffness in the cortex. This dynamic response pattern is further supported by the evidence presented in Figure 4f and Extended Data Figure 11. We have therefore changed the statement to "stiff epidermis-soft cortex" instead of the "stronger epidermis-weaker cortex" used in the first submission. We have also revised our manuscript to better emphasize this important distinction.

(e) The cellulose concentrations are based on results from a kit where the principle of the method is not described and no citations for validation are given. In reading the online methods booklet, I can guess that the reducing sugars that are reported may include cellulose and other glycans such as mixed-linked glucan and xylan (I don't think xylan is removed in the procedure). Thus it is difficult for me to assess the soundness of the method. What have the authors done to validate the cellulose measurements? The reported values of cellulose concentration may be subject to error. For instance, if the absolute cellulose amount is unchanged but other substances are accumulated (pectins, solutes, proteins), then the 'concentration' will appear to be reduced. Could this be the basis for the reduced value reported for compacted roots?

Response to comment: Thank you for your comments. We have addressed this concern by implementing a well-established and widely-cited method for cellulose measurement (as detailed in the Methods section; Kumar et al., 2015). Our new results, presented in Figure 3j, consistently demonstrate that soil compaction leads to reduced cellulose content. It is of course also important to put these measurements in the context of other observations, such as a decrease in *CESA* expression and cell swelling, that correspond well with our cellulose measurements. We believe this methodological improvement provides more robust support for our conclusions.

Some additional points:

1) Line 86: The text reads “To establish a connection between the changes in cellulose synthesis...” I did not see measurements of cellulose synthesis or changes in its synthesis.

Response to comment: We thank you for this comment. In the new version, we have measured cellulose content in WT, *arf1-1* and OE-ARF1 roots grown in normal and dense media conditions (Fig. 3j). We have also revised the text to start from the cellulose biosynthesis inhibitor (Indaziflam) results to better connect different aspects of the paper.

2) Extended Fig. 7a shows a cellulose concentration of 0.062 for the *arf1* mutant, which is the same value as the value for WT quoted elsewhere. Shouldn't it be higher than in WT, according the proposed action of ARF1 in suppressing CESAs? And according the results in Figure 4? This seems to be an internal inconsistency.

Response to comment: We agree with the reviewer, and we have therefore changed the cellulose measurement method (as indicated also above) and included the new results. As shown in Figure 3j, our results reveal that under compacted soil conditions, cellulose content significantly decreases in WT roots, while *arf1* mutants maintain higher cellulose levels compared to WT. Notably, OE-ARF1 plants exhibit reduced cellulose content even under non-compacted soil conditions. Indeed, based on our previous results, we established that *OsARF1* is induced in response to soil compaction, affecting rice root elongation and radial expansion. Given that *OsARF1* expression remains relatively low under non-compacted conditions, the compaction stress likely amplifies this regulatory effect, leading to changes in cellulose content.

3) Line 179 “Extended Data Fig. 8a;” - this figure does not show stiffness data. Maybe you mean Fig 8b? But Fig 8 b – needs numbers on the two axes;

Response to comment: We agree with the reviewer and have revised this statement to: “Indeed, we found that the WT, OE-ARF1 and *cesa6* mutants all contained thick epidermal cell walls when grown on compacted media (Extended Data Fig. 10a,b), with little difference across the genotypes. The epidermal wall stiffness was not significantly different between WT and *arf1-1*, and only slightly lower in OE-ARF1 and *cesa6* (Extended Data Fig. 10c).” in line 199-203.

4) Line 390 – “depicted in (b)”... but b is a graph, not an image.

Response to comment: Thank you. We have changed this phrasing.

In summary, this study provides new information implicating inhibition of cellulose synthesis in the root response to compacted soils, but lacks crucial information about cellulose anisotropy and lacks a specific model of how thinner walls and swelling of the cortex leads to root elongation, not merely root swelling (as occurs in the *rsw1* CESA mutant).

Response to comment: We appreciate this comment from the reviewer. In the revised manuscript, we have used ForSys-based inferences of mechanical stresses across tissue/organ sections. This algorithm corroborates the "stiff epidermis-soft cortex" scenario of our other results. We propose that this scenario underpins root responses to compacted media and soil. We note that the organization of cortical microtubules in the epidermal root cells of the elongation zone did not show major differences based on media conditions. As stated above, we would be happy to include these results, and appropriate quantitative data, if the reviewer deems this necessary. We furthermore acknowledge the need to image microtubule organization and dynamics in response to changes in media conditions in future works. While we cannot rule out whether or not the cell wall anisotropy influences the responses to soil compaction of these cells, our data indicate that other mechanisms certainly also are at play. We show through several complementary approaches that the cellulose levels, and cell wall thickness and mechanics, change in a contrasting manner across the root tissue layers when grown on dense/compacted media. This, in turn, supports the root's ability to adjust growth to such conditions.

Referee #3 (Remarks to the Author):

Soil compaction induced ethylene accumulation, leading to root length reduction and radial expansion. This study reports a candidate pathway from compaction to ethylene and ARF1, which causes inhibition of CESA6, leading to root expansion. A *cesa6* mutant is first identified to show longer and swelled roots under compaction. Its upstream regulator ARF1 inhibiting CESA6 expression is then found to have longer and swelled roots under compaction. The root expansion/swelling is likely related to larger cortical cells with thin cell walls and reduced stiffness. Ethylene/ACC induced ARF1 but reduced CESA6 expression. This study provides some novel insights into the regulatory pathway of root expansion growth under compaction stress. However, the relation of the pathway with root elongation under compaction seems not that consistent. The following points should be addressed further.

1, Line 70. The statement that 'the *oscesa6* mutant roots grew better than WT in compacted soil' is not accurate. 'grew longer' may be appropriate. In Fig 1c, the data may be subjected to further statistical analysis for multiple comparison. Then we could know clearly whether the mutant root length is longer than the WT in compact soils.

Response to comment: We appreciate this suggestion. In the revised manuscript, we have modified the term 'grew better' to 'grew longer' in the manuscript (line 77). Furthermore, we applied two-way ANOVA for all datasets requiring multiple comparisons. These analyses support our conclusions. As shown in Figure 1a, the *cesa6* mutant exhibits longer roots as compared to WT specifically under compacted soil conditions. Importantly, this phenotype is not attributed to inherent characteristics of the *cesa6* mutant, but rather represents a specific response to soil compaction, as demonstrated by the results shown in Figures 1c and 1d.

2, Fig 1b, the 'NG-CESA6' is a mutant complementation line. However, its root diameter in cross-section is much smaller than the WT one in compacted soil. Is the complementation affected by the transgene?

Response to comment: We greatly appreciate your suggestion. To address this concern, we have conducted phenotypic analyses using new batches of NG-CESA6 complemented *cesa6* plants, as shown in Figure 1e and Extended Data Figure 2c and 2d. These results demonstrate that NG-CESA6 successfully complements the root elongation and radial expansion phenotypes of the *cesa6* mutant under compacted soil conditions.

3, Fig 3d, since the CESA3 and CESA5 are not significantly induced in mutants or repressed in OE-ARF1 plants, the description in lines 123-124 is not proper. Please clarify.

Response to comment: We are grateful for your suggestion. We have revised to "The expression levels of *CESA5*, *6*, and *8* were significantly elevated in the *arf1-1* mutant but, with the exception of *CESA5*, suppressed in the OE-ARF1 lines. These findings substantiate that OsARF1 functions as a transcriptional repressor of the *CESA* gene family" in line 129-132. As ARF1 did not bind to the *CESA3* promoter, we removed this gene in the new Figure 3c.

4, Although the ARF1 binds to the promoter region of *CESA6* and homologous genes, and inhibits their promoter activity and gene expression for possible root elongation in compacted soil, the genetic analysis of double mutant *arf1 cesa6*, in comparison with the single mutants and WT, should be performed to further confirm the interaction of the two genes in compacted soils.

Response to comment: Thank you for your suggestion. To address this question, we generated *arf1cesa6* double mutants and compared their growth with WT and *arf1* single mutants on media of varying densities. We found that the mutation of OsCESA6 in *osarf1* rescued the penetration defects of *arf1* under both normal and dense media conditions (see Extended Data Fig. 9).

5, In *arf1* mutant, the expression of *CESA6* etc. is increased compared to WT. However, the cellulose content in the *arf1* is very similar to the WT level, in contrast to the expectation that the *arf1* should have more cellulose than WT. I would suggest that the cellulose contents in OE-ARF1 plants and *cesa6* mutant were all measured under non-compact and compact soils, and then compared to those in the WT and *arf1* under the same conditions to see if the cellulose contents are consistent with the authors' conclusion.

Response to comment: We appreciate your suggestions and questions. Following your advice, we have re-measured cellulose content in WT, *arf1*, and OE-ARF1 plants grown under different soil compaction conditions using a different and widely used method (detailed in Methods section). As shown in Figure 3j, our results reveal that under compacted soil conditions, cellulose content significantly decreases in WT roots on dense media, while *arf1* mutants maintain higher cellulose levels compared to WT. Notably, OE-ARF1 plants exhibit reduced cellulose content even under non-compacted soil conditions. Based on our previous results and supplementary experiments (Figures 3a and 3b), we have established that *OsARF1* expression is elevated in response to soil compaction, affecting rice root elongation and radial expansion. Given that *OsARF1* expression remains relatively low under non-compacted conditions, the compaction stress likely amplifies this regulatory effect, leading to more pronounced changes in cellulose content.

6, In Fig3c, the ARF1 gene is induced by high-density gel. The CESAs genes could be examined similarly to see what happens.

Response to comment: We agree with your suggestion. Following your recommendation, we have conducted several additional experiments. First, we analyzed the expression levels of *CESA5*, *CESA6*, and *CESA8* under varying media densities. Our results demonstrate that increased media density leads to reduced expression of these *CESA* genes (Figure 4i). Furthermore, we generated *ProCESA6*:GUS transgenic plants. Here, GUS staining reveals that dense media reduces *OsCESA6* expression in cortex cell layers of elongation zone (Extended Data Fig. 6).

7, The OE-ARF1 plants showed constitutive root swelling. The authors may want to test the cellulose content to see if the ARF1 would inhibit this cellulose level.

Response to comment: We agree with your suggestion. Following your recommendation, we have conducted additional cellulose content measurements in OE-ARF1 plants grown under varying agar densities. Our results confirm that cellulose content in OE-ARF1 plants is indeed lower than that in WT (in Figure 3j).

8, Extended Fig1f, g. The cellulose contents are usually 30-50% in rice. However, in this analysis, the cellulose contents are only 5-6%. It is possible that the method used is not accurate. It would be much more accurate if the contents were measured by GC.

Response to comment: We are thankful for this suggestion. Following your advice, we have adopted a new method for cellulose content measurement (detailed in Methods section). Although our measured values remain below 30-50%, we believe this is biologically reasonable (see for example Xiong et al., 2009). The primary root of young seedlings is a highly active organ with cells at many different developmental stages. Therefore, the cellulose content in these tissues is expectedly lower compared to mature tissues.

9, Line 153-168, the authors should clearly mention whether cell wall thickness has any direct correlation with cell wall stiffness, which may lead to radial expansion of roots.

Response to comment: In the revised manuscript, we have incorporated additional references and discussion to address this relationship (line 175-179).

10, Fig 4b, the cell wall thickness in WT is very similar to that in OE-ARF1 and *cesa6*, while the latter two genotypes grow longer roots in compacted soils. So this cell wall parameter may not be consistent with the longer root phenotype.

Response to comment: We would like to clarify our previous statement. Both OE-ARF1 and *cesa6* mutants exhibit significantly thinner cortical cell walls and more pronounced cell width compared to WT, which is consistent with our conclusions (in Figure 4c).

11, Fig 4d, the meaning or indication of the ratio should be explained.

Response to comment: We greatly appreciate your suggestion. In the revised manuscript, we have removed the previous Figure 4d (also based on reviewer 2's comments). Additionally, we have conducted comprehensive inferences of cell wall stiffness across all cell types in the cross-sectional view using the ForSys algorithm (Figures 4f, Extended Data Figure 11; Methods section). These new measurements support our proposed model, demonstrating increased cell wall stiffness in epidermal cells while showing reduced stiffness in cortical cell walls upon compacted media.

12, Extended Fig 4, the expression of CESA6 and other homologs could be examined to see if they have any difference in different cell types of roots. Then readers may have an idea where the genes may work since WT cortical cells have thin cell wall while the epidermis cells have thick cell walls.

Response to comment: We agree with this important suggestion. As per your comment, we confirm that dense media causes down-regulation of *CESA5*, *6*, *8* in Figure 4i; and we generated *ProCESA6*:GUS transgenic plants. Here, GUS staining revealed that dense media reduces *OsCESA6* in cortex cell layers of elongation zone (Extended Data Fig. 6). Importantly, our bulk RNA-seq analysis corroborates the findings presented in our co-submitted manuscript (see below, NC: Non-compacted soils; CMP: Compacted soils, Zhu et al., 2024), providing additional validation for our conclusions.

13, Fig 5b, ethylene-induced ARF1 expression depends on OsEIN2 but not on OsEIL1. Is it possible that this induction requires both OsEIL1 and OsEIL2?

Response to comment: We are grateful for this insightful comment, and we therefore include the double mutant of *eil1eil2* (please also see comments to reviewer 1 above re analyses on *ein2* and *eil1eil2*). We have rerun assays for both *ein2* and *eil1eil2* in the revised version of our manuscript. The results show that EIN2 may have a larger influence on the expression of *OsARF1* than the EILs; however, both the EIN2 and the EILs may contribute to the suppression of specific *CESAs* on compacted media. We have changed the text and figures accordingly.

14, Fig 5f, the authors found that the module involving compaction-ethylene-ARF1-CESA6 controls root swelling probably through cell wall remodeling. However, the relation of the root elongation with the pathway could not be well explained under compaction stress. The effects of compaction and ethylene are not consistent with those of ARF1 and *cesa6* regarding to root length. Actually, compaction stress and ethylene all induced short but swelled roots as derived from the Science (2021) paper of the same team. However, OE-ARF1 seems to induce longer but swelled roots under compaction, and *cesa6* mutant also had longer but swelled roots under compaction stress. The authors should recheck their data carefully to smooth their pathway for a specific root phenotype.

The authors provided extensive data. However, the connection of these should be further examined in the proposed pathway, especially in terms of root elongation under compaction stress.

Response to comment: We greatly appreciate your comment on this and agree with the point. In the revised manuscript, we have modified previous Figure 5j by incorporating both *OsARF1* expression changes and root length data, expanding on the original model that primarily illustrated root diameter changes (see Fig. 4j). Previous research has shown that *ein2* and *eil1* mutants do not exhibit increased root diameter under soil compaction due to deficient ABA accumulation (Huang et al., 2022). In our study, we focus on processes downstream of ethylene, i.e. that ethylene activates the 'OsARF1-CESAs' pathway to maintain growth. However, ethylene controls many processes, including for example ABA-related responses during root compaction responses, and so it is possible that these other processes then also contribute to the responses seen in the ethylene mutants. We therefore believe our findings are complementary rather than contradictory to previous observations. We have tried to better clarify these relationships in the discussion of our paper.

15, Throughout the MS, all the figures and supplemental data Figures should be further analyzed statistically for multiple comparison, specifically for Fig1cd, Fig3fh, Fig4b-d, and Fig5e.

Response to comment: Yes, we totally agree with this comment. And, in response to your concern, we have revised our statistical testing and applied a two-way ANOVA with Tukey test to re-analyze the significance among multiple groups in Figures 1b, 1f, 3h, 3i, 3j, 4c, and Extended Data Figures 1b, 1d, 2d, 4c, 4e, 5c, 7b, 8b, 9b, 10b and 13b; a one-way ANOVA with Dunnett test in Figures 2f, 3b, 3c, 4e, 4g, 4h, and Extended Data Figures 10c, 12b, and 12c; a Welch's t-test with Bonferroni-Dunn in Figures 2e and 4i; a Welch's t-test in Figures 1d and 3e. The results of this re-analysis confirmed our original conclusions.

Referees' comments:

Referee #1 (Remarks to the Author):

The authors have addressed all issues that I raised and I commend them on their work. A few minor issues:

Line 180 : “compliant walls” unclear what this means

#Response to comment: We thank the reviewer for this comment. We have revised “more compliant walls” to “more flexible walls” on line 189.

Line 197: “Complementary Atomic Force Microscopy (AFM) analysis” not clear

#Response to comment: We agree that this was not clear. We have revised “Complementary Atomic Force Microscopy (AFM) analysis...” to “To further characterize cell wall properties, we harvested root tips from all genotypes grown in dense agar media, prepared 50- μ m thick longitudinal sections and assessed cortex cell wall stiffness using atomic force microscopy (AFM). Consistent with our hypothesis, WT exhibited lower stiffness compared to *arf1-1*, while OE-ARF1, *cesa6*, and *arf1cesa6* displayed the lowest stiffness values” on line 203-208.

Line 215: “stress accumulates in the outer rims” specify outer rims

#Response to comment: We appreciate this comment. Following Reviewer 4's suggestion, we have replaced our ForSys algorithm with Finite Element (FE) modeling and have accordingly completely revised the original statement in the manuscript.

Line 312: “we did not observe any significant trends in BR gene expression under different soil conditions (Zhu et al., co-submitted)³²” as the author list is different, “we” is not accurate

#Response to comment: We apologize for this oversight. We have revised “we did not observe any significant trends in BR gene expression under different soil conditions (Zhu et al., co-submitted)” to “However, Zhu et al. did not observe any significant trends in BR gene expression under different soil conditions” on line 329-330.

Rongfeng Huang lab: Specify affiliation

#Response to comment: Thanks. We have added the affiliation “We thank Prof. Rongfeng Huang (Biotechnology Research Institute, Chinese Academy of Agricultural Sciences, Beijing 100081, China) for providing the rice *ein2* and *eil1eil2* lines” on line 866-868.

We greatly appreciate your recognition of our work and your valuable suggestions.

Referee #3 (Remarks to the Author):

This MS has been improved based on my previous comments. However, the following points may be further addressed for clarity.

1, For extended Data Fig 9, the single *cesa6* mutant should also be included for comparison of phenotypes and root length measurement. After all the data are included, the genetic interaction of the ARF1 and CESA6 genes can be estimated. If the root (compacted) of the double mutant looks like the *cesa6*, then one may say that the CESA6 acts downstream of ARF1, further confirming other results. However, the root length of the double mutant under compacted condition showed no significant difference compared to *arf1* root length under the same condition. Did this mean that the CESA6 does not have strong effects on root length, acting downstream of ARF1? These should be further clarified.

#Response to comment: We greatly appreciate these suggestions. We now compared the primary root length of WT, *arf1-1*, *cesa6*, and *arf1cesa6* together. We found that although primary root length of *arf1cesa6* is significantly longer than WT and more similar to the *cesa6* single mutant phenotype, it is still significantly shorter than the *cesa6* single mutant (new Extended Data Fig. 9). The incomplete reversion to the *cesa6* single mutant phenotype may be due to the fact that ARF1 regulates many other genes (see for example our Y1H results in the ms). Nevertheless, based on these, and several other, results, we conclude that CESA6 acts downstream of ARF1, affecting the ability of rice roots to penetrate compacted soil (see also next point).

Extended Data Fig. 9

a, Root penetration phenotype and length of WT, *arf1-1*, *cesa6* and *arf1cesa6* seedlings grown on normal and dense media conditions for 5 days. The experiment was conducted in triplicate, yielding consistent trends ($n=11-17$ per trial). Scale bar=1 cm. *b*, Statistical analysis of (*a*) was performed from one replicate using two-way ANOVA followed by Tukey test. Different letters denote significant differences at 0.05 level. The p (interaction) <0.0001 .

2, For Fig4 b, c and e, data from double mutant of *arf1 cesa6* is better included for comparison.

#Response to comment: We agree that more detailed analysis of the double mutant phenotype is relevant to confirm the genetic relationship between ARF1 and CESA6. As per your suggestion, we reanalyzed the cell wall thickness of WT, *arf1*, OE-ARF1,

cesa6, and *arf1cesa6*, as well as the the cortex cell wall stiffness in the *arf1cesa6* double mutant (new Fig. 4c-g; see also comments to reviewer 4 regarding new measurements of cell wall thickness in TEM). As shown in the new Fig. 4 (below), we found that *arf1cesa6* reverted the thickened and stiffer cortex cell wall phenotype of *arf1-1*, more closely resembling the *cesa6*. These results further confirm that *CESA6* acts downstream of *ARF1*, regulating the genetic mechanism by which rice roots adapt to compacted soil.

Figure 4:

a, Longitudinal schematic indicating approximate locations for Transmission electron microscopy (TEM) and Atomic Force Microscopy (AFM) force spectroscopy measurements in cortex (blue box) and epidermal (red box) regions. **b**, Schematic cross-section of a rice root cell, highlighting cortex (blue box) and epidermal (red box) cells. **c**, TEM images showing cell wall thickness (red lines) in root epidermis and cortex of 5-day-old WT, *arf1-1*, OE-ARF1, *cesa6* and *arf1cesa6* under normal and dense media. Scale bar=200 nm. **d-e**, Epidermal (d) and cortical (e) wall thickness statistical analysis (15 points per cell in 20 cells per genotype) was performed using two-way ANOVA followed

by Tukey test, different letters denote significant differences at 0.05 level. The p (interaction) < 0.0001. Measurements were repeated four times. **f-g**, Root cortical (f) and epidermal (g) cell wall stiffness in *WT*, *arf1-1*, *OE-ARF1*, *cesa6* and *arf1cesa6* under dense media conditions. Two sections per genotype were analyzed, with the experiment repeated four times (P values [ranges] of Dunnett test are shown). **h-i**, Root cortical (h) and epidermal (i) cell wall stiffness of *Nipponbare* (*Nip*), *ein2*, and *eil1eil2* under dense media conditions. Two radial sections per genotype were analyzed, with the experiment repeated four times (P values [ranges] of Dunnett test are shown). **j**, Expression of *ARF1*, *CESA5*, *6*, and *8* in the elongation zone of *Nip* and *ein2* seedling roots grown under normal and dense media conditions for 5 days ($n=6$ per trail, three biological replicates). Statistics presented are from one representative trial (The adjusted P values of Welch's t -test with Bonferroni-Dunn method are shown). **k**, Soil compaction, and associated ethylene accumulation, induces *OsARF1* expression mainly in the cortex cells, leading to suppressed primary wall *CESA* expression. The subsequent reduction in cellulose biosynthesis leads to radial expansion of cortex cells, and thus root swelling, due to reduced cell wall thickness around cortex cells. By contrast, the epidermal wall thickness, and stiffness, is maintained to support the penetration ability of rice roots in compacted soils. The intensity of the colored walls on epidermal and cortical cells denotes cell wall thickness.

3, The data about the epidermal cell wall thickness and stiffness from Extended Data Fig 10 may be placed in the normal Fig 4. Then readers would easily know the difference between cortex and epidermal cells.

#Response to comment: Thank you. We agree that this will make it easier for readers to understand our conclusions. Therefore, we have included both the epidermis and cortex results in the new Fig. 4 (see revised figure above).

4, Fig 4j, the pathway at the most right needs some clarification. I would suggest that the 'T' symbols indicating negative regulation above cellulose biosynthesis and cell wall thickness may be replaced by arrows indicating positive regulation? Could the authors add cortex cells or epidermal cells here in the pathway? Otherwise, the pathway is quite difficult to understand since the authors mentioned two different cell types namely cortex and epidermal cells.

#Response to comment: Thank you for raising this point, which we certainly agree with. As per your suggestion, we have re-drawn our model pathway in new Fig. 4k (see above). We have separated the pathway in epidermis and cortex, and attempted to clarify every element.

We thank you again for these useful comments, which substantially improved our manuscript.

Referee #4 (Remarks to the Author):

I am commenting on the revised paper entitled “Ethylene modulates cell wall mechanics for root responses to compaction”. I read the initial reviews, author responses, and the revised paper. The initial reviews were comprehensive and constructive, and the authors were responsive. I will focus on follow-up with prior comments rather than launching into anything new. Some of the revisions were un-nerving. New results that contradict the prior work are now introduced, the authors hypothesize that it was due to genetic background effects. Other new results greatly improved the paper, as the original submission contained minimal and barely sufficient results to support the interconnections proposed to mediate the growth response (e.g. including DCB was a good idea). Several new experiments with additional mutants and promoter analyses bolster the case for the ethylene-ARF-CESA transcriptional response components of the paper.

The major weaknesses that remain relate to trying to cesa and arf mutant cell wall phenotypes with the biomechanics of the root growth response. The geometries, tissue layer interactions and biomechanical parameters that drive root swelling are not known, and the authors make a mistake by trying to close this entire loop. It is not possible for any lab to cover all of this ground with limited data in one figure. Despite this major weakness, the work is highly novel and interesting without this poorly executed work in Figure 4.

#Response to comment: We greatly appreciate your comments on the manuscript. In brief, we have now replaced the ForSys inference analysis with FE simulations. We have remeasured the cell wall thicknesses using sections from both in house and from a company, carefully assessing heterogeneity across cell walls and performed blind test measurements. In addition, we have included acknowledgments of assumptions and limitations regarding the AFM measurements. We hope that these revisions address your concerns regarding our work.

Major weaknesses:

1) The cell wall thickness measurements are not reliable and could not be reproduced based on the methods. There is no description of the protocols followed to identify similar regions across genotypes. There is no analysis of the spatial heterogeneity in wall thickness within individual cells, only the aggregated datapoints in Figure 4c. The example images in 4b are at an insufficient magnification to reveal the wall boundaries that are used to measure thickness. Many example images should be shown in the supplemental and the sampling needs to be better justified. The way it is done now is very subjective, creating opportunities for investigator bias in choosing the points for measurement. This type of experiment would be best conducted blind.

#Response to comment: Thank you for these comments. In the previous version of our ms, we used the same types of measurements as those used in (Zhou et al., 2024, <https://academic.oup.com/plcell/article/36/9/3751/7701754>). However, we definitively

acknowledge the limitations you have highlighted and agree that this methodology would greatly benefit from improvement. In the revised version, we provide a detailed protocol for how we first made semi-thin sections of 1 cm long root tips and verified that these targeted the elongation zone (see for example below figure from Pandey et al., 2021, where different zones are readily identified, <https://www.science.org/doi/full/10.1126/science.abf3013>) across all genotypes (see new Extended Data Fig. 10). We have further included zoomed-in images of the epidermal and cortical cell layers in a new Extended Data Fig. 11.

[Redacted figure and text]

Extended Data Fig. 10

*Semi-thin sections (2 μm thickness) of WT, *arf1-1*, OE-ARF1, *cesa6* and *arf1cesa6* lines. Scale bar=50 μm .*

Extended Data Fig. 11

Magnified views of root epidermis and cortex sections from WT, *arf1-1*, OE-ARF1, *cesa6*, and *arf1cesa6* lines. The outermost layer was identified as epidermis, and measurements at intercellular spaces were used to ensure single cortex cell wall thickness (indicated by red boxes). Scale bars: 5 μm for epidermis and 1 μm for cortex regions.

We acknowledge that cell wall thickness around individual cells does indeed vary. To address this, we measured cell wall thickness at eight regions around selected cells, where we measured cell wall thickness at 15 points, and used these data to analyze cell wall width heterogeneity (see example WT cortex cell and data points on cell wall thickness in new Extended Data Fig. 12). Our results confirm thickness variations across the cell wall (see spreads for each region), but our sampling approach yielded no statistical differences in wall thickness across a cell's cell wall. We used this approach to re-evaluate cell wall thickness across 20 cells per genotype (15 points per cell), resulting in 300 measurements per genotype (see new Fig. 4d and 4e).

Extended Data Fig. 12

Cell wall thickness and statistical analysis of individual WT cells, divided into 8 segments to evaluate cell wall thickness variations. Fifteen points were measured per segment to determine cell wall thickness. Scale bar, 200 nm in segmental micrographs, 2 μm in the central micrograph.

Statistical analysis was performed using one-way ANOVA and no significant difference of variance was detected.

Furthermore, to reduce potential subjectivity in the measurements, we firstly used two different sources to perform both imaging and measurements (company vs in house; the company was not informed about our expectations). Both yielded similar results, confirming the reproducibility of our results. Additionally, we asked a colleague who is a co-author on our paper to measure the wall thickness, ensuring that the genotypes were blind. These measurements also led to similar results as our initial results (See Figure below).

A colleague received 100 TEM images (10 per genotype), where cortex and epidermal cells had been indicated, while genotypes were blind. The colleague then measured cell wall thickness at 5 points per cell. We then collated the data, associated the measurements with genotype, plotted the results and performed statistics. Different letters denote significant differences at 0.05 level, p (interaction) < 0.0001 (two-way ANOVA followed by Tukey test).

2) The stiffness values and methods are better described. However, collecting data on selected regions of sectioned material does not capture the stiffness of intact cells and the parameters measured do not necessarily correspond to those that generate the radial swelling response. These limitations need to be fully acknowledged rather than over-sold.

#Response to comment: We agree with this comment and appreciate the reviewer's feedback. We have included a sentence "We acknowledge, however, that the AFM measurements were done on cross sections of fixed cells, which might not directly relate to cell wall stiffness in living root cells." in the AFM segment of the text (line 212 to 214).

3) The links between wall material properties (thickness and stiffness) are unreliable, but even if those data were collected and analyze with more care, using these data and the geometries of the transitions of tricellular junctions to make conclusions about the biomechanical controls of organogenesis is not acceptable. These questions are beyond what can be addressed with the current data and methods in any paper, because the tools to accurately measure stresses and the underlying material properties that are important are not know. The work and conclusions which would only muddy the water for the

field and create confusion. I agree with the reviewer in the first round that was directing the authors in the direction of FE modeling, which has its limitations, but at least the underlying shell theory and ability to accommodate complex geometries enables FE to make plausible and testable predicts. The unpublished mechanical model used here was developed based on the biomechanics of epithelium morphogenesis and is centered on tensional forces at vertices driven by coupling of cortical actomyosin-dependent contraction with the ECM. These mechanics do not apply to the system under study here. The authors would be better served to use their time lapsed data to analyze strain rather than to simulate the biomechanics of the system. The would at least have direct and accurate data on growth rate differences between tissues and among genotypes.

#Response to comment: We greatly appreciate these valuable suggestions. We agree that performing ForSys stress inference may need further characterization and development to make predictions in context of plant biomechanics. We have therefore performed finite element (FE) simulations to investigate how variations in cell wall thickness and stiffness influence the mechanical behavior of root tissues during alterations in turgor pressure and external pressure. In brief, our simulations indicate that thick epidermal cell walls are able to better withstand increases in external pressure compared to thin ones, and that thin cortex cell walls allow for radial root expansion more readily than thick ones.

The details of the modeling and approaches are explained below and in the supplemental data of the revised manuscript:

We first constructed “models” for the root sections. Here, we extracted the geometry of root cells from skeletonized high-resolution transversal microscopic root sections and then optimized these using CAD software (<https://www.rhino3d.com>). Then, the optimized 2D cross section geometry was extruded to a pseudo 3D shell model for further modeling.

To perform FE simulations, we used Abaqus (DS Simulia) static general analysis, which has been widely used for FEM analyses of plant tissues (Zhou et al., 2019, <https://www.sciencedirect.com/science/article/pii/S1537511019308177>). To avoid large computational costs, we used a radial segment of approx. one sixth of the root section, which were considered to represent different cell layers of the root with proper boundary conditions (see Extended Data Fig. 13). In the root section model, all cells are fully separated by cell walls. However, some air spaces associated with tricellular junctions are also present in the pseudo 3D shell model. These air spaces were treated separately when assigning material properties and loading conditions in subsequent steps (see specifics below).

Extended Data Fig. 13

The red points represent the junctions where cell walls meet, specifically the tricellular-junction points where cell walls intersect. The green curves are approximations of the cell walls between pairs of red tricellular-junction points. No distinction was made between cells and air spaces (associated with some tricellular junctions) during geometric modeling. However, the air spaces were treated differently in the subsequent simulation setups.

The radial segments were divided into four cell groups labelled as “inner” (endodermis and stele), “cortex”, “middle” (exodermis and sclerenchyma), and “outer” (epidermis) (Extended Data Fig. 14). Initially, cell walls across the four cell groups were assigned the same initial material properties including elastic modulus. Specifically, the cell walls located at the boundaries between two groups were assigned to one of the adjacent groups: cell walls at the inner–cortex boundary were assigned to the cortex group; those at the cortex–middle boundary were assigned to the middle group; and those at the middle–outer boundary were assigned to the outer group. Since our biological experiments focus on the properties of cortex and epidermis cell walls, we only adjusted the parameters of these groups during different simulations. Using these set-ups, we next explored root deformations when altering external and internal stresses.

Extended Data Fig. 14

The radial segments were divided into four cell groups labelled as “inner” (endodermis and stele), “cortex”, “middle” (exodermis and sclerenchyma), and “outer” (epidermis). Each indicated cellular group is depicted in red while the others are colored in green.

The cell walls were modeled using linear quadrilateral shell (S4R) elements with reduced integration, as the major mechanical entities in our system were the cell walls rather than any other cellular parameters. We set an initial value of cell wall thickness to an arbitrary value of this isotropic model as 1 for all single layer cell walls. Please note that the thickness for adherent cell walls from adjacent cells were manually processed as double wall layers. We applied the material for cell walls with an elastic modulus of 1 kPa which corresponded to our AFM experiments (Poisson’s ratio of close to 0.5 (0.479)), and which has been used in previous FE modelling for primary cell wall mechanics. The default elastic modulus and thickness of the cell walls were treated as plausible initial estimates and were adjusted during the simulation process to explore how variations in stiffness and thickness relate to mechanical behavior.

We first performed FE simulations on compression as perturbation where we expected that the root system would maintain its structural integrity during changes in external pressure. Here, a default value of 0.5 MPa was used for turgor pressure for all root cells (Bassel et al. 2014, <https://www.pnas.org/doi/abs/10.1073/pnas.1404616111>). Only cells were loaded with turgor pressure, while air spaces were not. To counteract this turgor, an equal wall pressure was loaded to the surfaces of outer cell walls with an initial value of 0.5 MPa. To investigate the deformation response of the root cell wall system under varying mechanical constraints, this external pressure was systematically increased in subsequent simulations. Boundary conditions were set as follow: the bottom edges of the “inner” group was constrained, while lateral edges of the radial segments were constrained along corresponding axis. Such configuration made the system immobile to show direct deformation by compression. We first performed a simulation using the default material parameters and default loading conditions as a reference for comparison. We next explored how changes in cell wall parameters would enable the simulated root cell walls to resist increases in external mechanical pressure.

As depicted in Extended Data Fig. 15, elevated external pressure (1 MPa) triggered collapse of 'outer' (epidermal) cells (Extended Data Fig. 15c) that was not observed in default settings (0.5 MPa) (Extended Data Fig. 15b). Notably, this collapse was mitigated by increasing the thickness (Extended Data Fig. 15e) or stiffness (Extended Data Fig. 15d) of the epidermal cell walls. This effect was maintained when we decreased cell wall thickness of cortex cell walls (Extended Data Fig. 15e). Hence, structural modifications to strengthen the epidermal cell walls prevented failure in the FE simulation and substantially reduced overall system deformation.

Extended Data Fig. 15

a, We initialized all cell walls with a uniform thickness of 1 and set the default elastic modulus to 1 kPa. The inner cells were constrained in position, while a uniform turgor pressure of 0.5 MPa was applied individually to all cells (but not air spaces) throughout the segments. Red arrows indicate the external pressure applied to the outer epidermal cell walls. **b-c**, External mechanical pressures of 0.5 MPa (**b**) and 1 MPa (**c**) were applied to the outer epidermal cells. **d-e**, The elastic modulus and cell wall thickness of the epidermis and cortex were adjusted as indicated and the segments were placed under 1 MPa external mechanical pressure.

As we observed radial expansion of roots in compacted soils (or by over-expressing ARF1), we next investigated how cell wall characteristics influenced radial expansion. Such analyses would be best done in temporal simulations where mechanical properties would be altered during the simulations. However, in our set-ups, such approach was not possible. We instead first removed lateral boundary conditions and increased the turgor pressure, introducing a perturbation which allowed the cells to expand against given external pressure. We note that such conditions are not directly applicable to our biological system, but our goal here was to investigate the potential of radial expansion of the segments when we altered cell wall thickness and stiffness. We applied a lateral turgor pressure to the systems to simulate the internal turgor pressure exerted by neighboring cells that were not explicitly modeled (i.e. cells from neighboring radial segments not included in the simulations).

The simulations show the behavior of the root cell wall system while increasing turgor pressure, which was designed to cause force imbalance, with a set external mechanical constraint (0.5 MPa pressure). Variations in the thickness and stiffness of the epidermal cell walls had little effect on the extent of system expansion (as reflected by the simulated root diameter). By contrast, thinner cortex cell walls promoted overall system expansion. While these simulations support a scenario in which cortex wall characteristics influence radial expansion, we acknowledge (also noted above) that the simulations are unlikely to represent real conditions. We have therefore not included

the radial expansion simulations in the revised version, but we are of course happy to do so if the reviewer finds them relevant to the study.

Extended Data Fig

We used an initial cell wall thickness for all cells of 1. Default elastic modulus for all cell walls was set to 1 kPa. The external pressure was set to 0.5 MPa, and the turgor pressure was changed from 0.5 MPa to 1 MPa to induce expansion. The measurements of system size were conducted between the given reference nodes. 2 x indicates double, 0.5 x indicates half.

In summary, our FE simulations indicate that thicker cell walls, or a higher elastic modulus, in the epidermis contribute to reduced deformation, preventing collapse, during increasing external pressure. These responses did not depend on the thickness of the cortex cell walls. In addition, roots with thinner cortex cell walls displayed larger root diameter in response to increased turgor pressure.

We would like to highlight that our new FE simulations involve several assumptions and were not designed to quantitatively delve with the intricate biomechanics nor the dynamic growth processes. Reduced integration shell elements (i.e., S4R) can be more susceptible to numerical artifacts such as occasional excessive folding, which consequently made the system prone to deformation. Nevertheless, the simulations still provide information about the relative mechanical roles of different root cell layers, and their cell walls, during stresses. We again agree with the reviewer that this approach adds a mechanistic dimension to our experimental findings, which the simulations also recapitulate. Accordingly, parts of the FE simulations have been included in the new Figure 4 in the revised manuscript, and other parts in the supplemental material, while the radial expansion simulations have currently been left out.

We thank you for your constructive comments on our manuscript. The comments have strengthened the work and have helped us address key weaknesses. We have carefully incorporated your feedback and believe the revised manuscript represents a substantial improvement. We hope our responses adequately address your concerns and meet your expectations.

Referees' comments:

Referee #3 (Remarks to the Author):

This MS has been improved further based on my comments. I have one comment about the working model. In Fig 4k, the genetic relation in the working model in the most right panel is not consistent with the results, and from this model, the soil compaction/ethylene would inhibit root swelling and penetration. Actually, the relation between 'cellulose synthesis' and 'thinner/softer cortical wall' should be a negative regulatory manner since *cesa6* mutant showed thin cortical wall (Fig4e). A blunt 'T' end should be added above the term 'thinner/softer cortical wall' to replace the arrow. In this way, the 'soil compaction/ethylene accumulation' would finally promote 'root swelling' and 'root penetration ability', largely consistent with the results. Please clarify the model.

Response: We thank the reviewer for this careful observation and valuable suggestion. The reviewer is absolutely correct - the relationship between 'cellulose synthesis' and 'thinner/softer cortical wall' should be depicted as negative regulation, consistent with our finding that the *cesa6* mutant exhibited thinner cortical walls (Fig. 4e).

We have now amended Fig. 4k by replacing the arrow with a blunt 'T' end to properly indicate this inhibitory relationship. We have also updated the corresponding description in the main text (lines 277 to 285) to clarify this regulatory relationship.

We are grateful to the reviewer for all the valuable comments and suggestions provided throughout the three rounds of revision. These constructive feedbacks have definitively enhanced the quality of our manuscript, and we appreciate the reviewer's time and effort in helping us improve this work.

Referee #4 (Remarks to the Author):

The authors took the prior comments seriously and the resubmitted work clearly demonstrates the cell wall thickness phenotype. However the AFM experiment and the FE models developed are not conducted or interpreted in a reasonable way. As a result the overall biomechanical explanation of ethylene-mediated radial growth and the conclusions presented throughout the paper are not substantiated.

1) AFM on sectioned root material is a useful approach to compare the relative stiffness of cell wall domains from different genotypes. However, it cannot be used to estimate the properties of cell walls in vivo. The loading conditions, wall

geometry, and properties in the AFM experiment are completely different from the living state. The stiffness values, estimated by the authors to be in the ~1 kPa range are about 10^4 to 10^5 less than the stiffness measured from native or hydrated and intact cell walls (e.g. Cosgrove, Science, 2021; Turner, Plant Phys. 2021). The AFM should be treated as a phenotyping tool and not a method to extract realistic values for cell wall properties. The authors continue to over interpret the AFM data:

“We acknowledge, however, that the AFM measurements were done on cross
212
sections of fixed cells, which might not directly relate to cell wall stiffness in
living 213 root cells.”

Response: We thank the reviewer for this comment. We fully agree that AFM measurements on sectioned root material are suitable for comparing **relative** cell wall stiffness across genotypes and cell types under controlled conditions, but cannot be used to infer **absolute** mechanical properties of cell walls in vivo. In response, we have thoroughly revised the relevant sections of the manuscript to make it explicitly clear that our AFM data are used solely as a **comparative phenotyping tool**, rather than as a means to extract biophysically realistic stiffness values. We have removed any language that might suggest otherwise and have added a clear acknowledgment of the limitations of the approach (see lines 199–210 in the revised manuscript).

These errors are compounded further by using AFM-generated stiffness values to inappropriately parameterize an FE model (see #2 below).

Response: Following the reviewer's valid criticisms above, we have elected to remove the FE simulations from the article (Please see responses to #2 below).

2) In this paper, at best FE method can be used to determine if a model in which wall thickness in the cortex could alter root expansion patterns is plausible. There are too many unknowns and assumptions in the model for it to “validate”: many wall material parameters are guessed, the geometry and mechanics of the tissue connectivity are not known, the boundary conditions have unknown effects. This point is important because the power of the model is vastly overestimated.

The authors state: “These observations suggest that a “thicker 221 epidermis-thinner cortex” cell wall model facilitates root adaptation to soil compaction. 222 As an attempt to validate this model, we conducted simplified finite element (FE) 223 simulations examining how variations in cell wall thickness and stiffness influence root 224 tissue mechanical behavior during changes in external

mechanical pressure.”

Response: We concur with the reviewer’s comment that *“In this paper, at best FE method can be used to determine if a model in which wall thickness in the cortex could alter root expansion patterns is plausible”*. Indeed, the many unknown parameters—including cell wall material properties and heterogeneity, tissue connectivity mechanics, cellulose microfibril orientation and boundary conditions—make reliable and realistic modeling extremely challenging. Hence, even if the FE simulations would support our biological observations, the reviewer rightly highlights the substantial limitations of this approach. We also note that due to these limitations, the FE modelling would add limited value to the study as the biological results robustly support our main conclusions. Just to reiterate, the main findings/conclusions of our work are:

- A comprehensive regulatory network of transcription factors governing cell wall synthesis.
- Identification of Auxin Response Factor 1 (ARF1) as a direct regulator of cell wall cellulose synthesis in rice roots.
- Demonstration that ARF1 suppresses cellulose synthesis and wall thickness in cortex cells in response to soil compaction.
- Evidence that ARF1 is upregulated by entrapped ethylene, forming the core of the “ethylene—ARF1—cell wall” regulatory pathway.
- Cytological characterization of root thickening, and proposal of the “thicker epidermis—thinner cortex” coordination model for root adaptation.

After careful consideration, and after consultation with the *Nature* editorial team, we have decided to exclude the FE model from the current version of our manuscript. We thank the reviewer once again for prompting us to improve the rigor and focus of the manuscript.

More problematic is the inappropriate parametrization and design of the model. A key parameter, Modulus/E is assigned to be 1 kPa based on the AFM, this is not at all realistic (see above). All of the resulting strains are therefore not physiologically relevant.

Response: We agree with the reviewer’s comment and appreciate the opportunity to clarify this point. In the original FE modeling, many parameters were indeed based on assumptions. To mitigate this, we attempted to anchor the model using our own experimental AFM data. However, we acknowledge that this was an oversight, as the AFM measurements—being conducted on sectioned tissue—do not reflect the mechanical properties of cell walls in vivo. As such, the use of AFM-derived modulus values (~1 kPa) led to inappropriate parameterization and physiologically unrealistic outputs in the model.

Given these limitations, and in line with the reviewer's earlier comments, we have decided to **remove the FE modeling entirely** from the revised manuscript.

In addition to these major weaknesses in model implementation, the author's fail to conduct simulations that directly test the central concept of the paper which is that cortical wall thinning is a plausible mechanism to promote radial expansion. The FE implementation therefore is severely flawed and fails to advance the hypothesis.

Response: We sincerely appreciate the reviewer's thoughtful critique and fully acknowledge that our previous tissue-level FE modeling did not directly test the central hypothesis—namely, that cortical wall thinning promotes radial expansion. As noted in our earlier responses, our cytological and genetic data provide robust experimental support for this mechanism. To maintain the clarity and rigor of the manuscript, we have removed the FE model from the revised version.

We agree that a well-parameterized FE model could offer valuable insights into root mechanics. However, such modeling currently remains limited by the lack of essential mechanical parameters under realistic soil conditions. **We view this as a promising direction for future research** once more comprehensive in vivo data become available.

Minor points.

1) The FE element approach needs further explanation in order for the reader to understand why it is an appropriate method and its demonstrated utility in the field of morphogenesis. The authors could cite its prior use to discover of an importance of cell wall thickness gradients in morphogenesis control (Yanagasawa et al. Nat. Plants , 2015). The key features of the model and how cells are connected needs to be introduced. Each model in any publication should be uniquely named and made publicly available.

Response: We thank the reviewer for the guidance regarding the FE modelling. As mentioned in the response above, we have removed the FE model-related content of the root tissue from our manuscript.

2) To much space is allocated to the cell wall thickness methods. This section can be condensed with methodological details moved to the methods section.

"To test this 189 hypothesis, we examined cortical cell wall thickness of WT,

arf1-1, OE-ARF1, cesa6, 190 and arf1cesa6 in the root elongation zone using transmission electron microscopy 191 (TEM). Tips (1 cm) of roots grown in normal and dense media for five days were 192 chemically fixed and processed. After determining the elongation zone position using 193 semi-thin transverse sections (Extended Data Fig. 10), we prepared ultrathin sections 194 for measuring cortical cell wall thickness (Extended Data Fig. 11). To account for cell 195 wall thickness heterogeneity, we measured a single WT cortical cell divided into eight 196 segments and found that cell wall thickness variations can effectively be averaged by 197 measuring wall thickness at fifteen different points (Extended Data Fig. 12). Using this 198 approach, we found that the wall thickness of WT cortical cells decreased significantly 199 under dense media conditions compared to normal conditions (Fig. 4a-c,e). By contrast, 200 arf1-1 maintained thick root cortex walls in both non-compacted and compacted 201 conditions, whereas OE-ARF1, cesa6, and arf1cesa6 consistently exhibited thinner 202 walls regardless of growth conditions (Fig. 4a-c,e). To further characterize cell wall 203 properties, we harvested root tips from all genotypes grown in dense agar media, 204 prepared 50- μ m thick longitudinal sections and assessed cortex cell wall stiffness using 205 atomic force microscopy (AFM).”

Response: We thank the reviewer for this suggestion. We have now condensed this section in the main text (line 190 to 210) and moved the methodological details to the Methods section (line740 to 747) as suggested.

We are grateful to the reviewer for all the valuable comments and suggestions provided throughout the two rounds of revision. We would also like to thank the reviewer for inputs regarding the FE modelling. The constructive dialogue has ensured improved quality of our manuscript, and we very much appreciate the reviewer's time and effort in this regard.

Referees' comments:

Referee #3 (Remarks to the Author):

The working model has been corrected based on my previous comment and I have no further comments.

Response: We greatly appreciate the time and expertise you dedicated to helping us strengthen our work, and thank you for your thorough and constructive feedback throughout the four rounds of review. Your insightful comments have significantly improved the quality and clarity of our manuscript.

Referee #4 (Remarks to the Author):

The authors have responded to comments regarding over interpretation of the AFM results. The FE has been removed. This was an unfortunate waste of time. The FE could have been useful, but it is a lot to take on. I understand the change. The cost of this is that there is no test of the biomechanical plausibility of this system in terms of explaining root adaptation to compaction. This gap should be integrated into the paper and two suggestions in this vein are provided below. Otherwise my concerns have been addressed and the paper is now greatly improved.

Response: Thank you very much for your guidance on our manuscript, including your constructive suggestions on cell wall statistics, AFM descriptions, the FE model proposal and other comments. We also appreciate your understanding regarding our removal of the FE model section, which will certainly become a direction we strive to tackle in our future work. We are deeply grateful for your help in improving our research.

1) The first sentence of the conclusion is strange. The paper does not demonstrate how the model responds to compaction. The FE would have been the first step in that direction. The paper aggregates the data to generate a plausible model of how root response to compaction occurs.

275 Our study provides new conceptual insights into how roots adapt to soil compaction 276 stress through cortical cell expansion, demonstrating how a "thicker epidermis-thinner 277 cortex" model effectively responds to soil compaction (Fig. 4k).

Is something like this more accurate?

275 Our study provides new conceptual insights into how roots adapt to soil compaction 276 stress through cortical cell expansion, and provides a plausible "thicker epidermis-thinner 277 cortex" model of how the root effectively responds to soil compaction (Fig. 4k).

Response: Thank you for pointing out the imprecise wording in our conclusion. We agree that your suggested revision more accurately reflects what our study demonstrates. We have made the corresponding changes in the manuscript on line 275 to 276.

2) Again in the conclusions section. The authors add the nice analogy:

“The " thicker 289 epidermis-thinner cortex " model therefore parallels engineering principles used in pipe 290 design, where maintaining perimeter stiffness while increasing diameter improves 291 structural stability.

This is where a sentence about the need for future testing/analyzing of the biomechanical plausibility of this model in roots should be added. It is not proven in this paper.

Response: Thank you for your valuable suggestion regarding the need to acknowledge the biomechanical testing gap in our conclusions. We have added a sentence “Future biomechanical testing and modeling, such as finite element modeling, will be well suited to explore the plausibility of this mechanical scenario in roots.” in the conclusion part on line 290 to 292.

We sincerely thank you for your constructive revision suggestions throughout these review rounds. These valuable comments have undoubtedly enhanced the quality of our work.